# Pluripotency factors determine gene expression repertoire at zygotic genome activation

Meijiang Gao[1,2,11], Marina Veil[1,11], Marcus Rosenblatt [3], Aileen Julia Riesle[1], Anna Gebhard[1], Helge Hass[3], Lenka Buryanova[1], Lev Y. Yampolsky[4,5], Björn Grüning [6,7], Sergey V. Ulianov[8,9], Jens Timmer[2,3] & Daria Onichtchouk [1,2,10 ✉]

Awakening of zygotic transcription in animal embryos relies on maternal pioneer transcription factors. The interplay of global and specific functions of these proteins remains poorly understood. Here, we analyze chromatin accessibility and time-resolved transcription in single and double mutant zebrafish embryos lacking pluripotency factors Pou5f3 and Sox19b. We show that two factors modify chromatin in a largely independent manner. We distinguish four types of direct enhancers by differential requirements for Pou5f3 or Sox19b. We demonstrate that changes in chromatin accessibility of enhancers underlie the changes in zygotic expression repertoire in the double mutants. Pou5f3 or Sox19b promote chromatin accessibility of enhancers linked to the genes involved in gastrulation and ventral fate specification. The genes regulating mesendodermal and dorsal fates are primed for activation independently of Pou5f3 and Sox19b. Strikingly, simultaneous loss of Pou5f3 and Sox19b leads to premature expression of genes, involved in regulation of organogenesis and differentiation.

[1] Department of Developmental Biology, University of Freiburg, 79104 Freiburg, Germany. [2] Signalling Research Centres BIOSS and CIBSS, 79104 Freiburg, Germany. [3] Institute of Physics, University of Freiburg, 79104 Freiburg, Germany. [4] Department of Biological Sciences, East Tennessee State University, Johnson City, TN 37614-1710, USA. [5] Zoological Institute, Basel University, Basel CH-4051, Switzerland. [6] Department of Computer Science, University of Freiburg, 79110 Freiburg, Germany. [7] Center for Biological Systems Analysis (ZBSA), University of Freiburg, 79104 Freiburg, Germany. [8] Institute of Gene Biology, Russian Academy of Sciences, Moscow, Russia. [9] Faculty of Biology, M.V. Lomonosov Moscow State University, Moscow, Russia. [10] Koltzov Institute of Developmental Biology RAS, 119991 Moscow, Russia. [11] These authors contributed equally: Meijiang Gao, Marina Veil. ✉email: daria.onichtchouk@biologie.uni-freiburg.de

Following fertilization, the differentiated cells, egg, and sperm, are reprogrammed into the totipotent state of the zygote. The zygotic genome initially remains silent. It awakens through a process known as maternal-to-zygotic transition (MZT), during which the degradation of maternal transcripts is coordinated with zygotic genome activation (ZGA). In the current model of ZGA, the gradual increase in the ratio of transcriptional activators to transcriptional repressors, accompanied with local changes of chromatin accessibility create a permissive environment for ZGA to occur[1]. In zebrafish, *Xenopus* and *Drosophila*, where development starts with rapid cell cycles, excessive maternal core histones serve as general transcriptional repressors before ZGA[2–4]. Several types of activators are translated before ZGA and reach critical levels at ZGA, including basal transcription factors[5], the regulators of H3K27ac enhancer mark[6], and maternal enhancer-binding transcription factors (TFs). TFs that broadly activate zygotically expressed genes have been identified in *Drosophila*[7], zebrafish, *Xenopus*, and mammals[8]. In lower vertebrates, zygotic transcription is activated by homologs of mammalian pluripotency factors: Pou5f3, Sox19b, and Nanog in zebrafish[9,10], Pou5f3 and Sox3 in *Xenopus*[11].

Nucleosome positioning plays a dominant role in regulating genome access by TFs. The widespread action of genome activators is thought to result from their ability to function as pioneer factors, first displacing nucleosomes so that other TFs can bind[12]. Indeed, reduction or loss of genome-activating TFs in *Drosophila*, zebrafish, and *Xenopus* resulted in the decreased chromatin accessibility on their binding sites[13–16]. Out of them, direct pioneer binding to nucleosomes was demonstrated thus far only for *Drosophila* genome activator Zelda[17,18]. The mechanisms underlying nucleosome-displacing activity of zebrafish and *Xenopus* activators are less clear: they may bind to nucleosomes similarly to their mammalian homologs[19], or compete with nucleosomes for DNA binding[20], or both.

Mammalian POU5F1 and SOX2 reprogram somatic cells to pluripotency and are in several cases sufficient for reprogramming[21]. The mechanisms underlying their partnership in vivo are still not resolved. Until recently, POU5F1 and SOX2 were thought to act cooperatively, binding as heterodimers to bipartite *sox:pou* cognate motifs[22]. This view was challenged by Soufi et al. (2015)[19], who demonstrated that POU5F1 and SOX2 target distinct motifs on the nucleosome-wrapped DNA[19], and by four studies that suggested different scenarios of how POU5F1 and SOX2 interact with each other and with chromatin in embryonic stem (ES) cells. These scenarios are: (1) assisted loading, whereby SOX2 first engages the target DNA, then assists the binding of POU5F1[23]; (2) negative reciprocity, where POU5F1 and SOX2 sometimes help and sometimes hinder each other in binding to the genome[24]; (3) conditional cooperativity of POU5F1 and SOX2 binding, depending on the motif positions in the nucleosomal context[25] and (4) independent binding, even at co-occupied sites[26].

In the case of zebrafish genome activators Pou5f3 and Sox19b, it remains an open question how their broad nucleosome-displacing activity at ZGA relates to their different functions later in development, as judged by their distinct loss-of-function phenotypes. Maternal-zygotic Pou5f3 null mutants MZ*spg* have abnormal epiboly and arrest during gastrulation[27]. The quadruple morpholino knockdown (QKD) of redundant SoxB1 family members (sox19b, sox19a, sox3, and sox2) leads to severe defects during organogenesis, with the first morphological defects visible at the end of gastrulation[28]. The relatively late QKD phenotype is at odds with the earlier roles for Sox19b protein suggested by dominant-negative approaches[29], and by combined knockdowns of SoxB1 genes with Nanog and/or Pou5f3[9].

The mechanisms of Sox19b activity at ZGA and its molecular connection to Pou5f3 remain poorly understood. In this study, we use maternal-zygotic (MZ) Sox19b, Pou5f3, and the double mutants to investigate how two zygotic genome activators interact in vivo. We show that Pou5f3 and Sox19b act as independent pioneer factors and on different motifs. We further dissect the contribution of each factor to the early gene expression.

## Results

### Maternal-zygotic Sox19b mutants are delayed in gastrulation but develop normally.

To abolish the expression of Sox19b in zebrafish, we introduced a mutation in *sox19b* using gene disruption via TALEN[30] (Fig. 1a). The MZ*sox19b* embryos lacking both maternal and zygotic Sox19b, and M*sox19b* embryos lacking maternal Sox19b developed into fertile adults, albeit more slowly than controls, and were smaller in size (Movie S1).

The zebrafish midblastula transition (MBT) begins at cell cycle 10, at 3 h postfertilization (hpf). MBT is characterized by cell cycle lengthening, loss of cell synchrony, activation of zygotic transcription (referred to as the major wave of ZGA), and appearance of cell motility[31]. The duration of the pre-MBT cell cycles was the same for MZ*sox19b* and wild-type embryos (Fig. 1b, Fig. S1), but the appearance of morphological landmarks of subsequent development was delayed in MZ*sox19b* (Fig. 1c, Fig. S2).

Normal development of MZ*sox19b* embryos can be plausibly explained by the presence of zygotic SoxB1 members, sox19a, sox3, and Sox2 (Fig. 1d). The TALEN-induced *sox19b* mutation resulted in a premature stop codon before the first intron of *sox19b*; nonsense-mediated mRNA decay in this type of mutants can trigger a compensatory response by upregulation of the genes that exhibit sequence similarity with the mutated gene's mRNA[32]. To investigate if the transcription of SoxB1 genes is changed in the MZ*sox19b* mutant, we quantified the levels of *sox19b*, *sox19a*, *sox3*, and *sox2* by RNA-seq (Fig. S3a). *sox19b* maternal message was reduced 15-fold already before MBT, indicating that nonsense-mediated decay takes place. Although *sox19a*, *sox3*, and *sox2* bear the closest sequence similarity to *sox19b*, we did not detect compensatory upregulation of these genes which were instead rather delayed in MZ*sox19b* (Fig. S3a–c).

To address if maternal Sox19b protein masked an early requirement for SoxB1 in quadruple knockdown (QKD) experiments[28], we injected Sox3, Sox19a and Sox2 morpholinos into MZ*sox19b* mutant embryos (triple knockdown, or TKD). MZ*sox19b*-TKD and wild-type-QKD embryos showed similar developmental defects in tail bud formation, anterior–posterior axis elongation, and neural system development (Fig. 1e, Fig. S3d–f). The MZ*sox19b*-TKD phenotype could be completely rescued by co-injection of *sox19b* mRNA (Fig. 1f, Fig. S3g). We concluded that combined zygotic activity of Sox2/3/19a/19b proteins becomes critical for the embryo starting from the end of gastrulation.

### Double MZ*sox19bspg* mutants are dorsalized.

To investigate the early requirements for maternal Sox19b and Pou5f3, we obtained a double mutant MZ*sox19bspg* by crossing MZ*sox19b* to Pou5f3 null-mutant MZ*spg*[m793] [27]. MZ*spg*[m793] mutants develop severe epiboly defects[33,34] and are weakly dorsalized[35]. Epiboly defects in double mutant were similar to MZ*spg* while dorsalization was stronger (Fig. 2a, Movie S1). We produced maternal-only M*sox19bspg* mutants by fertilizing the mutant eggs with wild-type sperm and used these to determine if a combined maternal contribution of Pou5f3 and Sox19b is critical for epiboly or dorso-ventral (D/V) patterning. Single maternal mutants developed normally. The double mutant embryos were severely dorsalized, as judged by radially expanded domains of dorsal markers Noggin1[36] and Chordin[37] (Fig. 2b), and severe embryonic phenotypes (Fig. 2c and Movie S1).

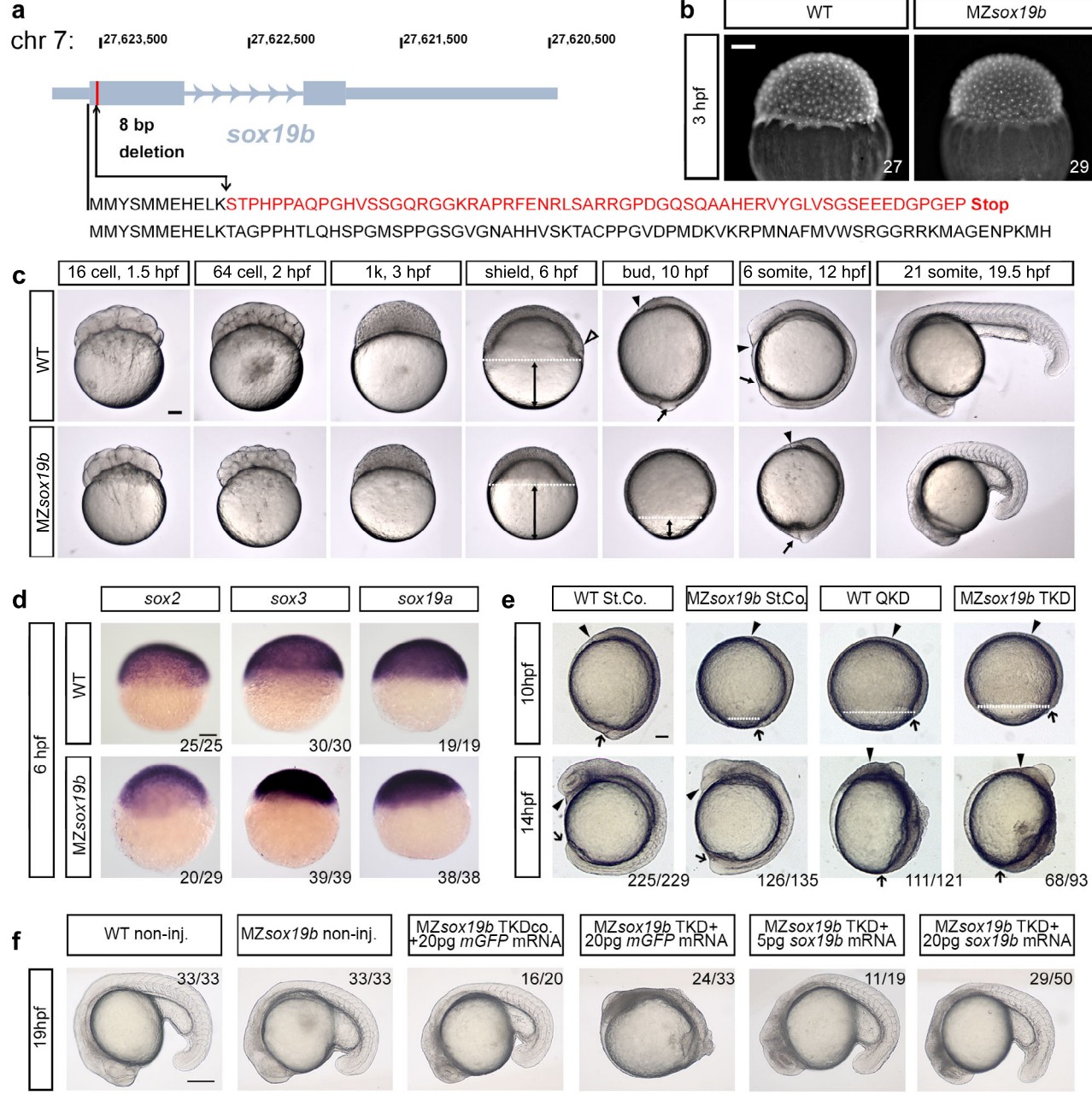

**Fig. 1 Redundant activity of SoxB1 family factors Sox19b, Sox19a, Sox2 and Sox3 is required at post-gastrulation developmental stages. a** Disruption of the *sox19b* gene on chromosome 7 by introducing an 8 bp deletion. **b** No difference in cell division rates between WT and MZ*sox19b* was observed prior to MBT (see also Fig. S1). **c** MZ*sox19b* embryos are delayed in gastrulation (see Fig. S2 for statistics). Simultaneously collected WT and MZ*sox19b* embryos were let to develop at 28,5 °C, pictures of representative embryos were taken at the indicated time points/developmental stages of the wild type. In zebrafish embryos, embryonic shield forms at 6 hpf at the dorsal side during gastrulation (hollow arrowhead in the wild type). MZ*sox19b* embryos are still phenotypically at 40% epiboly (blastula). Gastrulation ends with tail bud formation at 10 hpf. MZ*sox19b* embryos are still at 80–90% epiboly gastrula stage. **d** In situ hybridization for *sox2, sox3, and sox19a*, in WT and MZ*sox19b* embryos, lateral views. **e** Quadruple Sox19a/b, Sox2, and Sox3 knockdown embryos complete gastrulation, but show later defects in tail bud formation and axis elongation. 1-cell stage wild-type or MZ*sox19b* embryos were injected with control morpholino (StCo), or QKD (quadruple knockdown) mix (Sox2, Sox3, Sox19a, and Sox19b morpholinos), or TKD (triple knockdown, Sox2, Sox3, Sox19a morpholinos), as indicated. **f** Axis elongation defects in MZ*sox19b*-TKD are rescued by injection of *sox19b* mRNA. 1-cell stage MZ*sox19b* embryos were injected with either TKD mix or TKDco mix (Sox2, Sox3, Sox19b morpholinos), together with Sox19b or control GFP mRNA, as indicated (see Fig. S3d–g for additional statistics). Double black arrows show the distance from epiboly border (white dotted line) to the vegetal pole. Black arrow—tail bud, black arrowhead—head process. Scale bar 200 μm. Scale bars in **b**–**e**: 100 μm, in **f**: 200 μm.

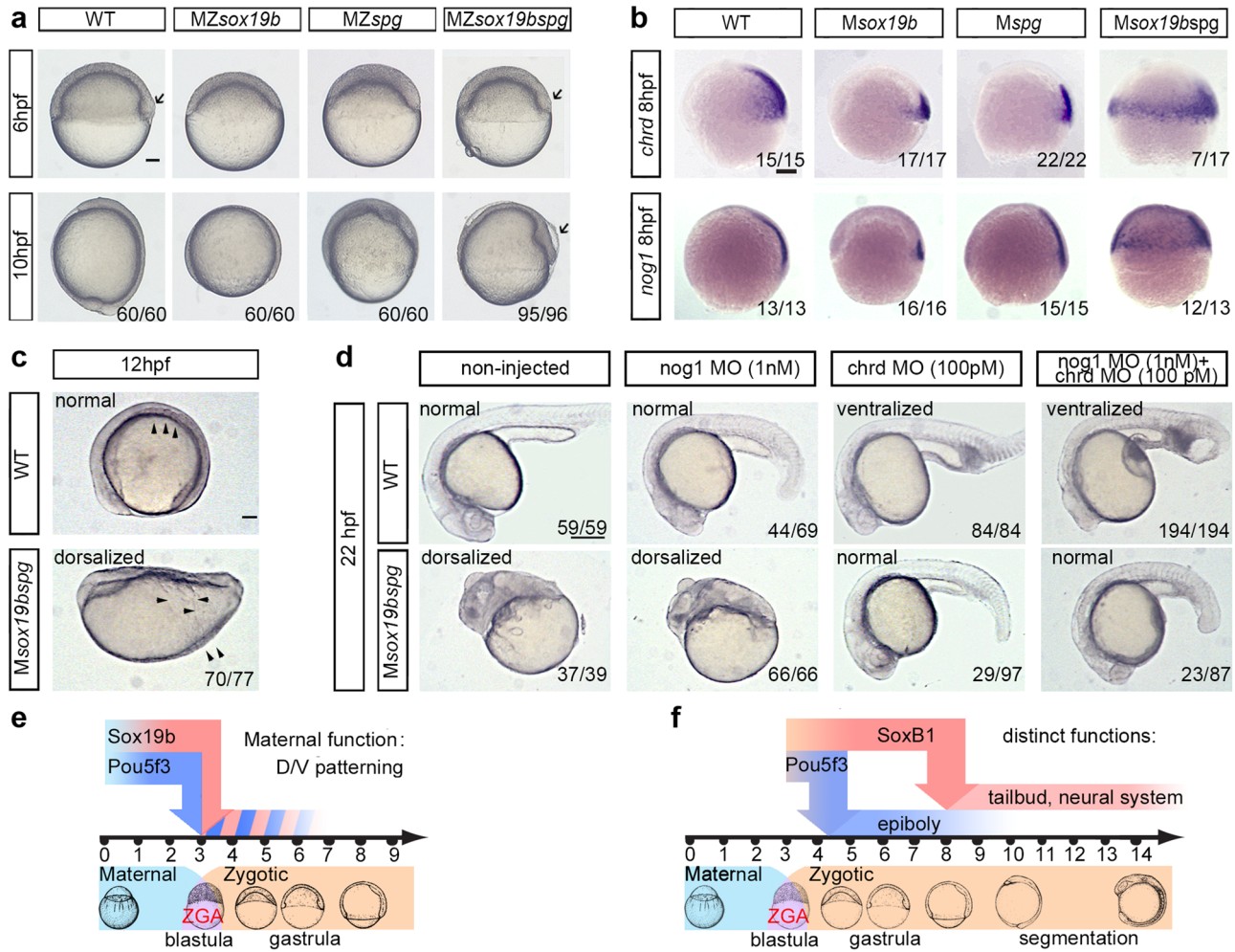

**Fig. 2 Maternal Pou5f3 and Sox19b safeguard correct dorso-ventral patterning. a** Comparison of the single mutants, double mutant and wild-type embryos. 10 hpf: MZ*sox19bspg* mutants are arrested in gastrulation similarly to MZ*spg*. Arrow shows abnormally enlarged shield in MZ*sox19bspg*. **b, c** Double maternal mutants M*sox19bspg* are dorsalized. **b** In situ hybridization for dorsal markers *noggin1* and Chordin, lateral views, dorsal to the right. Note the circumferential expansion in the M*sox19bspg*. **c** Somites (arrowheads*)* form on the dorsal side in WT, but spread over the M*sox19bspg* embryo. Dorsal up, anterior to the left. **d** Normal development of M*sox19bspg* mutants can be rescued by reducing Chordin, but not Noggin1 levels. The wild-type or M*sox19bspg* embryos were injected with the indicated morpholinos or non-injected. The numbers show the ratio of embryos with indicated phenotype/ all embryos alive at 22 hpf. The arrows show abnormally expanded blood progenitor cells in the ventralized wild-type embryos. Anterior to the left, dorsal up. **e, f** Combinatorial (**e**) and distinct (**f**) functions of Pou5f3 and SoxB1. **e** Maternal Sox19b and Pou5f3 safeguard correct D/V patterning. **f** Pou5f3 is critical for epiboly and gastrulation, redundant action of zygotic SoxB1 (Sox19a, Sox19b, Sox2, and Sox3) is critical for organogenesis. Scale bars in **a**, **b**, **c**: 100 µm, in **d**: 200 µm.

In zebrafish, similar to other vertebrates, Chordin blocks the flow of BMPs to the dorsal side of the embryo. The action of multiple gene products within dorso-ventral self-regulatory network converges on defining the size of the Chordin domain[38]. Reduction of Chordin levels by morpholinos was sufficient to rescue M*sox19bspg* phenotype to normal (Fig. 2d). This result suggests combinatorial action of maternal Pou5f3 and Sox19b, i.e. two TFs act additively inducing ventral regulators, and/or repressing dorsal regulators. The combined changes override the self-regulatory capacities of dorso-ventral gene network, while changes in single mutants can be buffered (Fig. 2e). In addition, Pou5f3 becomes essential at the beginning of gastrulation, and Sox19b factors at the end (Fig. 2f).

**Sox19b and Pou5f3 activate ventral genes and are dispensable for dorsal genes.** To characterize the mutant transcriptomes, we performed time-resolved RNA-seq analysis of wild-type, MZ*sox19b*, MZ*spg,* and MZ*sox19bspg* embryos. The embryos

were collected starting from 2.5 hpf (pre-MBT) every 30 min until 6 hpf (Fig. 3a, Fig. S4a).

Two processes shape the transcriptional landscape of embryos at ZGA: the burst of zygotic transcription and regulated decay of maternal mRNAs. About 70% of zebrafish zygote mRNAs are maternally loaded[39], so that the mRNA present in the embryo early time points is a mixture of maternal and zygotic transcripts for most genes. To account for the maternal and zygotic differences between the wild type and mutants, we developed a tool for dynamic RNA-seq data analysis, which we called RNA-sense. The RNA-sense 3-step analysis is explained in the Movie S2 and the Methods. At the first step, RNA-sense builds a time profile for each expressed transcript in one condition (i.e. wild type), and tests if the transcript abundance grows or decays significantly. Dynamic transcripts are then sorted to non-overlapping groups by the time point of switch UP or switch DOWN. For all the genes in switch UP groups, zygotic increase in the transcript levels exceeds maternal RNA decay.

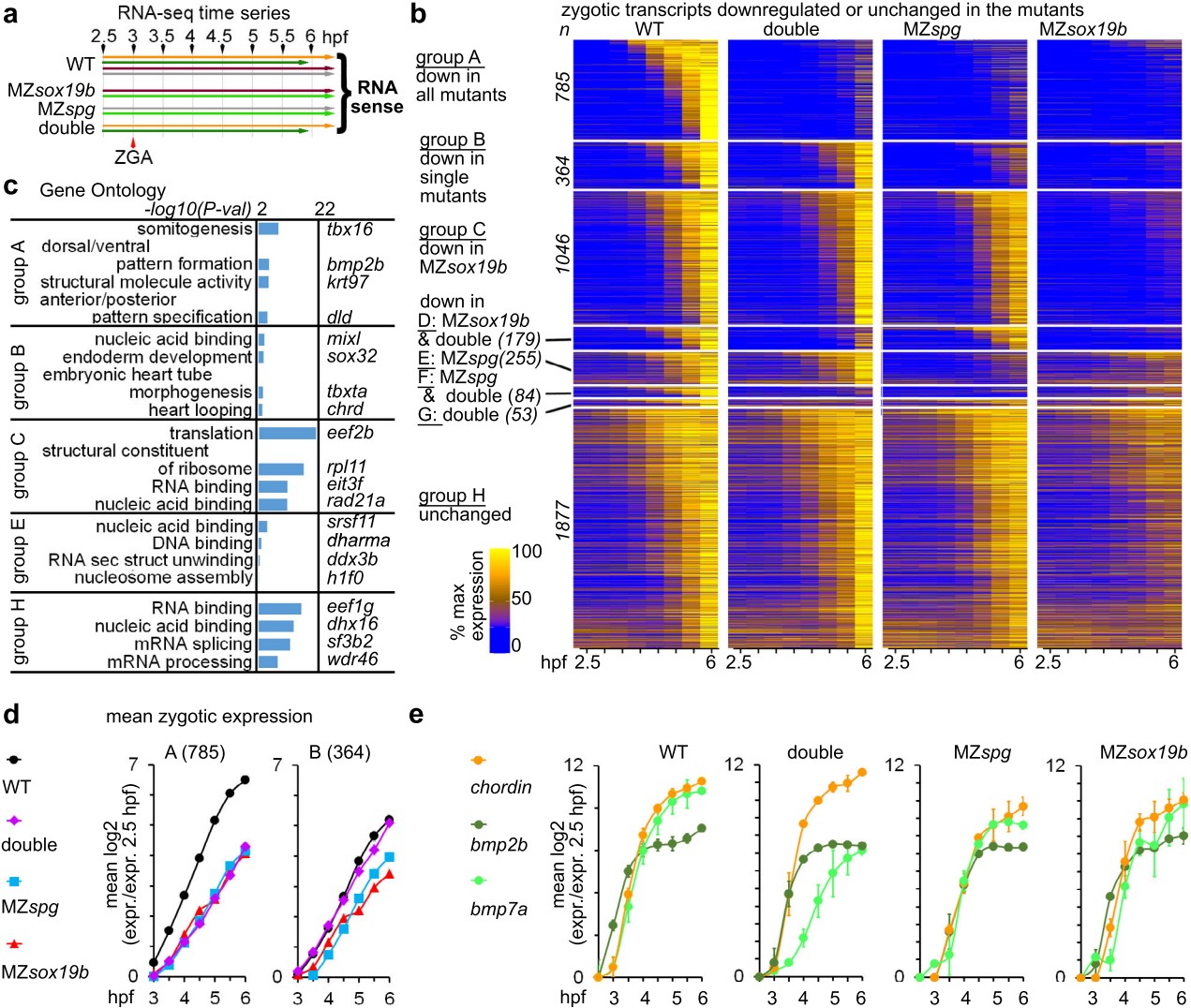

**Fig. 3 Pou5f3 and Sox19b non-additively activate transcription. a** Experimental setup: RNA-seq time series included eight time points, four replicates for the wild type, and two replicates for each MZ*spg*, MZ*sox19b*, and double mutant MZ*sox19bspg* (see Fig. S4a for the details). Time series, where material was collected in the same experiment, have the same color. **b** Heatmap of all 4643 zygotic transcripts in the indicated genotypes. 2766 transcripts were downregulated in the mutants (groups A–G),1877 transcripts were not (group H). Zygotic transcripts were sorted by ascending switch time and switch *p*-value in the wild type. *n* number of transcripts. **c** Enrichment in Gene Ontology terms (DAVID). Top four categories per group and example genes are shown. GO: enrichment for "structural molecule activity" in group A was due to the battery of eight keratins, activated by Klf17 in the epithelial layer[46]. **d** Non-additive (group A) and compensatory (group B) effects of Pou5f3 and Sox19b on the earliest zygotic transcription. Mean zygotic transcription profiles for the groups A and B, relative to 2.5 hpf. **e** Chordin/BMPs ratio is increased in the double mutant, but not in the single mutants throughout the time curve. Note that the transcription start of *bmp2b* and *bmp7a*, but not Chordin is delayed in the double mutant compared to the WT. In **d**, **e**: *n*(wt) = 4, *n*(MZ*spg*) = 2, *n*(MZ*sox19b*) = 2, *n*(double) = 2, where *n* is a number of biologically independent experiments. Data are presented as mean values ± SEM. Source data for **b**, **d**, and **e** are provided as a Source Data file. Source data for **b**–**d** are provided as Dataset S1.

The switch UP and switch DOWN groups in the wild type were in agreement with the zygotic and maternal transcript groups identified in three previous studies[9,39,40] (Dataset S1, Fig. S4b), and are referred below as "zygotic" and "maternal".

Analysis of differential expression revealed delays in zygotic transcription (or absence of transcription for some genes) and maternal mRNA degradation in all mutants. Unexpectedly, we detected stronger delays in MZ*sox19b* than in the other mutants (Fig. 3b, Fig. S4c, S5a, b). Out of 4643 zygotic transcripts, 51% were at least 2-fold downregulated in MZ*sox19b*, 32% in MZ*spg*, and 24% in the double mutants. This result implied that Pou5f3 and Sox19b compensate some of each other's effects on the zygotic transcription.

We divided all zygotic genes into eight groups, A–H, based on the patterns of downregulation in the mutants (Fig. 3b,

Dataset S1). Groups A and B were enriched in distinct developmental regulatory gene categories (Fig. 3c) which we discuss below; the other groups were enriched in general housekeeping categories or showed no enrichment.

Group A transcripts (17% of zygotic genes, downregulated in all mutants) included major regulators of ventro-posterior development: BMP pathway ligands *bmp2b*[37] and *bmp4*[41], the ventral transcription factors *vox, vent* and *ved*[42,43], regulators of ventrolateral mesoderm and ectoderm patterning *draculin, gata2, tbx16* and *vgll4l*, which start their expression at the ventral side or in the ectoderm[44]. Group A also included known direct transcriptional targets of Pou5f3: *mych*[45] and *klf17*[46]. Additional genes related to the BMP pathway were repressed in the double mutant and in one of the single mutants: the second major BMP

ligand *bmp7a*[47] was repressed in the double mutant and MZ*sox19b* (Group D), BMP targets *klf2b*[46] and *foxi*[48] in the double mutant and MZ*spg* (group F).

Group B transcripts (8% of all zygotic genes, downregulated only in the single mutants) were enriched in the category "endoderm development". Group B included the main critical components of the early endomesoderm and dorsal specification network: *mixl1*[49], *sox32*[50], *sebox*[51] *dusp4*[52], *tbxta*[53], and *foxa3*[54], which are activated by the maternal transcription factor Eomesodermin and by Nodal signaling[55,56]. Group B also included double Nodal and maternal β-catenin targets Chordin[38] and *noto*[57], and the maternal β-catenin target *hhex*[58]. In contrast to group A, group B transcripts start their expression at the dorsal side of the embryo, or in the yolk syncytial layer (YSL).

To characterize the earliest effects of Pou5f3 and Sox19b in all groups, we tested if the expression in the wild type was significantly different from the mutants from 3 to 4.5 hpf. The group A transcripts were non-additively downregulated in all mutant genotypes starting from the beginning of the major ZGA wave: the transcripts levels were not statistically different in three mutants until 4.5 hpf (Fig. 3d, Fig. S5c). We concluded that Pou5f3 and Sox19b are required for activating group A zygotic transcription on the ventral side and act either redundantly or sequentially.

The levels of group B transcripts were significantly downregulated from at 3 to 4.5 hpf in the single mutants only (Fig. 3d, Fig. S5d). We concluded that group B transcripts do not require Pou5f3 and Sox19b for their expression but are sensitive to Pou5f3/Sox19b balance starting from ZGA onset. Similar analysis of the groups C–F (Fig. S5f–i, Table S1) suggested non-additive and compensatory effects of Pou5f3 and Sox19b on early transcription.

Ventral (group A) zygotic transcripts were expressed to significantly higher level than dorsal (group B) in the wild-type embryos at 3–4.5 hpf. In contrast, the double mutant expression of dorsal transcripts was higher than that of the ventral (Fig. 3d, Fig. S5e). We wondered if the severe dorsalization phenotype of the double mutants could be explained by the increase in the ratio of Chordin (group B) to *bmp2b* (group A) and *bmp7a* zygotic transcripts at ZGA onset. Indeed, the transcription of BMPs was reduced relative to Chordin in the MZ*sox19bspg* embryos, but not in the single mutants, from the beginning of major ZGA (Fig. 3e). The ratio of Chordin to BMPs is critical for the size of the dorsal domain, hence, the early transcriptional disbalance between these factors in MZ*sox19bspg* embryos plausibly explained their phenotypic dorsalization.

In sum, we demonstrated that Pou5f3 and/or Sox19b activate 24% of zygotic transcripts (groups A, D, F, G) including but not restricted to the components of BMP signaling pathway, ventral genes, and ectodermal genes. Pou5f3 and Sox19b are dispensable for the activation of mesendodermal regulators, targets of Eomesodermin/Nodal, starting their expression on the dorsal side of the embryo.

**Differentiation genes are prematurely expressed in MZ*sox19bspg*.** We next inquired if Pou5f3, Sox19b, or both of them suppress zygotic transcription for certain genes. We scored zygotic transcripts which were upregulated in the mutants (Dataset S2), and divided them into three non-overlapping groups by the strongest upregulation in one of the genotypes (I-K, Fig. 4a). Transcripts upregulated in MZ*sox19bspg* (Group I) were enriched in developmental and regulatory ontologies (Fig. 4b) and included two categories: transcription factors such as *pax8, dlx5a, nkx6.3*, and eight T-box factors (*tbx1, tbx2b, tbx3a, tbx6, tbx16l, tbx18, tbx20, tbxtb*) that are involved in tissue

differentiation and were normally expressed in the wild type starting later than 6 hpf[44], and dorsal genes like *nog1, noto*, and *hhex*, which were expressed in the wild type at lower levels. In MZ*spg* and in MZ*sox19bspg*, but not in MZ*sox19b*, selected transcripts were upregulated starting from the major ZGA onset (Fig. 4c).

In sum, we show that Pou5f3 and Sox19b together prevent the premature expression of late differentiation genes. Together with early bias in the dorsal to ventral transcription, this effect contributes to the change of zygotic gene expression repertoire in the double mutants.

**Changes in chromatin accessibility underlie the changes in zygotic expression repertoire in the double MZ*sox19bspg* mutant.** In order to connect the effects of Pou5f3 and Sox19b on transcription with their effects on chromatin, we mapped chromatin accessibility in all genotypes using ATAC-seq[59,60] at 3.7 and 4.3 hpf. Zygotic expression of redundant SoxB1 family members Sox3, Sox19a and Sox2 at these stages is still low (Fig. S3a), which enabled us to study the effects of maternal Sox19b on chromatin accessibility. Biological replicates for each genotype clustered together, demonstrating the high quality of our data (Fig. S6a). We selected 102 945 regions which were accessible in the wild type in both time points (ARs, Dataset S3). We split ARs to three groups, "down", "unchanged" and "up", according to the accessibility changes in the double mutant versus wild type (Fig. 5a, Fig. S6b, Fig. 5d for examples). Genes neighboring "down" and "up" ARs associated with different expression categories: "down" ARs with patterned gene expression in gastrula and early segmentation stages ("portion of tissue"), "up" ARs with genes expressed in specified organs and cells during organogenesis (Fig. 5b). We inquired if the changes of chromatin accessibility were linked to zygotic transcription. We found that down- and upregulated AR groups were enriched in the putative regulatory regions of zygotic transcripts, down- and upregulated in MZ*sox19bspg*, respectively (Fig. 5c). Remarkably, seven out of eight T-box factors transcriptionally upregulated in MZ*sox19bspg* were associated with "up" ARs and contributed to the enriched "Interpro" category "T-box" (Fig. 5b). These results indicated that Pou5f3 and Sox19b-dependent changes in chromatin accessibility underlie the changes in zygotic gene expression repertoire.

**Pou5f3 and Sox19b directly promote and indirectly repress chromatin accessibility on the gene regulatory regions with different GC content.** To distinguish between direct and indirect effects of Sox19b and Pou5f3 on chromatin accessibility, we investigated, if "down" and "up" ARs contain consensus binding motifs for Pou5f3, Sox19b, or other TFs. SoxB1 and Pou protein families can recognize *sox* or *pou* motifs, respectively, or bipartite *sox:pou* motifs, which they are thought to bind together[61]. We calculated enrichment over genomic background for *sox:pou, sox, pou*, and 12 other TF-binding motifs (Methods, Dataset S4). *Sox:pou, sox* and *pou* motifs were overrepresented on "down" ARs (black frame in Fig. 5e, Fig. S6c), almost to the level found in the regions, where the binding of both Pou5f3 and SoxB1 was documented by ChIP-seq[10] (sp peaks, Fig. 5f). This implied that Pou5f3 or Sox19b directly bind most of the "down" ARs. In contrast, in "up" ARs *sox:pou, sox*, and *pou* motifs, but not the other motifs, were underrepresented (Fig. 5e). Hence, Pou5f3 and Sox19b direct sequence-specific binding to gene regulatory regions promoted chromatin accessibility, while their repressive effects on chromatin accessibility were indirect. To corroborate this conclusion, we calculated the overlaps between ARs with the regions bound by Pou5f3 and SoxB1[10], and also with the regions bound by third zygotic activator Nanog[62]. Pou5f3 or SoxB1

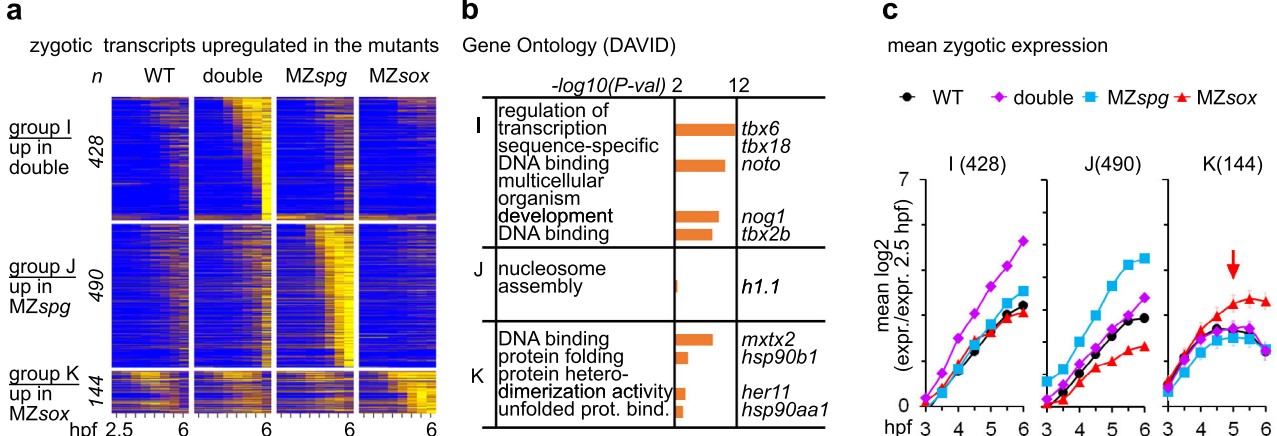

**Fig. 4 Simultaneous loss of Pou5f3 and Sox19b leads to increase in dorsal gene expression and premature activation of differentiation factors. a** Heatmap for 1062 zygotic transcripts, upregulated in the mutants compared to the wild type. The upregulated transcripts were grouped as indicated in the left (groups I, J, K), and sorted by ascending switch time and switch *p*-value in the respective mutant. **b** The groups I, J, K were tested for enrichment in Gene Ontology terms using DAVID. Top four categories per group/example genes are shown. Note that the group I upregulated in MZ*sox19bspg* is enriched for the regulators of transcription and for developmental genes. **c** Mean zygotic transcription profiles for the groups which are upregulated in the mutants compared to the wild type (I, J, K), relative to 2.5 hpf. Note the early upregulation of the transcripts over the WT in MZ*sox19bspg* (group I) and MZ*spg* (group J), but not in MZ*sox19b* (K), where the transcripts are mostly upregulated at 5 hpf (red arrow). MZ*sox* = MZ*sox19b*. *n*(wt) = 4, *n*(MZ*spg*) = 2, *n*(MZ*sox19b*) = 2, *n*(double) = 2, where *n* is a number biologically independent experiments. Data are presented as mean values ± SEM. Source data for **a**, **c** are provided as a Source Data file. Source data for **a**, **b** are provided as Dataset S2.

bound "down" ARs four times more often than "up" ARs (Fig. S6d), while Nanog bound both groups similarly (Fig. S6e, examples on Fig. 5d)

We have previously shown[16] that pluripotency factors bind to the regions with elevated nucleosome occupancy, in vitro nucleosome prediction and GC content (High Nucleosome Affinity Regions, or HNARs). We asked if these parameters differed between AR groups. In all ARs, real and predicted nucleosome occupancy and GC content were significantly higher than in genomic control (Fig. 5g, Fig. S6f–i). In addition, GC content (Fig. 5g) and in vitro predicted nucleosome occupancy (Fig. S6h, i) were lower in "down" ARs compared to "unchanged" and "up" ARs.

In sum, we found that sequence-specific binding of Pou5f3 and Sox19b to their motifs in the gene regulatory regions promoted chromatin accessibility. In the gene regulatory regions with high GC, content accessibility was indirectly repressed by Pou5f3 and Sox19b.

**Sox19b and Pou5f3 act as independent pioneer factors**. We next addressed the question of how Pou5f3 and Sox19b binding promotes chromatin accessibility in vivo: if they cooperate, act additively or independently. We analyzed ATAC-seq signals on ARs overlapping with Pou5f3 or SoxB1 ChIP-seq peaks (Fig. S7a). We also performed MNase-seq on 4.3-hpf MZ*sox19b* mutants, and used MZ*spg*, wild-type[16], and MZ*sox19b* nucleosome maps as supporting datasets. As expected from previous studies[13,16], the changes in chromatin accessibility and nucleosome occupancy were stronger on ARs bound by both factors, than by one (Fig. S7b–e). We used the regions bound by both factors (sp peaks, Fig. 6a–c) for further analysis. To address how chromatin accessibility depended on sequence-specific binding of the TFs, we made pairwise comparisons of the accessibility changes on the peaks with and without *sox:pou*, *pou* or *sox* motifs (Fig. 6d–f). Note that accessibility was reduced on *sox* motifs in MZ*sox19b*, on *pou* motifs in MZ*spg*, and on *sox:pou* motifs in both single mutants (Fig. 6e). Nucleosome occupancy on the same regions increased accordingly (Fig. S7f). Hence, we have shown by two

independent methods that Pou5f3 and Sox19b displaced nucleosomes from their cognate motifs. To address if Pou5f3 and Sox19b act additively or redundantly on these motifs, we compared the changes in the double to the single mutants (Fig. 6f). Note that chromatin accessibility on *sox:pou* and *pou* motifs in the double mutant and MZ*spg* was similarly reduced (blue frame in Fig. 6f). Hence, Pou5f3 sequence-specific binding was the only responsible for the gain of chromatin accessibility on *sox:pou* and *pou* motifs. Sox19b binding to *sox:pou* motifs at the absence of Pou5f3 either did not occur or was not sufficient to gain accessibility. Accessibility on *sox* motifs was further reduced in the double mutant compared to MZ*sox19b* (red frame in Fig. 6f), suggesting additive or redundant effects of Pou5f3 on *sox* motifs. As shown in Fig. S7g–i, these effects were due to the presence of *sox:pou* or *pou* in addition to *sox* motifs in the same regions.

In sum, we concluded that Sox19b and Pou5f3 are both involved in establishment of chromatin accessibility and act as independent pioneer factors. Sox19b promoted accessibility on *sox* motifs. Sox19b binding to *sox:pou* motifs was not sufficient to gain accessibility, but it could i.e. stabilize Pou5f3 binding to some of them. Pou5f3 promoted accessibility on *pou* and *sox:pou* motifs. Two factors could act additively or redundantly if their cognate motifs were present within one accessible region.

**Pou5f3-dependent, codependent and redundant direct cis-regulatory elements activate early zygotic genes**. Once we established that Pou5f3 and Sox19b act independently, we asked, how many of the putative direct cis-regulatory elements ("down" ARs, Fig. 5a) are regulated by binding of each factor. We split 20653 "down" ARs into four groups considering reduction of chromatin accessibility in the single mutants (Fig. 7a, b). The groups were: (1) Codependent elements, where both TFs were necessary for chromatin accessibility (9.5% of all "down" elements), (2) Pou5f3-dependent elements, where only Pou5f3 was necessary (56%), (3) Sox19b-dependent elements, where only Sox19b was necessary (6%), (4) redundant elements, where either Pou5f3 or Sox19b was necessary (28,5%). To approximate the sequence-specific binding of each factor in each group, we

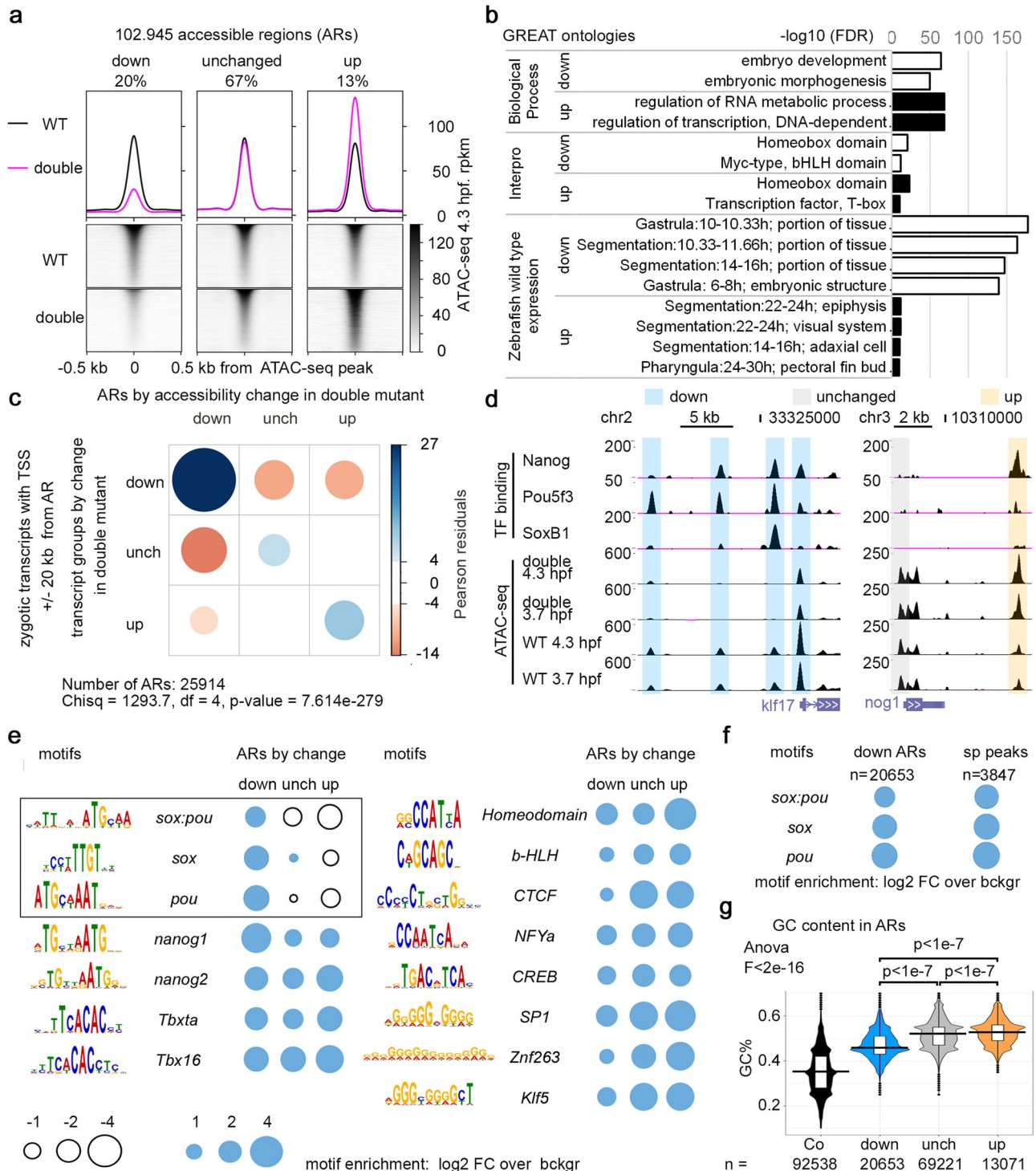

compared their enrichment for *sox:pou*, *sox* and *pou* motifs to sp peaks (bound by both factors). We assumed that most regions of group 1 and 2 were bound by both factors, group 3 only by Sox19b, group 4 by Sox19b and to less extent by Pou5f3 (Fig. 7c). We next inquired if the changes of accessibility on the groups 1–4 were linked to the changes in zygotic transcription in the double mutant. Note that the AR groups 1, 2, and 4 but not group 3 were enriched in the putative regulatory regions of zygotic transcripts, downregulated in the double mutant (Fig. 7d).

To understand why some zygotic transcripts changed in the single mutants only (transcript groups B, C, E in Fig. 3b), we also analyzed the ARs on which chromatin accessibility was changed

in the single but remained unchanged in the double mutants ("compensated" ARs, Fig. S8a). We found that most of the compensatory changes were indirect (*i.e.* "compensated" ARs were not specifically bound by both TFs, Fig. S8b–e). Compensated ARs were significantly associated with the zygotic transcriptional changes in MZ*spg* only (Fig. S8f), but not in MZ*sox19b* only (Fig. S8g). Thus, general transcriptional delay in MZ*sox19b* was not related to the changes in chromatin accessibility. In contrast, indirect antagonistic effects of Sox19b and Pou5f3 on the chromatin accessibility could explain why some transcripts (groups B and E, Fig. 3b) were changed only in MZ*spg*, but not in double mutants.

**Fig. 5 Changes in chromatin accessibility on the cis-regulatory regions underlie expression changes in the double mutants. a** Three groups of accessible regions (ARs), where chromatin accessibility was reduced ("down"), unchanged, or increased ("up") in MZsox19bspg double mutants relatively to the wild type. Regions were sorted by descending ATAC-seq peak score. **b** GREAT[91] enrichment in Gene Ontology terms for "down" and "up" regions shown in a. Top two categories are shown in "Biological Process" and "Interpro", four top categories in "Zebrafish wild-type expression". **c** "down", "unchanged" and "up" ARs are enriched around Transcription Start Sites (TSS) of zygotic genes, down, unchanged, and upregulated in MZsox19bspg double mutant, respectively. Chi-square test, p-value is two-tailed. **d** Pou5f3, SoxB1, and Nanog binding on the cis-regulatory regions of klf17 and nog1 genes, down- or upregulated in MZsox19bspg respectively. UCSC genomic browser view, TF binding and ATAC-seq signal—normalized reads. Note that all three TFs bind to the "down" cis-regulatory regions of klf17, but only Nanog to the "up" cis-regulatory region of nog1. **e** Enrichment for TF-binding motifs on "down", "unchanged" and "up" ARs, relative to genomic background. Cognate motifs for Pou5f3 and Sox19b are enriched in "down" and underrepresented in "up" group (black frame). **f** Enrichment for sox:pou, sox and pou motifs on all "down" regions (ARs down) is close to that on the regions directly bound by Pou5f3 and SoxB1 (sp peaks). **g** GC nucleotide content of "down" ARs is higher than control genomic regions but lower than in unchanged and "up" ARs. To obtain control regions, genomic coordinates of ARs were shifted 1 kb downstream. Numbers of ARs (n) are indicated below the graphs. The lower border, middle line, and upper border of the white box plots correspond to 0.25, 0.5 (median), and 0.75 quantile, respectively. 1-way ANOVA, p-values in Tukey–Kramer test. Source data for **a**, **b**, and **d**–**g** are provided as a Dataset S3. Source data for **e**, **f** are provided as Dataset S4. Source data for **c** are provided as Dataset S5.

In sum, we delineated four direct types of cis-regulatory elements. Three of them, Pou5f3-dependent, codependent and redundant elements, were involved in the activation of early zygotic genes.

**Pou5f3 and Sox19b regulate enhancer activity**. In parallel with the increase of chromatin accessibility during ZGA, zebrafish genome acquires the H3K27ac histone mark on active enhancers of the earliest expressed genes[6,63]. To test if Pou5f3 and Sox19b regulate enhancer activity, we profiled H3K27ac by ChIP-seq in the wild type, MZspg and MZsox19b 4.3 hpf embryos (Fig. S9a, b) and compared H3K27ac changes around differentially regulated ARs in the single mutants and wild type. We found that H3K27ac changed in parallel to chromatin accessibility (Fig. S9c, d). Note that on the four types of direct cis-regulatory elements, the same transcription factor which was essential for chromatin accessibility was also essential for enhancer activation (Fig. 7e, compare with Fig. 7b). Interestingly, on the Sox19b-dependent group of ARs, H3K27ac was strongly increased in MZspg (Fig. 7e, group 3). This indicated that Sox19b-dependent enhancers were indirectly silenced by Pou5f3.

To estimate the contribution of enhancers and promoters to our list of putative cis-regulatory elements (ARs), we used H3K27ac, genome annotations, and H3K4me3 promoter mark[64] which we also profiled by ChIP-seq in the single mutants and the wild type (Methods, Dataset S3). According to our estimations, 33% of all ARs, 16% of promoters, and 48% of enhancers were regulated by Pou5f3 or Sox19b, directly or not (Fig. 8a). Figure 8b shows the examples of five regulated enhancer types. GC content of enhancers directly activated by Pou5f3 and/or Sox19b was lower than of the others (Fig. S9e). Different types of enhancers were associated with distinct gene expression ontologies: Sox19b-dependent and indirectly repressed enhancers associated with genes expressed after gastrulation, in the nervous system, and in the derivatives of dorso-lateral mesoderm, respectively (Fig. S9f).

## Discussion
Gene expression starts as a result of a multi-step regulatory program involving a rapid succession of changes in chromatin accessibility, histone modifications, and transcription[65]. In this study, we dissected the contributions of Pou5f3 and Sox19b to chromatin accessibility, transcription, and to the developmental processes shaping the embryo during gastrulation. This yielded three insights.

First, on the mechanistic level, we found that Sox19b and Pou5f3 are both involved in establishment of chromatin accessibility at blastula stages and act mostly independently. Sox19b promotes accessibility on sox motifs, Pou5f3 on pou and sox:pou

motifs (Fig. 6d–f). Sox19b and Pou5f3 can act additively or redundantly if their cognate motifs are present nearby (Fig. S7g–i). Out of four different scenarios for the interactions of the mammalian homologs in ES cells, which were outlined in the introduction[23–26], our results agree with the conclusions of the last study: POU5F1 and SOX2 operate in a largely independent manner even at co-occupied sites[26].

Second, our results strongly suggest that the establishment of transcriptional competency at the onset of major ZGA is regional. ZGA is mediated by Pou5f3 and Sox19b at the ventral side of the embryo, and independent of Pou5f3 and Sox19b at the dorsal side (summarized at Fig. 8c). This view is supported by several observations. (1) Pou5f3 and Sox19b promote zygotic expression of the regulators of ventral development (zygotic genes down-regulated in MZsox19bspg, Fig.3, group A). Pou5f3-dependent, codependent and redundant cis-regulatory elements are enriched around these genes (Fig. 7d). (2) Pou5f3 and Sox19b are dispensable for the zygotic activation of the targets of Eomesodermin/ Nodal and maternal β-catenin pathways which are first expressed on the dorsal side of the embryo (Fig. 3, group B). (3) In the MZsox19bspg mutant, the zygotic transcription ratio of "ventral" group A to "dorsal" group B is reversed (Fig. 3d,e, Fig. S5e), resulting in the dorsalized phenotype (Fig. 2). Dorsal and mesendodermal genes are probably primed for activation by maternal transcription factors Eomesodermin, ß-catenin or Nanog[66–68]. The idea of independent zygotic activation of mesendodermal genes is compatible with recent studies in mice and Xenopus. In Xenopus, endodermal enhancers are separately primed for zygotic activity by three endodermal maternal TFs[69]. In mice, Eomesodermin and Brachyury are the sole factors responsible for establishing the competence for activation of mesodermal enhancers, and they repress the pluripotency and neuroectodermal programs driven by POU5F1 and SOX2[70].

Third, we show that Pou5f3 and Sox19b control the order of transcriptional onset for zygotic genes not only by activating early genes but also by indirectly shutting down the genes which elicit later developmental programs. Early studies of Alexander Neyfakh described two distinct periods of zygotic gene function in teleost fish[71,72]. The genes expressed from mid to late blastula (first period) provided the instructions for gastrulation. The genes expressed starting from mid gastrula (second period) provided the instructions for organogenesis. Our results suggest that maternal Pou5f3 and Sox19b ensure the proper time gap between the first and second period in two cases. First, Pou5f3 suppressed activation of Sox19b-dependent enhancers (Fig. 7e, group 3), which were associated with genes expressed in the nervous system during organogenesis (Fig. S9f), but were not associated with early zygotic genes (Fig. 7d, group 3). Second, Pou5f3 and Sox19b

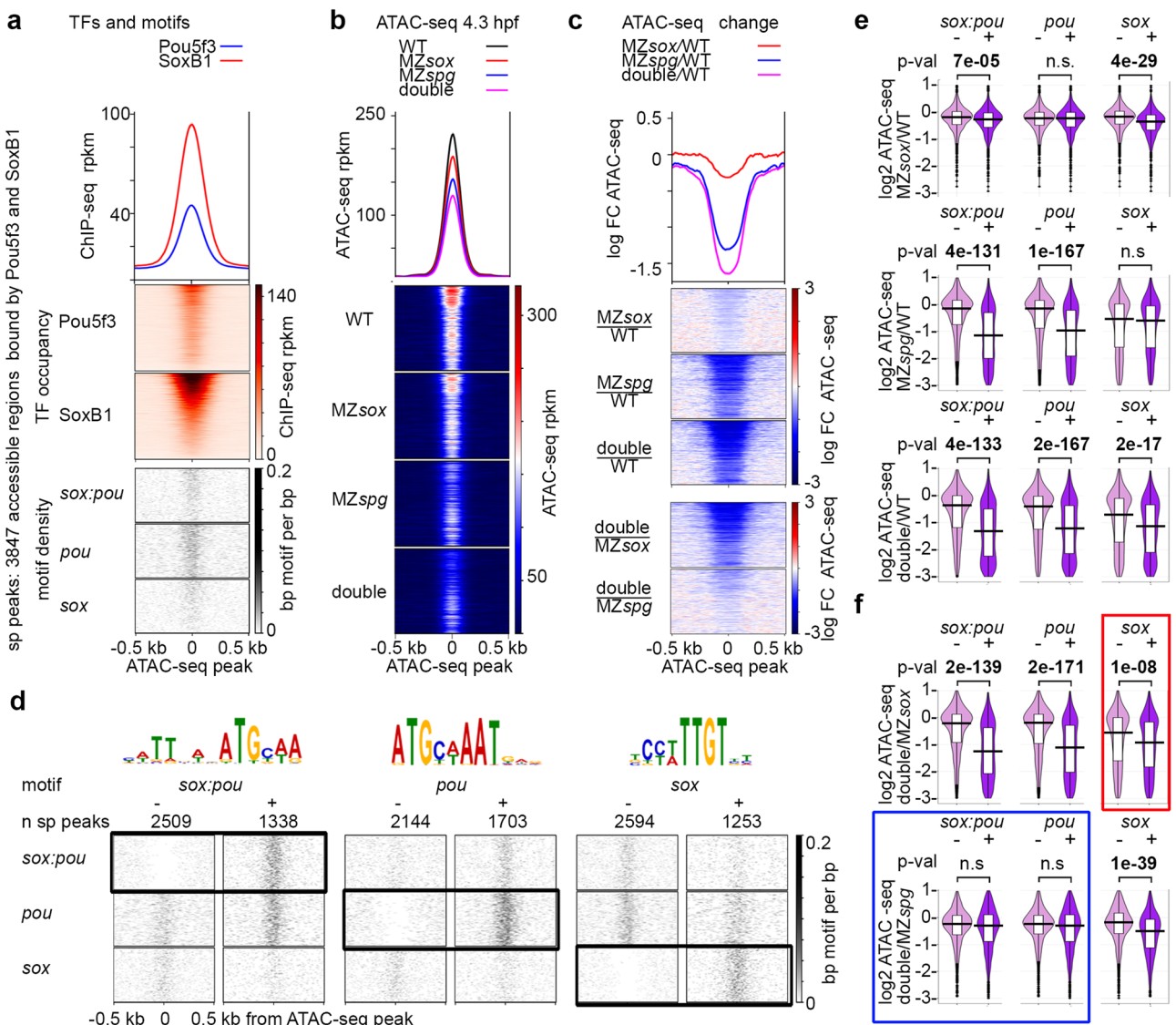

**Fig. 6 Pou5f3 and Sox19b regulate chromatin accessibility on different motifs. a–c** 3847 accessible regions bound by both Pou5f3 and SoxB1 (sp peaks) were used for analysis. For the heatmaps, the regions were sorted by descending SoxB1 Chip-seq signal. **a** Heatmaps show TF occupancy and motif density on sp peaks. **b** Chromatin accessibility on sp peaks is reduced in the single and double mutants at 4.3 hpf. **c** Chromatin accessibility change. Three top heatmaps: mutant to wild-type change. Two bottom heatmaps: double to single mutant change. Note the strongest chromatin accessibility reduction in the double mutant and the weakest in MZ*sox19b*. **d** Chromatin accessibility changes at 4.3 hpf were compared between sp peaks without (−) or with (+) motifs indicated above (**e**, **f**). Scale-motif density. **e** Mutants to wild-type change. Sox19b binding to *sox* motifs and Pou5f3 binding to *pou* motifs promotes chromatin accessibility. On *sox:pou* motifs, binding of any factor has a significant effect, but Sox19b effects are weaker. **f** Double to single mutant change. Blue frame: on *sox:pou* and *pou* motifs, chromatin accessibility is similarly reduced in the double MZ*sox19bspg* mutant compared to MZ*spg*. Hence, the gain of chromatin accessibility on *sox:pou* and *pou* motifs results from Pou5f3 binding. On some of *sox:pou* motifs Sox19b may act by facilitating or stabilizing Pou5f3 binding. Red frame: on *sox* motifs, chromatin accessibility is stronger reduced in MZ*sox19bspg* compared to MZ*sox19b*, suggesting additive or redundant effects of Pou5f3. As shown at Fig. S7g, this effect is due to the sp peaks where both types of motifs (*sox* and *sox:pou* or *sox* and *pou*) are present. Hence, Pou5f3 and Sox19b may promote chromatin accessibility additively or redundantly when they bind to different motifs within the same AR. MZ*sox* = MZ*sox19b*. At **e**, **f**: The lower border, middle line and upper border of the white box plots correspond to 0.25, 0.5 (median), and 0.75 quantile, respectively. n(wt) = 3, n(MZ*spg*) = 3, n(MZ*sox19b*) = 3, n(double) = 4, where n is a number biologically independent ATAC-seq experiments. P-values in two-tailed Student's t-test. Source data for **b**, **c**, **e**, and **f** are provided as a Dataset S3. Source data for **a**, **d** are provided as Dataset S4.

indirectly repressed GC-rich regulatory regions of transcription factors involved in differentiation and patterning of tissues after gastrulation (Fig. 5, "up" group, Fig. 8a, b, "repressed" enhancers), and premature expression of these factors (Fig. 4, group I, summarized in Fig. 8d).

One known conserved regulatory mechanism that keeps the late developmental genes from premature expression is DNA methylation of their enhancers. Massive demethylation of enhancers occurs between the end of gastrulation and the second

day of zebrafish development[73]. As far as we can judge, indirect repressive effects of Pou5f3 and Sox19b were not related to methylation: we found no correlation between the list of demethylated regions[73] and differentially accessible regions from our study. Balancing of early and late developmental programs by Pou5f3 and Sox19b directly parallels reprogramming in mouse fibroblasts, where Pou5f1 and Sox2 shut down the somatic gene expression programs and activate early embryonic genes[74]. The mechanism by which this is achieved is currently unclear.

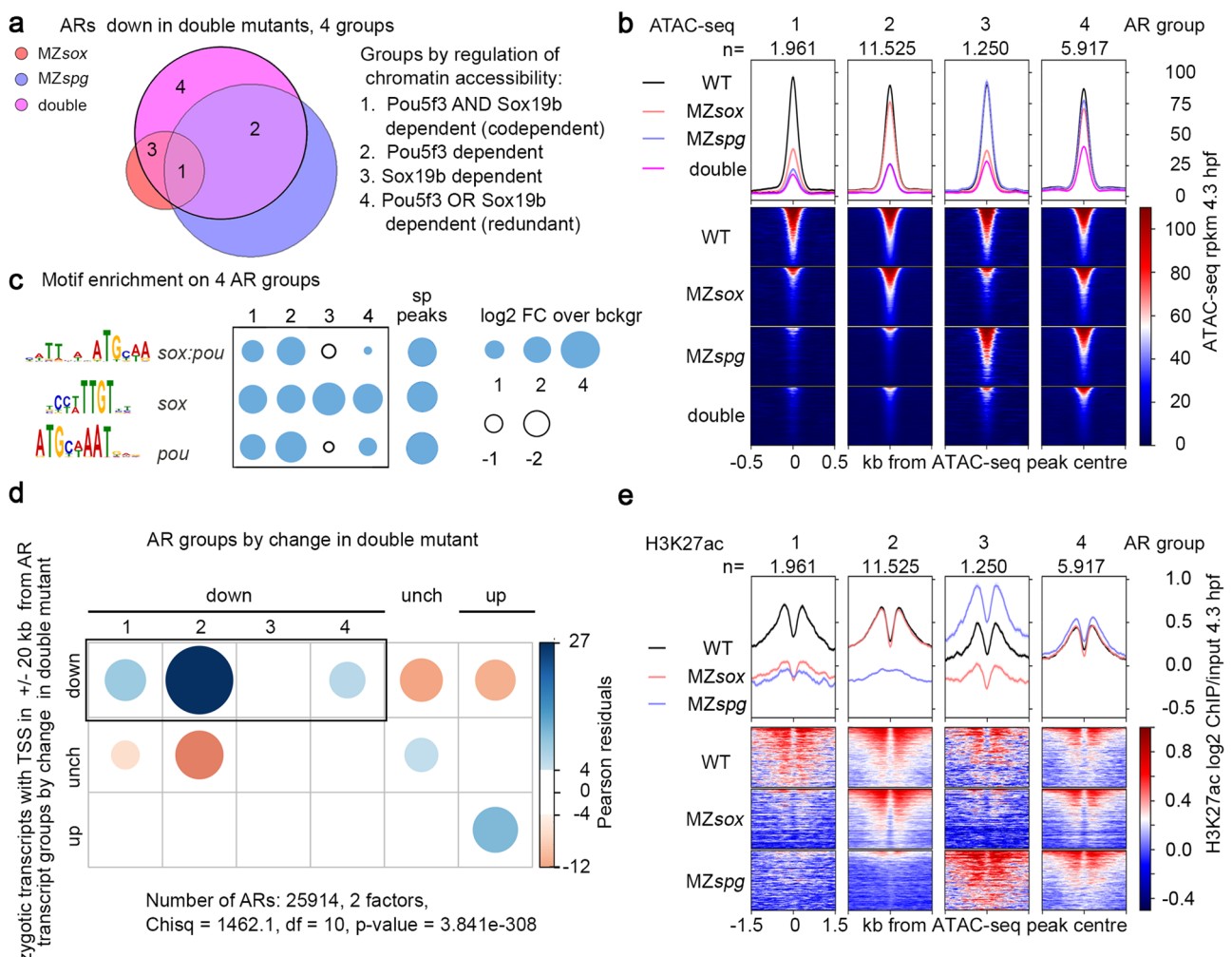

**Fig. 7 Codependent, Pou5f3-dependent, and redundant cis-regulatory elements regulate the major ZGA wave. a** Venn diagram: 20653 "down" ARs (regions where chromatin accessibility was reduced in the double mutant) were subdivided to groups 1–4 by reduction of chromatin accessibility in the single mutants. **b** ATAC-seq signals on AR groups 1–4 in the wild type, single and double mutants. ARs were sorted by descending ATAC-seq peak score. **c** Enrichment for *sox:pou*, *sox* and *pou* motifs in AR groups 1–4 compared to the regions bound by Pou5f3 and SoxB1 (sp peaks). **d** Chi-squared test. Codependent(1), Pou5f3-dependent (2), and redundant (4) ARs, but not Sox19b-dependent ARs, are enriched around Transcription Start Sites (TSS) of zygotic genes, downregulated in MZ*sox19bspg* double mutant. *P*-value is two-tailed. **e** H3K27ac on 4 AR groups depends on the same TFs as accessibility (compare with ATAC-seq in **b**). (1) codependent ARs: Pou5f3 and Sox19b are required for H3K27ac, (2) Pou5f3-dependent ARs: Pou5f3 is required for H3K27ac, (3) Sox19b-dependent ARs: Sox19b is required for H3K27ac, (4) redundant ARs: any factor is sufficient for H3K27ac, no reduction in the single mutants. ARs were sorted by descending ATAC-seq peak score. MZ*sox* = MZ*sox19b*, double = MZ*sox19bspg*, *n* number of regions in each group. Source data for **a–c**, and **e** are provided as a Dataset S3. Source data for **c** are provided as Dataset S4. Source data for **d** are provided as Dataset S5.

Answering this question will require the identification of TFs which activate transcription of the late genes in the absence of Pou5f3 and Sox19b. As the activation may be region-specific, these studies will require a combined single-cell analysis of transcription and enhancer activity.

## Methods

**Experimental model and subject details.** Wild-type fish of AB/TL and mutant sox19b[m1434] strains were raised, maintained, and crossed under standard conditions as described by Westerfield[75]. The progeny of MZ*spg*[m793] and MZ*sox19b*[m1434];*spg*[m793] homozygous lines was rescued to viability by microinjection of Pou5f3 mRNA into 1-cell stage embryos as described previously[27]. Embryos were obtained by natural crossing (4 males and 4 females in 1,7 l breeding tanks, Techniplast) or by in vitro fertilization. In vitro fertilization was used to synchronize the embryos for scoring the duration of pre-ZGA cell cycles (experiment in Fig. S1), natural crosses were used for all other experiments. Wild type and mutant embryos from natural crosses were collected in parallel in 10–15 min intervals and raised in egg water at 28.5 °C until the desired stage. Staging was performed following the Kimmel staging series[31]. Stages of the mutant embryos were indirectly determined by observation of wild-type embryos born at the same

time and incubated under identical conditions. All experiments were performed in accordance with German Animal Protection Law (TierSchG) and European Convention on the Protection of Vertebrate Animals Used for Experimental and Other Scientific Purposes (Strasburg, 1986). The generation of double mutants was approved by the Ethics Committee for Animal Research of the Koltzov Institute of Developmental Biology RAS, protocol 26 from 14 February 2019.

**Genomic DNA isolation and PCR for genotyping.** Genomic DNA was isolated from individual tail fin biopsies of 3 months old fish or from the pools of 24 h post-fertilization (hpf) embryos. Tail fin biopsies or embryos were lysed in 50 μl lysis buffer (10 mM Tris pH 8, 50 mM KCl, 0.3% Tween 20, 0.3% NP-40, 1 mM EDTA) and incubated at 98 °C for 10 min. After cooling down Proteinase K solution (20 mg/ml, A3830, AppliChem) was added and incubated overnight at 55 °C. The Proteinase K was destroyed by heating up to 98 °C for 10 min. The tail fin biopsies material was diluted 20x with sterile water. 2 μl of was used as a template for PCR. PCR was performed in 25–50 μl volume, using MyTag polymerase (Bioline GmbH, Germany) according to the manufacturer's instructions, with 30–35 amplification cycles. The primer sequences are listed in the subsequent sections and in the Table S2.

**Generation and maintenance of MZ*sox19b*[m1434] mutant line.** To generate *sox19b* zebrafish mutants, we used the TALEN technique[30] and targeted the first

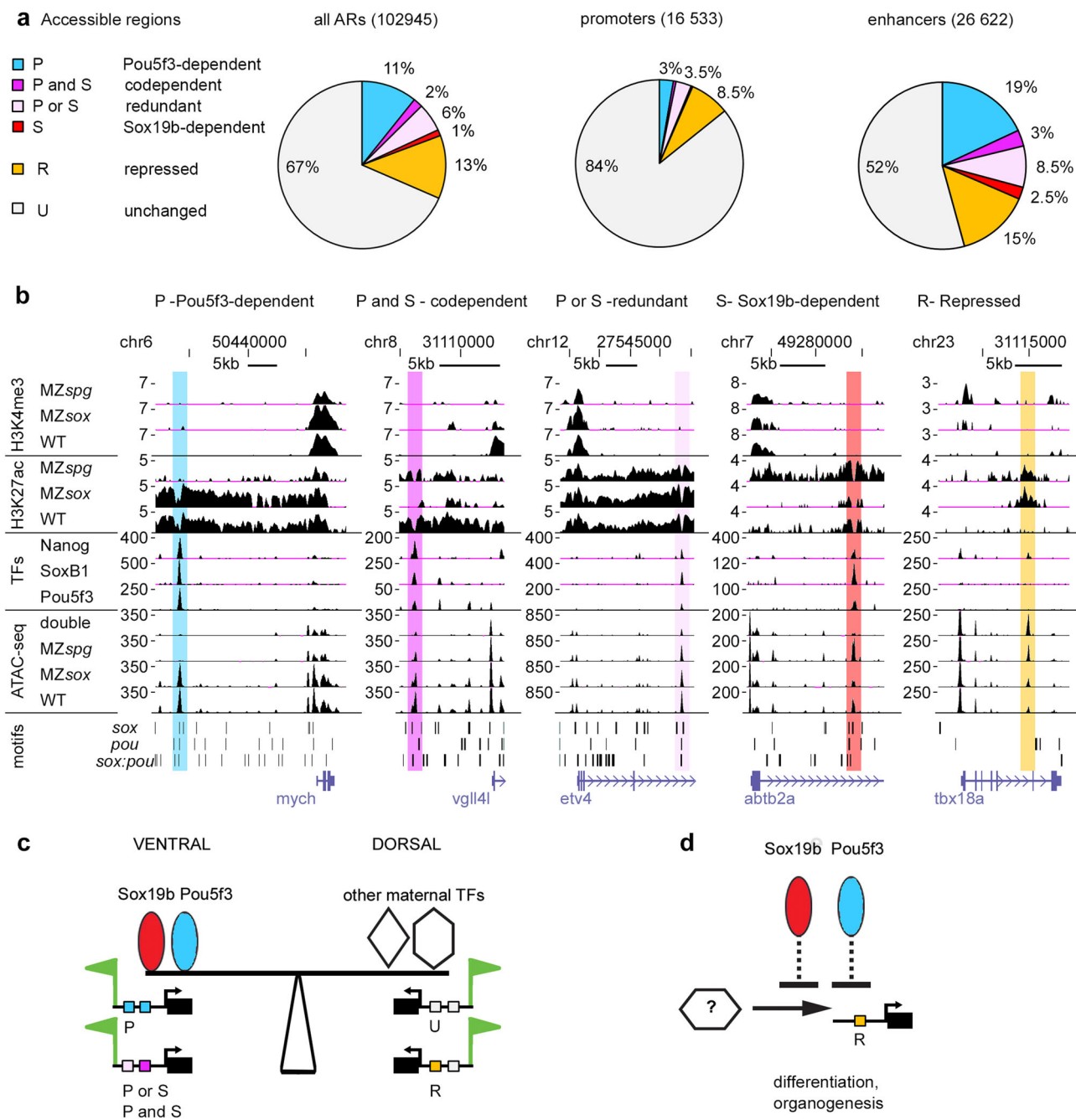

**Fig. 8 Pou5f3 and Sox19b regulate enhancer activity. a** Pou5f3 and Sox19b regulate chromatin accessibility on more enhancers than promoters. Six types of cis-regulatory elements include four types downregulated in the double mutants (Pou5f3-dependent, codependent, redundant, and Sox19b-dependent), one type upregulated in the double mutant (repressed), and unchanged in the double mutant. Percentages of these types for all ARs, promoters, and enhancers are shown **b** Examples of enhancer types where chromatin accessibility is directly regulated by Pou5f3 or Sox19b, as indicated (*mych, vgll4l, etv4, abtb2a*) and indirectly repressed enhancer (*tbx18a*). UCSC genomic browser view, H3K27ac and H3K4me3 are shown as log2 ChIP/input ratio, ATAC-seq and TF binding as normalized reads, motif occurrence as bars. Note the parallel changes in H3K27ac and chromatin accessibility (for MZ*spg* and MZ*sox19b*). MZ*sox* = MZ*sox19b*, double = MZ*sox19bspg*. Source data for **a**, **b** are provided as a Dataset S3. **c** The model of dorso-ventral balance at ZGA: Pou5f3 and Sox19b prime ventral and ectodermal genes for activation, activating Pou5f3-dependent, codependent and redundant enhancers ("P", "P and S", "P or S"). Dorsal and mesendodermal genes are primed by other maternal TFs via independent enhancers (I) and enhancers indirectly repressed by Pou5f3 and Sox19b ("R", i.e. *noggin1* enhancer on Fig.5d). **d** Scheme: Pou5f3 and Sox19b indirectly repress premature activation of transcription factors involved in organogenesis and differentiation.

exon for the mutation. The square brackets indicate the spacer sequence, flanking sequences should bind to TAL1 and TAL2: 5′-GATGGAGCACGAGCT[GAAGAC CGCTGGTCCA]CCCCACACCCTCCAGC-3′. For restriction digest the enzyme BbsI was selected with the corresponding restriction site 5′-GAAGAC-3′. After injecting the TALENs (100 ng/μl each) into 1-cell stage wild-type embryos we tested the proper activity of TALENs. We extracted genomic DNA from 20 of

24 hpf old wild-type injected embryos and 20 non-injected embryos for the control, and used it as a template for PCR. We used the following primers for PCR: Sox19b-f1 5′-ATTTGGGGTGCTTTCTTCAGC-3′ and Sox19b-r1 5′-GTTCTCCTGGGCC ATCTTCC-3′. This gives a product of 362 bp length, which contains two restriction sites for BbsI. 5 μl of the PCR mix was digested overnight with 5 units of BbsI (New England Biolabs) in 30 μl volume. The digestion of the wild-type PCR

product resulted in three bands with sizes of 40 bp, 132 bp, and 190 bp. In the successfully mutated embryos one of the BbsI sites was destroyed, and additional 322 bp band appeared. After successful TALEN injection, we let the fish grow and found through an outcross with wild type two founders out of 19 tested fish. We chose the founder with an 8 bp deletion resulting in a frameshift and a stop codon after 62 amino acids (5′-GATGGAGCACGAGCTGAAGA | CCACCCCACAC CCTCCAGC-3′, the line shows the position where the deletion occurred). We incrossed the heterozygous progeny of this founder (sox19b[m1434]) and selected homozygous fish by PCR-BbsI restriction digest (322 bp band, but no 190 bp and 132 bp bands should be present). To obtain MZsox19b[m1434] line, we incrossed the homozygous fish. MZsox19b[m1434] line has been maintained by incrossing over 6 generations. To confirm that all the fish carry sox19b[m1434], allele, we genotyped tail fin biopsies of 3 months old fish in each generation. In all generations, the fish were viable, fertile, and phenotypically normal, except that the embryos were somewhat smaller than the wild type and exhibited the gastrulation delay. To obtain Maternal mutant progeny (Msox19b), MZsox19b females were crossed to the wild type (AB/TL) males.

**Generation of MZsox19bspg double mutants.** We obtained MZsox19bspg mutants in three subsequent crossings. First, MZspg homozygous males were outcrossed with MZsox19b homozygous females. The double heterozygous progeny developed into phenotypically normal and fertile adults. The heterozygous fish were incrossed to obtain the double mutants. To bypass the early requirement for Pou5f3 in the spg793 homozygous mutants, one-cell stage embryos were micro-injected with synthetic Pou5f3 mRNA. The fish were raised to sexual maturity (3 months), genomic DNA from tail fin biopsies was isolated and used for geno-typing. We first selected sox19b homozygous mutants, by PCR-with Sox19b-f1/Sox19br1 primers followed by restriction digest with BbsI, as described in the previous chapter. To select the double homozygous fish, we used the genomic DNA from sox19b homozygous mutants to PCR-amplify the region, flanking the spg[m793] allele. Spg[m793] allele carries an A- > G point mutation in the splice acceptor site of the first intron of Pou5f3 gene, which results in the frameshift starting at the beginning of the second exon, prior to the DNA-binding domain. Spg[m793] is considered to be a null allele[76]. We used the following PCR primers: spg-f1 5′-' GTCGTCTGACTGAACATTTTGC -3′ and spg-r1 5′-' GCAGTGATTCTGAGG AAGAGGT -3′. Sanger sequencing of the PCR products was performed using commercial service (Sigma). The sequencing traces were examined and the fish carrying A to G mutation were selected. To obtain the Maternal-Zygotic (MZ) homozygous mutants, Zsox19bspg were incrossed. MZsox19bspg fish have been maintained by incrossing over three generations. Each generation of MZsox19bspg was rescued to viability by microinjection of Pou5f3 mRNA into 1-cell stage embryos. Fish in each generation were genotyped to confirm that they carry sox19b homozygous allele. To obtain Maternal (M) mutants, MZsox19bspg females were crossed to the AB/TL males.

**Preparation of synthetic sox19b mRNA.** Sox19b open reading frame was amplified from zebrafish total cDNA (4.3 hpf), using the PCR primers according to sox19b mRNA sequence in UCSC: sox19Bf1 with BamHI: 5′-GGGGGATCCA TGGAGCACGAGCTGAAGAC-3′, Sox19bR1 with Xho1: 5′- GGGGCTCGAGT CAGATGTGAGTGAGGGGAAC-3′. The PCR product was cloned in pCRII-TOPO vector using TOPO TA Cloning® Kit (Invitrogen), and further sub-cloned into PCS2 + vector via BamH1/Xho sites. mRNA was in vitro transcribed with mMESSAGE mMACHINE® SP6 Kit (Ambion) according to the user manual. Sox19b mRNA was cleaned up with QIAGEN RNeasy® Mini Kit.

**SoxB1 morpholino knockdown.** Two translation-blocking morpholinos per each SoxB1 gene (Sox2, Sox3, Sox19a, Sox19b (see the Table S2 for the sequences)) were designed and validated by Okuda et al. 2010[28] and provided by Gene Tools. LLC (Philomath, USA). To reproduce SoxB1 quadruple knockdown phenotype (QKD), 1-cell stage wild-type embryos were microinjected with the mix of eight morpholinos, (0.9 ng each morpholino, 7.2 ng per embryo total, as in[28]). Standard Morpholino control (Gene Tools) injection at 7.2 ng per embryo was used as a control. To generate the triple knockdown of zygotic SoxB1 genes in MZsox19b mutants, 1-cell stage MZsox19b embryos were microinjected with Sox2, Sox3, and Sox19a morpholino mix. Microinjection of 0.9 ng per morpholino (5.4 ng in total) resulted in the MZsox19b-TKD phenotypes similar to WT QKD in two experi-ments: the tail bud and head were not properly formed and anterior–posterior axis was severely truncated (Fig. 2b).

To investigate the dose-dependent effects of the morpholinos, we injected different amounts of TKD mix and TKD control mix (TKDco) into MZsox19b embryos. In TKDco mix, Sox2 and Sox3 morpholino pairs were present as in TKD, while Sox19a was replaced with Sox19b morpholino pair, so that only three out of four SoxB1 genes would be blocked upon injection. As it was previously shown that triple Sox2/3/19b knockdown causes mild defects detectable only after 31 hpf[28], the main purpose of using TKDco mix was to better control for non-specific effects of TKD mix. Non-specific developmental defects, especially cell death in the nervous system, are common caveats of the morpholino use in zebrafish, which depend on the morpholino concentration and composition[77]. TKD injection of 0.9 ng each morpholino (5.4 ng total) per MZsox19b embryo, or 50% of this

amount (0.45 ng each) resulted in the range of three phenotypic classes (II–IV, Fig. S3d, e) in two experiments. The "middle" phenotype (class III) was similar to previously published QKD phenotype resulting from either 0.45 and 0.9 ng morpholino injections into the WT[28]. The TKDco mix injection into MZsox19b embryos resulted in the embryos, which were often somewhat shorter than non-injected (class I and normal in Fig. S3d, e); non-specific axial defects were observed in <5% of MZsox19b-TKDco injections (0.9 ng, Fig. S3f). Since TKD injection into MZsox19b caused different and much more severe phenotypes than TKDco, we concluded that the MZsox19b-TKD phenotype specifically results from the reduction of zygotic SoxB1 activity.

**MZsox19b-TKD rescue experiments.** For the rescue experiments, 5 or 20 pg of sox19b mRNA or 20 pg GFP mRNA was co-injected into 1-cell stage MZsox19b embryos together with 2.7 ng TKD Morpholinos. For the control, MZsox19b embryos were injected with 2.7 ng TKDco mix and 20 pg control GFP mRNA. The embryos of all experimental groups were let to develop until 19 hpf at 28,5 °C. At 19 hpf, the phenotypes were scored, representative pictures of living embryos were taken, and the whole experiment was fixed in 4% paraformaldehyde (PFA) in PBS. The rescue extent was quantified by measuring the body axis length according to Okuda et al. 2010[28]. Namely, we took lateral images of all fixed embryos, drew the midline from head to tail for each embryo (Fig. S3g), and measured it using ImageJ 1.53a. The representative phenotypes of one of two experiments (rep1) are shown in Fig. 2c, body length statistics for two independent experiments are shown in Fig. S3g.

**Noggin and Chordin morpholino knockdown.** Noggin and Chordin morpholinos were designed by Dal-Pra et al. (2006)[36]. To block Noggin translation, 1 nM Noggin MO was injected. To moderately reduce Chordin translation, 100 pM Chordin morpholinos were injected, as recommended by Dal-Pra et al. (2006)[36]. All morpholinos were provided by Gene Tools. LLC (Philomath, USA), sequences are provided in the Table S2.

**In vitro fertilization.** Adult male fish were anesthetized with Tricain (4% 3-Aminobenzoic acid ethyl ester, pH 6.7) and then positioned with anal area above. The sperm was taken using a capillary, mixed with 5 μl E400 buffer (9.7 g KCl, 2.92 g NaCl, 0.29 g CaCl$_2$ −2H$_2$O, 0.25 g MgSO$_4$ −7H$_2$O, 1.8 g D-(+)-Glucose, 7.15 g HEPES in 1 L dH$_2$O, pH 7.9) and then with 150 μl SS300 buffer (0.37 g KCl, 8.2 g NaCl, 0.15 g CaCl$_2$ −2H$_2$O, 0.25 g MgSO$_4$ −7H$_2$O, 1.8 g D-(+)-Glucose 20 ml 1 M Tris-Cl, pH 8.0, in 1 L dH$_2$O). The collected sperm was kept at room tem-perature. Adult females were anesthetized, placed into 35 mm petri dish and to squeeze the eggs. For fertilization, we activated the sperm by adding 200 μl sterile water and transferred it onto the freshly collected eggs. After 2 min, the dish was filled with sterile water and transferred to 28,5 °C incubator.

**Quantification of the pre-MBT cell cycle duration in the wild-type and MZsox19b mutant embryos.** To obtain synchronously developing wild-type and MZsox19b embryos, in vitro fertilization of approx. 200–400 eggs per genotype was performed by two people in parallel. The fertilized embryos were scored and left to develop at 28.5 °C. 30~35 embryos from each genotype were simultaneously fixed in 4% PFA every 15 min starting from 1.75 hpf (5th cell cycle, 32 cell stage) till 3 hpf (10th cell cycle, 1 K stage). The embryos were incubated at 4 °C overnight, washed 3x in PBST (PBS with 0,1% Tween 20) and manually dechorionated. The embryos were then permeabilized in PBSDT (PBS with 0,1% Tween 20, 1% DMSO, and 0.3% TritonX-100) for 48 h at 4 °C, stained in 0.025 μM SYTOX Green (ThermoFisher SCIENTIFIC) for 1.5 h and washed 3x in PBST. The pictures were taken using LEICA MZ16F stereomicroscope equipped with epifluorescence. The pictures were contrasted in ImageJ 1.53c, and used to estimate the stage and the cell cycle phase (Fig. S1a). The embryos where the most nuclei were mitotic (metaphase to anaphase) were scored as intermediate between the cycles (i.e. cycle 5–6, 6–7 etc.,Fig. S1a). We did not detect the differences in the pre-MBT cell cycle lengths in two experiments. The quantification of representative experiment out of two is shown in the Fig. S1b.

**Live imaging.** Zebrafish embryos were imaged using Leica MZ APO stereo-microscope with Axiocam 305 camera and AxioVision SE64 Rel. 4.9.1 or Zen2 software (Carl Zeiss).

**Parallel time-lapse recording of developmental rates in the wild type and mutants.** The wild-type and MZ or M mutant embryos were obtained by natural crossings in the mass-crossing cages (4 males and 4 females of each genotype). The freshly laid eggs were collected in 10–15 min intervals. At 4–8 cell stage, one embryo per genotype was manually dechorionated and placed on a 1.5% agarose chamber filled with 0.3x Danieau's buffer. Images were taken with 3 min intervals for 24 h by either Leica MZ APO stereomicroscope, Axiocam 305 camera, and AxioVision SE64 Rel. 4.9.1 software (Carl Zeiss). The video recording started between 1.5 and 2 hpf. Time after fertilization (hh . mm) was added in ImageJ 1.50i. Time-lapse recordings from five independent experiments are included in the

Movie S1, (Part1-Part5), which show parallel development of the embryos with the following genotypes:

Part1: WT, MZsox19b, Msox19b
Part2: WT, MZsox19b, MZspg, MZsox19bspg
Part3: WT, MZspg, MZsox19bspg
Part4: WT, Mspg, Msox19bspg
Part5: WT, Msox19b, Msox19bspg

**Whole mount in situ hybridization (WISH).** To visualize the expression pattern of some chosen genes we performed whole mount in situ hybridization as previously described[67]. The plasmids for anti-sense RNA probe synthesis were kind gifts of Wolfgang Driever, Matthias Hammerschmidt, Yusuke Kamachi and Liliana Solnica-Krezel. Embryos were fixed at proper stages with 4% paraformaldehyde at 4 °C overnight, 3 times washed in PBST for 5 min, dechorionated manually in PBST and dehydrated with ascending series of methanol to 100%. Embryos in 100% methanol were stored at −20 °C. After rehydrating with descending series of methanol to PBST, embryos were washed 3 times in PBST, pre-hybridized in 300 μl Hyb-Mix for 3 h at 65 °C. 2 μl of prepared probes were added to 300 μl Hyb-Mix for hybridization overnight at 65 °C. Embryos were placed into 24-well plates and washed with in situ robot BioLane™ HTI by series washing steps: firstly embryos were washed three times for 20 min in 300 μl 50% Formamide at 65 °C, in 500 μl 2x SSCT for 15 times at 65 °C, three times in 500 μl 0.2x SSCT for 20 min at 65 °C, twice in 1 ml PBST for 10 min at room temperature. Then embryos were incubated in blocking solution (2% goat serum (heat-inactivated)) in PBST/BSA (2 mg/ml) for 2 h at RT and incubated overnight with anti-DIG (1:5000 diluted in PBST) at 4 °C. After washing 6 times for 20 min in 1 ml PBST and once for 10 min in 100Mm Tris-HCl, (pH 9.5), embryos were incubated with staining buffer for 15 min and the robot program is finished. We replaced the staining buffer with 500 μl staining solution and stained for a proper time on a shaker. To stop the staining process embryos were washed with stop solution. Stained embryos were fixed in an increasing series of glycerol (25, 50, and 75% in PBST), finally stored in 100% glycerol at 4 °C. These embryos were imaged with Leica MZ APO stereo-microscope using AxioVision SE64 Rel. 4.9.1 software (Carl Zeiss) and the images were processed in Adobe Photoshop CS4 (Adobe).

**ATAC-seq and library preparation.** The omni-ATAC-seq protocol was modified from[59,60]. Fertilized embryos were enzymatically dechorionated with Pronase E (30 mg/ml). 30 embryos were deyolked manually with an eyebrow needle in Danieaus medium, centrifuged (500 × g, 5 min, 4 °C), washed with 100 μl ice-cold PBS, and then centrifuged again. The cells were resuspended by pipetting in 50 μl ice-cold lysis buffer (ATAC-RSB (10 mM Tris-HCl pH 7.4, 10 mM NaCl, 3 mM MgCl2), with 0.1% NP-40, 0.1% Tween 20, and 0.01% digitonin) and incubated on ice for 5 min. Then, the lysate was diluted with ice-cold dilution buffer (1 ml, ATAC-RSB containing 0.1% Tween 20). Nuclei were pelleted (500 × g, 10 min, 4 °C), the supernatant was carefully removed, and the nuclei resuspended in 16.5 μl PBST.

Tagmentation reaction was assembled according to the manufacturer instructions for Illumina small Tn5 buffer and enzyme kit in 50 μl volume and incubated in a thermomixer for 30 min (37 °C, 800 rpm). The sample was purified using Qiagen PCR MinElute Purification kit. The purified transposed mix was pre-amplified for 5 cycles (72 °C for 5 min, 98 °C for 30 sec, 5 cycles of (98 °C for 10 sec, 63 °C for 30 sec, 72 °C for 1 min)) using NEBNext 2x Master Mix (NEB) and Nextera adapters and put on ice. The number of additional cycles was determined using 5 μl of the pre-amplified mixture as in Buenrostro et al.[59]. The total number of PCR cycles was 10–12. The amplified libraries were purified using SPRI beads (Beckmann) according to the manufacturer's instructions. ATAC-seq was performed at two developmental stages, 3.7 hpf (oblong) and 4.3 hpf (dome). At 3.7 hpf, three biological replicates for the WT and two biological replicates for MZspg, MZsox19b and MZsox19bspg were used. At 4.3 hpf, three biological replicates for the WT, MZspg and MZsox19b and four for MZsox19bspg were used. Paired end 150 bp sequencing of resulting 22 ATAC-seq libraries was performed on NovaSeq6000 (Illumina) by Novogene company, with the sequencing depth 80 million reads per library. The raw and processed ATAC-seq data are deposited in GEO (NCBI), accession number GSE188364.

**ATAC-seq data processing and peak calling.** Data processing and analysis were performed on the european Galaxy server[78]. Adapters were removed, the reads were cropped to 30 bp from start, the reads with average quality < 30 and length <25 bp were removed using Trimmomatics. The trimmed reads were aligned to the danRer11/ GRCz11 genome assembly without alternative contigs using Bowtie2[79], with the parameters: dovetail, very-sensitive, no-unal. The aligned reads were filtered using BamTools Filter[80] with the parameters isProperPair true, mapQuality ≥ 30, isDuplicate false, and -reference!=chrM. The aligned reads were filtered further using the alignmentSieve tool from the DeepTools2 suite[81], with the parameters —ATACshift, minMappingQuality 1, maxFragmentLength 110. Thus, only fragments with a size of 110 bp or less were kept. The filtering steps were performed for the individual replicates as well as and for the merged biological replicates BAM files. Filtered reads were converted from BAM to BED using Bedtools[82] and were used as inputs for peak calling. Peaks were called from individual replicates and merged replicates using MACS2 callpeak[83] with the additional parameters 'format BED, nomodel, extsize 200, shift −100' and a 5% FDR cutoff. Regions mapping to unassembled contigs were excluded. Only peaks that were overlapping in merged and each of the single replicates were kept. ATAC-seq peaks overlapping between 3.7 and 4.3 hpf stages in the wild type were centered on ATAC-seq peak summits (4.3 hpf) and cut to 110 bp length. This set of 102 965 accessible regions (ARs, Dataset S3) was used in all subsequent analyses. Bigwig files for each stage and genotype were obtained from merged BAM files using Bamcoverage and used for data visualization in DeepTools2. Bigwig and peak files for each stage and genotype are included in GEO GSE188364 submission as processed files.

**Finding the regions with reduced or increased chromatin accessibility in the mutants.** The coverage of ATAC-seq reads on 110 bp accessible regions (ARs), common in 3.7 hpf and 4.3 hpf stages in the wild type, was scored using Multi-CovBed (Bedtools) in all ATAC-seq experiments. Deseq2[84] was used to normalize the reads and compare chromatin accessibility in each mutant to the wild type. Five MZspg, or five MZsox19b, or six MZsox19bspg replicates were compared to six wild-type replicates. The region was scored as downregulated in the mutant, if the fold change to the wild type was <0 with FDR 5%. The region was scored as upregulated in the mutant if the fold change to the wild type was >0 with FDR 5%. Out of 102 965 ARs, chromatin accessibility was reduced on 4521, 20990, and 20653 regions, and increased on 2046, 12695, and 13071 regions in MZsox19b, MZspg, and MZsox19bspg, respectively. The regions are listed in Dataset S3, Fig. S8a shows the overlaps between the genotypes.

**SoxB1, Pou5f3, and Nanog TF-binding peaks.** SoxB1, Pou5f3 (5 hpf), and Nanog (4.3 hpf) binding peaks were derived from the published ChIP-seq data[10,62]. The FASTQ files were mapped to danRer11/GRCz11 assembly. Alternate loci and unassembled scaffolds were excluded. Peak calling was performed using MACS2 algorithm with default parameters. The peaks overlapping Major Sattelite Repeats (MSRs) were removed. The peaks were extended to 300 bp (±150 bp from the peak summit). SoxB1 and Pou5f3 peaks were split into three groups: SoxB1 peaks, overlapping with Pou5f3 peaks for at least 1 bp (SP), SoxB1-only peaks (S), and Pou5f3-only peaks (P). The peak coordinates in BED format are provided in the supplementary Dataset S4. Accessible regions were considered as bound by TF, if they were overlapping at least 1 bp.

**Motif finding, scoring, and motif enrichment.** De-novo motif finding was performed using MEME suite[85] on the Galaxy server, using 60 bp regions around the summits of differentially regulated ARs groups. The motifs were compared to JASPAR vertebrate non-redundant database[61] using Tomtom[86]. The motifs derived from ATAC-seq were named by the closest match to human TFs in JASPAR, or by TF family (e.g. Homeodomain, b-HLH). In addition, we derived the motifs from ChIP-seq data for the transcription factors SoxB1, Pou5f3[10], 5 hpf, Nanog[62], 4.3 hpf, Tbxta and Tbx16[87] 8–8.5 hpf. Matrices in MEME format for all motifs are provided in the Dataset S4 (motifs). Genomic coordinates of all occurrences for each motif in the 3 kb regions around ATAC-seq peaks were obtained using FIMO[88] with p-value threshold $10^{-4}$ and converted to BED files. To calculate the background frequency of the motifs, we used the background control peak file, where the genomic coordinates of all ATAC-seq summits were shifted 1 kb downstream. The motif densities were calculated as the average number of motifs overlapping with 60 bp around the ATAC-seq or control peak summits. The log2 ratio of ATAC-seq to background motif densities was taken as a motif enrichment value. BED files were converted to Bigwig format with Bedtools and used for visualization of the motif density on the selected genomic regions using deepTools2.

**In vitro nucleosome predictions and GC content.** Nucleosome prediction program from Kaplan et al. 2009[89] was integrated into the Galaxy platform using the Galaxy tool SDK planemo (https://github.com/galaxyproject/planemo) and following the best practices for Galaxy tool development (http://galaxy-iuc-standards.readthedocs.io/en/latest/best_practices.html). The tool was uploaded into the european Galaxy ToolShed (ref. https://www.ncbi.nlm.nih.gov/pubmed/25001293) and is available at the European Galaxy instance. The genomic coordinates of ATAC-seq peaks were extended to ±5 kb to account for the edge effects, and in vitro nucleosome prediction value was derived for every bp. Nucleosome predictions were converted to BigWig files in DeepTools2 for plotting, or to bedGraph files for scoring. GC content was calculated with geecee program in Galaxy. GC content and nucleosome predictions were calculated for 110 bp around ATAC-seq peaks (Dataset S3).

**MNase-seq for MZsox19b, data processing, and visualization.** MNase-seq for MZsox19b at 4.3 hpf and data processing was performed as in[16]. Briefly, 200–400 MZsox19b embryos from mass-crossing cages were collected within 10–15 min. The embryos were enzymatically dechorionated and fixed 10 min in 1% Formaldehyde at the dome (4.3 h post-fertilization) stage. The nuclei were isolated and digested with MNase. The yield and degree of digestion were controlled using Agilent High Sensitivity DNA Kit on Agilent Bioanalyzer, according to

manufacturer instructions. Chromatin was digested so that it contained 80% mono-nucleosomes. Libraries were prepared using the Illumina sequencing library preparation protocol and single-end sequenced on an Illumina HiSeq 2 500 by Eurofins Company (Germany). All further data processing was done using the European Galaxy server useGalaxy.eu. Sequenced reads were mapped to the zebrafish danrRer11 assembly using Bowtie2. Resulting BAM files were converted to BED format, all mapped reads were extended to 147 bp in their 3′ direction, truncated to the middle 61 bp, and converted back to BAM format using BED-Tools. Alternate loci and unassembled scaffolds were excluded from the further analysis. To create MNase Bigwig files for the visualization of nucleosome density, BAM files for MZsox19b (this work), MZspg, and WT (remapped from[20]) were converted to bigwig format using BAM coverage program in deepTools, bin size = 10 bp, normalization—rpkm (reads per million reads per one kilobase). To create log2 MNase Bigwig files for the visualization of nucleosome displacement in MZsox19b and MZspg, the log2 mutant MNase/wild-type MNase ratio was obtained using BigWigcompare program in deepTools, bin size = 10. The heat-maps or profiles of selected genomic regions were plotted using plotHeatmap or plotProfile programs in deepTools. Visualization of selected genomic regions was done in the UCSC browser at https://genome.ucsc.edu/. For the violin plots and nucleosome displacement statistics on the TF-binding or control peaks, average MNase signals per 300 bp were calculated using Bigwig files for each region, and log2 (mutant/WT) ratio was taken (data available as a Dataset S3).

**Chromatin immunoprecipitation (ChIP) for histone marks**. The embryos were obtained from natural crossings in mass-crossing cages (4 males + 4 females). 5–10 cages were set up per genotype, the eggs from different cages were pooled. The freshly laid eggs from MZsox19b, MZspg mutants, and wild type were collected in 10–15 min intervals. Unfertilized eggs were removed at 2–4 cell stage. Collected embryos were transferred to 0.5x Danieau's solution (for 1 L of 30X stock: 1740 mM NaCl, 21 mM KCl, 12 mM MgSO4, 18 mM Ca(NO3)2, 150 mM HEPES buffer, pH 7.6; dilute 60X before use) followed by enzymatic dechorionation with pronase E (0.3 mg/ml). The reaction was stopped by adding 1% BSA (final conc. 0.04%) followed by two to three washing steps with 0.5x Danieau's. The eggs were cultured in glass Petri dishes to prevent the embryos from adhering to the dish and thus eventually rip. They were incubated at 28 °C until the 4.3 hpf stage was reached. In order to fix the chromatin state at developmental stage 4.3 hpf (dome) and avoid nucleosome shifts, the dechorionated embryos were homogenized in 10 ml 0.5% Danieau's containing 1x protease inhibitor cocktail (PIC) using a Dounce tissue grinder and immediately treated with 1% (v/v) Methanol-free Formaldehyde (Pierce) for exactly 10 min at room temperature. The homogenizate was transferred into a 15 ml falcon tube and shaken on a rotating platform for the rest of the 10 min. The fixation was stopped with 0.125 M Glycine by shaking for 5 min on a rotating platform. Subsequently, the homogenizate was centrifuged for 5 min, 4700 rpm at 4 °C, whereupon a white pellet formed. The supernatant was discarded, and the pellet was resolved in Wardle cell lysis buffer (10 mM Tris-HCl (pH 7.5), 10 mM NaCl, 0.5% NP-40, 1–4 ml/1000 embryos). The lysate was dis-tributed upon 2 ml Eppendorf tubes, followed by 5 min incubation on ice with subsequent 1 min centrifugation, 2700 g at 4 °C. The supernatant was discarded again and the pellet was washed two times with 1 ml ice-cold 1x PBST (for 1 L: 40 ml PO4 buffer (0.5 M), 8 g NaCl, 0.2 g KCl, 0.1% Tween 20, pH 7.5). In order to count the obtained nuclei, the pellet was resolved in 1 ml ice-cold 1x PBST, of which 10 µl were diluted 1:1 with 12 µM Sytox® green. The nuclei were scored under a fluorescence microscope using the Neubauer counting chamber.

The residual nuclei were again pelleted by 1 min centrifugation at 2700 g and 4 °C, subsequently snap frozen in liquid nitrogen and stored at −80 °C. The nuclei collected in different days were pooled together to reach the total number of 2.5–3 million, which was used to start one ChIP experiment.

The chromatin was thawn and resolved in 2 ml of Wardle nuclei lysis buffer (50 mM Tris-HCl (pH 7.5), 10 mM EDTA, 1% SDS) and incubated 1 h on ice. In order to shear the chromatin to 200 bp fragments (on average), the chromatin was sonicated using the Covaris S2 sonicator (DC 20%, Intensity 5, Cycles of Burst 200, Time = 3 * 40 cycles with 30 s each (3 * 20 min)). To ensure that the sonication was successful, 30 µl of the sheared chromatin was de-crosslinked with 250 mM NaCl overnight at 65 °C and then analyzed using the Agilent Expert 2100 Bioanalyzer® and Agilent high sensitivity DNA Chip kit.

The lysed and sheared samples were centrifuged for 10 min at 14,000 rpm and 4 °C. Sixty microliters of each sample was kept as input control. The chromatin was then concentrated to 100 µl using the Microcon centrifugal filters (Merck Millipore MRCF0R030) and diluted 1:3 by adding ChIP dilution buffer (16.7 mM Tris-HCl pH 7.5, 167.0 mM NaCl, 1.2 mM EDTA) containing protease inhibitors. The antibodies (listed in the Table S2) were added and incubated overnight at 4 °C on a rotating wheel. 150 µl of magnetic Dynabeads coupled to protein G (Stock 30 mg/ ml; Invitrogen DynaI 10003D) were transferred into a 2 ml Eppendorf tube and placed on a magnetic rack in order to remove the liquid from the beads. Subsequently, the beads were washed 3x with 5 mg/ml specially purified BSA in PBST and 1x with 500 µl ChIP dilution buffer. After removing the ChIP dilution buffer, the chromatin-antibody mix was added and incubated with the beads at 4 °C overnight on a rotating wheel. Beads were pulled down by placing the eppendorf tubes on the magnetic rack in order to discard the supernatant. The beads were resuspended in 333 µl RIPA buffer containing PIC. The Protein

G-antibody-chromatin complex was washed 4 × 5 min on a rotating platform with 1 ml of RIPA buffer (10 mM Tris-HCl (pH 7.6), 1 mM EDTA, 1% NP-40, 0.7% sodium deoxycholate, 0.5 M LiCl), followed by 1 × 1 ml TBST buffer (25 mM Tris-HCl,150 mM NaCl, 0.05% Tween 20, pH 7.6). The beads were pulled down again and the supernatant was removed. In order to elute the chromatin, 260 µl elution buffer (0.1 M NaHCO3, 1% SDS) was added and incubated for 1 h at 65 °C in a water bath. The samples were vortexed every 10–15 min. Afterward, the supernatant was transferred to a fresh eppendorf tube by pulling the beads down, using the magnetic rack. 12.5 µl 5 M NaCl was added to de-crosslink the chromatin and incubated overnight at 65 °C in a water bath. The input samples were treated as control in parallel (230 µl elution buffer per 30 µl input).

Purification of the de-crosslinked chromatin was performed using the QIAquick PCR Purification Kit from Qiagen. The concentration was determined using the Qubit fluorometer and Quanti-iT™ PicroGreen® dsDNA Kit according to manufacturer instructions.

**ChIP quality control and library preparation for histone mark ChIP-seq**. To estimate the signal to background ratio in each ChIP experiment, we have chosen the positive and negative reference genomic regions, enriched in or devoid of chromatin marks. According to previously published data[90], the chromatin region near tiparp gene was highly enriched in H3K27ac and H3K4me3 histone marks at 4.3 hpf, while genomic region near igsf2 gene was not enriched in any of these marks. We performed quantitative PCR in ChIP and Input control material, using the primers for these regions. PCR primers used were: tiparp_f_1 5′ CGCTCCCAA CTCCATGTATC-3′, tiparp_r_1 5′-AACGCAAGCCAAACGATCTC-3′,igsf2_f_2 5′-GAACTGCATTAGAGACCCAC-3′, igsf2_r_2 5′-CAATCAACTGGGAAAG-CATGA-3′. qPCR was carried out using the SsoAdvanced™ Universal SYBR® Green Supermix from BIO-RAD. ChIP and input were normalized by ddCT method, using negative reference region (igsf2). The ChIP experiment was considered successful, if the enrichment in ChIP over input control on the positive reference region (tiparp) was more than 5-fold.

In order to convert a small amount of DNA into indexed libraries for Next Generation Sequencing (NGS) on the Illumina platform, we used the NEBNext® Ultra™ DNA Library Prep Kit. As the DNA outcome of individual WT K27ac Chip and MZspg K27ac ChIP experiments did not reach the input DNA limit for this kit (5 ng), we pooled together the material from two successful ChIP experiments, as well as corresponding inputs, for the library preparation in these genotypes. Two libraries were prepared from two single MZsox19b K27ac ChIP experiments. Single libraries were prepared from single K4me3 experiments in three genotypes. The library preparation was carried out according to manufacturer instructions, with the modifications indicated below. The library preparation follows a 4-step protocol including end-repair (5′ phosphorylation, dA-tailing), adapter ligation, PCR enrichment including barcoding and clean up. Since the DNA input was <100 ng, in the adapter ligation step, the NEBNext Adaptor for Illumina® (15 µM) was diluted 10-fold in 10 mM Tris-HCl (pH 7.8) to a final concentration of 1.5 µM and used immediately. At the final clean-up step, the reaction was purified by mixing the samples with AMPure XP Beads (45 µl) before incubating at room temperature for 5 min. After a quick spin using the tabletop centrifuge, samples were placed on a magnetic rack and supernatant was discarded. 200 µl of 80% Ethanol were added and removed after 30 sec, two times. The beads were air-dried for 3 min. and the DNA target was subsequently eluted by adding 33 µl of 0.1x TE (pH 8) and incubating at room temperature for 2 min. 28 µl of the library were transferred to a fresh PCR tube and stored at −20 °C. 2 µl of the sample were diluted 5-fold with 0.1x TE and used to check the size distribution of the library using Agilent Expert 2100 Bioanalyzer® and Agilent high sensitivity DNA Chip kit. In order to reduce the peak of residual unligated adapters, the reaction was re-purified, by adding H2O up to 50 µl and 45 µl of AMPure XP Beads. The concentration was determined using the Qubit™ Fluorometer and Quanti-iT™ PicroGreen® dsDNA Kit. The seven ChIP-seq libraries were sequenced at 70 mln paired end 150 bp reads each: WT K27ac Chip, MZspg K27ac Chip, MZsox19b K27ac Chip1, MZsox19b K27ac Chip2, WT K4me3 Chip, MZspg K4me3 Chip, MZsox19b K4me3 Chip. The seven corresponding input libraries were sequenced to 30 mln reads. Sequencing was performed by the Novogene company (China).

**H3K27ac and H3K4me3 ChIP-seq data analysis and visualization**. H3K27ac and H3K4me3 ChIP-seq data processing was done using european Galaxy server useGalaxy.eu. Sequenced reads were mapped to the zebrafish danRer11 assembly using Bowtie. Alternate loci and unassembled scaffolds were excluded. To create H3K4me3 Bigwig files for data visualization, the log2 ratio between each ChIP and merged inputs (in rpkm) was obtained using BAM compare program in deepTools, bin size = 10. To create H3K27ac Bigwig files, the log2 ratio between each ChIP and merged inputs (in rpkm) was obtained using BAM compare program in deepTools, bin size = 10. We first used two biological replicates of H3K27ac MZsox19b to address, if there is a difference in H3K27ac mark between the gen-otypes—in this case, the difference between the genotype should be higher than the difference between the biological replicates within one genotype. We used deep-Tools2 mutliBigwigcompare tool to divide the whole zebrafish genome into 1680169 1 kb bins, and to compute a similarity matrix between four Bigwig files: H3K27ac MZsox19b-rep-1, H3K27ac MZsox19b-rep-2, H3K27ac MZspg, and H3K27ac WT. Pairwise Pearson correlations and Principle Component Analysis

(PCA) were performed using plotCorrelation and plotPCA tools. In both analyses, the difference between the biological replicates was negligible when compared to the difference between the genotypes (Fig. S9a, b). Close correlation between the H3K27ac MZsox19b biological replicates demonstrated the reproducibility of our ChIP-seq protocol and validated the use of single replicates for our purposes. For the subsequent analysis, we used merged replicates of H3K27ac MZsox19b. The heatmaps or profiles of H3K27ac levels on selected genomic regions were plotted using plotHeatmap or plotProfile programs in deepTools. Histone modification profiles in single genes were visualized using the UCSC browser.

**Selection of putative enhancers and promoters from accessible regions**. AR peak was scored as putative promoter, if it was marked with both H3K4me3 and H3K27ac in the wild type, or if it was marked by H3K4me3 in the wild type and mapped within 500 bp of the annotated transcription start site (ENSEMBL transcript, or zebrafish promoters annotated by CAGE in[40]). AR peak was scored as putative enhancer if it was marked by H3K27ac in any genotype and was not a promoter. All other elements were scored as "other". H3K4me3 and H3K27ac marks were assigned to ARs using arbitrary thresholds. For H3K4me3, the threshold was log2 ChIP-seq/input not less than 3 (8x enrichment) in at least 10 bp within 500 bp around AR summit. For H3K27ac, the threshold was log2 ChIP-seq/input not less than 2 (4x enrichment) in at least 10 bp within 1 kb around AR summit. The mean positive H3K4me3 ChIP/input values were calculated in 500 bp around the peak, the mean positive H3K27ac ChIP/input values were calculated in 1 kb around the peak (Dataset S3).

**Gene Ontology analysis**. For the genomic regions, we used GREAT analysis[91] at http://great.stanford.edu/great/public-3.0.0/html/. Genomic regions were converted to zv9/danrer9 genomic assembly, which is the only assembly available for this server, using Liftover Utility from UCSC and associated with genes using 20 kb single nearest gene association rule. The categories were ranked by ascending FDR value. For the transcript groups A–K we used DAVID[92]. The enriched GO: categories were ranked by ascending p-value (cutoff 0.05).

**RNA-seq time curves: material collection, processing, and sequencing**. The freshly laid eggs were obtained from natural crossings in mass-crossing cages (4 males + 4 females). Five to ten cages were set up per genotype, the eggs from different cages were pooled. At least 600 freshly laid eggs per each genotype collected within 10–15 min were taken for single experiment. To match the developmental curves as precisely as possible, the material was simultaneously collected for two genotypes in parallel. In each of five experimental days, the material was collected for two genotypes, as specified below: MZsox19b-rep1 and MZspg-rep1, WT-rep1 and MZspg-rep2, WT-rep2 and MZsox19b-rep2, WT-rep3 and MZspg-rep2, WT-rep1 and MZsox19spg-rep1, WT-rep4 and MZsox19spg-rep2. 45–60 min after the egg collection, we ensured that the embryos of both genotypes are at 2–4 cell stage, removed non-fertilized eggs and distributed the embryos to 8 dishes per genotype, 40–45 embryos per dish, to obtain 8 time points per genotype. The temperature was kept at 28.5 °C throughout the experiment. The embryos from one dish per genotype were snap-frozen in liquid nitrogen every 30 min, starting from 2.5 hpf (pre-ZGA, 256 cell stage, 8th cell cycle) till midgastrula (6 hpf). In total, we collected 4 biological replicates of the time curve for the wild type, and two biological replicates for each of MZsox19b, MZspg and MZsox19bspg mutants.

Total RNA was isolated with QIAGEN RNeasy kit following the user's manual. We checked RNA quantity and quality by using Agilent RNA 6000 nano Kit on Agilent Bioanalyzer, according to manufacturer instructions. The 6 hpf time point from MZsox19bspg -rep1 had to be excluded, due to non-sufficient RNA quality; the corresponding WT-rep1 time point was also removed. Poly-A enriched library preparation and single-end 50 bp sequencing (35 M reads per sample) on Illumina platform was performed by NOVOGENE company (China) for each sample. Base calling was performed with Illumina Casava1.7 software. To estimate the non-biological variation between the samples, we prepared and sequenced two technical libraries for each time point in the following time curves WT-rep1, MZsox19b-rep1 and MZspg-rep1.

**RNA-seq time curves: data processing and visualization**. FASTQ files were further processed on european Galaxy server useGalaxy.eu. All sequenced clean reads data were trimmed on the 5′ end for 5 bp by Trim Galore! program according to the sequencing quality control. Trimmed reads were mapped to danRer11 using RNA STAR[93] and ENSEMBL gene models. Number of reads for each gene was counted in Feature count[94]. Feature counts were cross-normalized using DEseq2[84]. This processed RNA-seq data table is deposited in GEO with the number GSE137424. To estimate biological and technical variance across the samples, we excluded non-expressed ENSEMBL gene models ("0" reads throughout all experiments) and calculated pairwise Pearson correlations between biological replicates (the same time point and the same genotype, material from different experimental days) and also between technical replicates (two libraries prepared from the same material). Pearson correlation coefficients for biological replicates were at the range 0.93–0.98, which indicated a very strong linear relationship and was close to the technical variance range (Pearson correlation coefficients 0.96-0-99

for technical replicate pairs). To account only for biological variance, we excluded technical replicates from the further analysis. The normalized expression data used for analysis (biological replicates only) is available as a source file. This data was used for gene expression visualization, statistics, and as an input for RNA-sense (see sub-chapter "RNA-sense program" below for explanations). The data visualization was done in R with custom R scripts (box plots, violin plots, heatmaps) or in Excel (individual genes, averaged time curves). For the heatmaps and summary graphs, the biological replicates for each genotype were averaged. For the heatmap view of gene expression change, we calculated the maximal expression value across the time curve for each transcript (max) in all four genotypes. Expression/max ratio for each time point was plotted. To compare the zygotic expression changes between the transcript groups, we calculated the log2 ratio of the expression in each time point from 3 to 6 hpf to the expression at 2.5 hpf for each transcript. The averaged results are shown as box plots with 1-way Anova and Tukey–Kramer test statistics, or as mean time curves (error bars show Standard Error of the Mean). For the time curves of the individual transcripts, the line graphs were drawn in Excel in linear or logarithmic scale (error bars show SEM of biological replicates).

**RNA-sense program**. In order to facilitate the biological interpretation of time-resolved RNA-seq data, we developed a 3-step procedure called RNA-sense (Movie S2). In principle, the usage of RNA-sense is not only restricted to RNA time series. RNA-sense can be applied to compare two groups of data series to capture the differences in the dynamic changes between the groups. The series data could be temporal, spatial, or any other continuous condition like the series concentration of drug treatment. The data itself could be any sequencing data, including DNA, RNA and protein, or any other comparable large datasets.

In step one, time-resolved RNA-seq data in one of two conditions, e.g. the wild type (user-defined parameter Experiment step detection = "WT"), are analyzed with respect to their temporal profile. The transcripts expressed below a user-defined threshold are excluded from the analysis. First, for each gene and for each measurement time point t, dynamic data is split into two groups before and after (after and equal to) time point t. The data is fitted by both a one-step model (two different means before and after time point t) and by a constant model (mean over all data points) and models are compared pairwise by means of likelihood ratio tests for each time point. If the one-step model is significantly better (with user-defined p-value cutoff, pVal switch) than the constant model, a switch is detected for this time point. The difference of the means before and after the time point defines the direction of the switch "up" or "down". If switches were detected at different time points for each gene, the first possible time point is chosen.

In step two, fold changes between wild-type and mutant data are analyzed. For each gene and for each time point, Robinson and Smyth exact negative binomial test (with user-defined p-value cutoff, pVal FC) is performed to determine whether genes are significantly up or downregulated in the mutant with respect to wild type. The function exact.nb.test from the R package NBPSeq is used for analysis (https://cran.rstudio.com/web/).

In step three, the results of step one and two are combined. Genes are grouped in a matrix form with respect to switch time (y-coordinate) and mutant fold change (x-coordinate). Genes for which fold change was detected at several time points appear several times in the matrix. For each tile of the matrix, Fisher's exact test for non-randomness is performed to analyze the correlation between the two properties switch time and fold change detection. Tiles with a low p-value in the Fisher test show a high correlation between switch time and fold change. This can be interpreted as a high number of genes for which the switch point is shifted in time in the mutant condition.

RNA-sense is a flexible tool with several user-defined parameters:
-Experiment step detection: tells which of two conditions (i.e. WT or mutant) should be used for switch detection in Step 1
-threshold—the transcript is included in the analysis, if the expression value in at least one data point reaches the threshold.
-pVal switch—p-value threshold for switch detection in Step 1
-FC—fold change value threshold (optional) for Step 2
-pVal FC—p-value threshold for fold change analysis at Step 2
In our time series data, a single 6 hpf data point was missing in one of the two biological replicates for MZsox19bspg and in parallel WT control sample. To get input for RNA-sense program, these 6 hpf data points were replaced by the mean of the other replicates. RNA-sense analysis was run with the following user-defined parameters: pVal switch = 0.15, pVal FC = 0.01, FC = 2, threshold = 100.

The code and example files for automatically performing the 3-step procedure are available in the R-package RNAsense that was developed jointly with the paper is available on Bioconductor https://bioconductor.org/, https://doi.org/10.18129/B9.bioc.RNAsense.

**Analysis of RNA-seq time curves using RNA-sense program**. To identify maternal and zygotic transcripts in the wild type, Step 1 of RNA-sense was performed for the WT condition, with the threshold 100 and pVal switch 0.15. Switching UP transcripts were considered zygotic, switching DOWN transcripts were considered maternal. To validate the assignment of zygotic and maternal transcripts identified in our study, we linked our transcript list to the previously defined zygotic and maternal groups from three studies[9,39,40] via ENSEMBL gene name. To assign the regulation status for each transcript, RNA-sense was

performed for the WT against each mutant, with 2-fold change and *p*-value < 0.05 cutoff (FC = 2, pVal FC = 0.05 at RNA-sense step 2). Zygotic transcript was considered to be downregulated in the mutant if it was downregulated at one or more time points at or after the switch UP. Maternal transcript was considered to be upregulated in the mutant, if it was upregulated at one or more time points after the switch DOWN. The zygotic and maternal transcripts were grouped by regulation in the mutants to 8 non-overlapping groups each. The list of zygotic genes, downregulated or unchanged in the mutants, the list of maternal genes upregulated or unchanged in the mutants, and all accompanying information is provided in Dataset S1.

To identify the genes zygotically upregulated in the mutants, we needed to consider the transcripts that are absent in the wild type and appear in the mutants only. For this purpose, Step 1 of RNA-sense was performed for each of the mutant conditions against the wild type, using the same parameters as above. The transcript was considered as zygotically upregulated in the mutant if (1) the transcript was switching UP in this mutant (2) the transcript was upregulated at least 2-fold in the mutant compared to the wild type, at least one time point after the switch. This procedure resulted in the list of 1062 transcripts. Out of them, 27 transcripts were upregulated in all three and 224 in two mutant genotypes. These transcripts were assigned as upregulated to the genotype where maximal expression over the time curve was the highest (e.g. using these criteria, *nog1* was assigned as upregulated in MZ*sox19bspg*). Three non-overlapping groups were assigned: MZ*spg* > WT, MZ*sox19b* > WT and MZ*sox19bspg* > WT. The list of zygotic genes, upregulated in the mutants, and accompanying information is provided in the Dataset S2.

**Chi-squared test and other statistical analysis**. To estimate the over- or underrepresentation of ARs in the putative regulatory regions of up- or down-regulated zygotic genes, chi-squared test was used. Two lists of zygotic transcripts: expressed in the wild type (Dataset S1, zygotic) and upregulated in the mutants (Dataset S2) were concatenated. The duplicates of transcripts (which are zygotically expressed in the wild type and also upregulated in one of the mutants) were randomly removed. ARs were linked to the transcripts within ±20 kb from TSS (Dataset S5). Dataset S5 was used to derive the input files for all chi-squared tests. Chi-squared tests were performed using chisq.test function and visualized using corrplot function in R. Statistical comparisons of three or more samples were performed using one-way ANOVA and Tukey–Kramer Test in R. Two samples were compared using Student two-tailed *t*-test in Excel.

**Reporting summary**. Further information on research design is available in the Nature Research Reporting Summary linked to this article.

## Data availability
The RNA-seq data generated in this study have been deposited in the GEO database under accession code "GSE137424". The ATAC-seq data generated in this study have been deposited in the GEO database under accession code "GSE188364". The H3K27ac and H3K4me3 ChIP-seq data generated in this study have been deposited in the GEO database under accession code "GSE143306". The MZ*sox19b* MNase-seq data generated in this study have been deposited in the GEO database under accession code "GSE125945". The WT and MZ*spg* MNase-seq data used in this study are available in the GEO database under accession code "GSE109410". The ChIP-seq data for Pou5f3, SoxB1, and Nanog TF binding used in this study are available in the GEO database under accession codes "GSE39780" and "GSE34683". The source data underlying main Fig. 3b, d, e, Fig. 4a, c, Supplementary Figs. S3a, S4a, c, and S5 are provided as a Source Data file. The source data underlying main Fig. 3b, c, d, Supplementary Figs. S4b and S5 are provided as a Dataset S1. The source data underlying main Fig. 4a, b are provided as a Dataset S2. The source data underlying main Fig. 5a, b, d–g, Fig. 6b, c, e, f, Fig. 7a–c, and Fig. 8a, b, Supplementary Figs. S6, S7, S8a–e, and S9 are provided as a Dataset S3. The source data underlying main Fig. 5e, f, Fig. 6a, d, Fig.7c, and Supplementary Figs. S7 and S8b are provided as a Dataset S4. The source data underlying main Fig. 5c, Fig.7d, Supplementary Fig. S8f, g are provided as a Dataset S5. All other relevant data supporting the key findings of this study are available within the article and its Supplementary Information files or from the corresponding author upon reasonable request. Source data are provided with this paper.

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

## Acknowledgements
We are grateful to Sebastian Arnold, Rainer Duden, and Christina Gross for commenting on the manuscript, to Sabine Götter for excellent fish care, and to Andrea Buderer and Cornelia Wagner for administrative support. This work was supported by DFG-ON86/4-2 for DO, DFG-EXC2189—Project ID: 390939984 for D.O. and J.T. S.V.U was supported by RFBR (20-54-12022), by the Interdisciplinary Scientific and Educational School of Moscow University 'Molecular Technologies of the Living Systems and Synthetic Biology', and by RSF (21-64-00001, zebrafish experiments). The Freiburg Galaxy Team is funded by DFG grant SFB 992/1 2012 and BMBF grant 031 A538A RBC.

## Author contributions
M.G., M.V., A.J.R., A.G., and L.B. and S.V.U. performed the experiments; M.G., B.G., L.Y.Y., and D.O. analyzed the data, M.R. and H.H. wrote RNA-sense program, D.O.—design of the study, J.T. and D.O.—supervision, funding acquisition, M.V. and D.O. wrote the manuscript, M.G., M.R., L.Y.Y., and J.T. edited the manuscript.

## Funding

## Competing interests
The authors declare no competing interests.
