## [Peer Review File · Nature Communications]

Reviewers' Comments:

Reviewer #1:

Remarks to the Author:

The manuscript by Gao et al addresses the independent roles and interplay between two key pluripotency factors, Pou5f3 and Sox19b during development. This is an important question, with relevance for understanding both zygotic genome activation and the pathways regulating early embryogenesis.

The authors uncover a rather complicated relationship, which they do a good job of explaining at a macroscopic level. At a more granular level, there are patches of the manuscript (especially toward the end of the results) that are a bit more challenging to follow. These concerns are somewhat alleviated by thoughtfully prepared figures. Data and software availability is presented in a very clear and accessible way. The methods provide very good detail for reproducibility, with a few exceptions. Most notably, this reviewer was unable to find the Key resources table cited in the methods, which presumably would have contained antibody information, morpholino sequences ect. Additional omissions are highlighted below, along with concerns and requests for clarification. Comments have been grouped to the most relevant figure for clarity

Figure 1:

1b: The author's sox19b mutant is an indel. Recently, there has been significant data suggesting that nonsense mediated decay of transcripts from these types of mutations can trigger poorly understood compensation mechanisms (see for example El Brolosy et al 2019). Ideally, transcript analysis of sox19b mutant should be presented in supplemental data to clarify whether RNA in this mutant is subject to nonsense mediated decay, and the authors should comment on this issue in the text.

Fig 1b /S1: While the imaging approach is clear, the way the authors are calculating the rate of cell division is not fully clear from the methods section. It seems the method used in Fig 1b and S1a are different. Ideally, the 10th cell cycle should be included in fig S1a with statistics indicating significance. In addition, some quantification (of cell size/number of nuclei?) should accompany the image in figure 1b. At present, cycle 10 appears to be the time point when the authors suggest lengthening of the cell cycle occurs, yet it seems to be the only time point lacking quantification.

Figure 1c: The images in figure 1c are of remarkable quality. However, they represent only a single embryo. Two additional embryos are shown in the supplementary movie. Because developmental delay is a trickier phenotype to assess, Ideally, I'd like some more detail here. How many embryos were examined, and was the extent of delay comparable between all embryos? Also, what was the method of generating embryos? Given these are maternal zygotics, I'm assuming these were generated by comparing wtXwt crosses to crosses between homozygous mutants. If this is the case, were embryos from multiple independent crosses compared, and if so, how many?

Figure 2:

Figure 2b: How can the authors be sure that their Morpholino deletion of SoxB1 is a complete knockdown of the proteins? If it were to be a partial knockdown, could that impact interpretation? There is also some concern re morpholino toxicity causing delay in these experiments. St. control morpholinos tend to provide poor controls, as all morpholinos behave differently. Rescue in supplemental data seems partial, and numbers aren't provided. Caveats for this experiment should be more explicitly stated, and controls clearly articulated.

The authors write: "We concluded that maternal and zygotic functions of SoxB1 proteins are uncoupled: zygotic SoxB1 activity becomes critical starting from the end of gastrulation, maternal Sox19b protein sets the timing of gastrulation"—I'm somewhat uncomfortable with this statement. Many mutants cause developmental delay, often due to indirect effects. While it is clear that sox19b loss impacts the timing of gastrulation, I'm not sure that the data presented supports the idea that maternal sox19 proteins set the time of gastrulation. Perhaps more cautions wording could be applied here.

Fig 2c: In addition to providing the reference, the authors should comment on whether spg is thought to be null allele in the text, as this is important for interpretation of their data.

Fig2. In figure 2 it would be helpful if more extensive comparison of the MZ sox19b, MZspg and double MZ mutants was presented in the main figure. In particular, 2 c and 2d would be easier to understand if the single MZ mutants were included. Number of embryos with phenotype/ number assessed should be indicated for all panels, as well as panels in s1 e and f.

Figure 3.

The authors write : In sum, we found that Sox19b and Pou5f3 act as independent pioneer factors during maternal-to-zygotic transition. They displace nucleosomes by binding to different consensus motifs. Non-consensus binding of two factors results in the opposing effects of local chromatin accessibility genome-wide. However, as best this reviewer can see, binding was not assessed in these experiments, only nucleosome displacement. If Pou5f and sox19b Chip data is incorporated here, it needs to be more clearly stated. If this data only assesses nucleosome displacement the more careful language should be used to delineate what is shown vs inferred.

The methods suggest this analysis and other genome wide was performed on Zv9 assembly, which is quite old. Is this correct? Although it likely will not affect their conclusions, ideally analysis would have been performed using the Zv11 assembly which has been available for quite a while.

Figure 4

The methods seem to suggest that for the following libraries there was only replicate: WT K27ac Chip, MZspg K27ac Chip, WT K4me3 Chip, MZspg K4me3 Chip, MZsox19b K4me3 Chip. And then two replicates for MZsox19b. Can the authors clarify? If only one replicate was actually performed, can the authors please provide the rational/justification for the lack of replicates, as it seems insufficient. Assuming these are on pooled embryos, how many? From an individual pairwise cross or multiple pairwise matings?

Figure 5

Here, and throughout the manuscript, the authors seem to refer to ZGA as occurring as burst at the 1000 cell stage, whereas current data (see review by Vastenhouw et al 2019) suggests this is a more gradual process that initiates around the 128 cell stage. The authors may want to revise their writing to better reflect this current view.

For RNA seq, the authors write "In total, we sequenced 78 samples in four biological replicates (Fig.5a) and 23 technical replicates for WT and single mutants. However, it is difficult to parse out how many replicates of what, and which were biological vs technical from this statement. Authors should be clearer on this point. Assuming these are on pooled embryos, how many? From an individual pairwise cross or multiple pairwise matings?

Figure 5 is by far the most complicated figure. Stylistically, reducing reliance on abbreviations w, s, p and d and instead writing these out (where possible) would likely reduce the mental effort required of the reader when assessing this figure.

Discussion:

The authors do a nice job of summarizing their conclusions, however the analysis they perform is complicated. Additional discussion of the assumptions made in this study and caveats for interpretation would be welcome (see some suggestions in comments above).

Reviewer #2:

Remarks to the Author:

In "Pluripotency factors select gene expression repertoire at Zygotic Genome Activation", by Gao et al investigate the contributions of two transcription factors, Pou5f3 and Sox19b, in zygotic genome activation in zebrafish. This is an interesting study that certainly has the potential to significantly further our understanding of the genome activators, Sox19b and Pou5f3, in zebrafish.

However, the current manuscript is very hard to follow and it is therefore difficult to assess whether the data supports the conclusions. We therefore think it would be premature to publish this work.

We suggest that the authors clarify their manuscript to help the reader reach the conclusions and interpretations drawn. We would be happy to review the work again when this has been done.

Below we have listed some of our concerns and suggestions that will hopefully help the authors to improve their manuscript.

SPECIFIC COMMENTS

Introduction (page 3)

- End of paragraph 1: the authors should include the following citation for Dux in mouse and human: Hendrickson et al. 2017, doi: 10.1038/ng.3844
- Pou5f3 and Sox19b are introduced as TFs with pioneer activity "In case of several genome activators, like Pou5f3, Sox19b, and Nanog in zebrafish, it remains an open question, how their broad pioneering activity at ZGA relates to their different functions later in development". The authors do not present any evidence that support the claim that Pou5f3, Sox19b, and Nanog are established pioneer factors. Although genomics data correlate nucleosome displacement with their binding and suggest that the zebrafish homologs can behave like other genome activators or like their mammalian counterparts, this is a model and there is no biochemical data supporting the pioneer factor feature (direct binding to nucleosome). Please clarify this in the text and provide references.
- The authors vaguely mention the studies on the "four different scenarios" for the mechanism of interaction between Pou5f1 and Sox2 with one another and with chromatin. Please elaborate on this to provide a more molecular context that help to nuance between the models and set up the problem for how the current study will contribute to our mechanistic understanding of Pou5f3 and Sox19b interaction in genome activation.

Figure 1 and related

- Fig. 1a: The authors begin this figure with a time point at which the MZsox19b embryo is already developmentally delayed (15-min). As a reference point, please start with images showing no detectable delay between the 2 genotypes (for example, a stage between the 6-9th cell cycle).
- Fig. 1: There is no mention (in the legend or Methods section) of how many embryos were used in this figure, nor the number of experiments these results represent.
- Fig. S1a: why is the 10th cell cycle quantification not included?

Figure 2 and related

- Fig. 2a: First, the use of 8 hpf is questionable, as Fig.1 had no 8 hpf. Presumably because all 4 SoxB1 members are expressed by this stage. In fact, they are already present by 30% epiboly (near 5hpf). Second, it is clear that the RNA expression patterns for sox2, sox3, and sox19a are already different between the two genotypes because of the time delay in MZsox19b. Although the authors want to make the point that the other Sox members are still present in MZsox19b, it is important to check that the expression pattern is also intact. For this, please include in situ data that match with earlier wild-type time point.
- Fig. 2b: We are puzzled by the results shown here and in Fig. S1D to make the point of the

specific maternal and zygotic functions of SoxB1 proteins.

- The rescue experiment (via injection of sox19b mRNA) should be included in the main figure.
- We agree that with the conclusion that "zygotic SoxB1 activity (although please specify Sox2/3/19a) becomes critical starting from the end of gastrulation" based on the panels WT QKD and MZsox19b TKD, where in the absence of Sox2, Sox3, Sox19a protein, post-gastrulation defects are obvious.
- However, we do not agree with the authors' conclusion that "maternal Sox19b protein sets the timing of gastrulation" (page 4). If this were the case, one would expect that the conditions in which there are still maternal Sox19b protein present (WT QKD, MZsox19b TKD + sox19b mRNA), the timing of gastrulation would be restored and the phenotype displayed at 10 hpf should be the same as WT. As they do not match, and there is still a delay, then the available early expressed Sox19b protein cannot correct for the timing of gastrulation. [unless it is not sufficient, or proper dosage not reached, but this is not a point the authors make].
- Fig. 2c-e (where 2e is erroneously called 2b): We do not think there is sufficient evidence from 2cde to draw models about the distinct functions of Pou5f3 and Sox19b. The ratio of Sox19b and Pou5f3 is brought into consideration, yet only maternal mutants were studied and to bring quantitation into the model, the authors would need to play with the levels of individual TFs (as RNA or protein contribution).
- The models in 2f-h do not fit here as no evidence has been presented yet to support them. In addition, Fig. 7 has a similar schematic diagram already.
- The model proposed for 2g is at odds with Fig. 6d. See below***.

Figure 3 (and related)

- End of page 4-start of page 5: The choice of 4.3 hpf is not rationalized, and considering how Figure 1 and 2 examined various time points, and looked at the contributions of each factor in the both gastrulation and post-gastrulation events, unclear.
- Page 5: about the statement "SoxB1 proteins can recognize two types of cognate sequences: bipartite pou:sox motifs, also bound by Pou5f3, and sox motif" we note that
 - As it reads, the same could also be said about Pou5f3: Pou5f3 can also recognize two types of cognate sequences: bipartite pou:sox motifs and pou motif.
 - Pou-only motifs are not discussed at all here. We looked at Veil et al. and all of the Onichtchouk lab previous papers, and there is never any mention of the Pou motif alone.
 - So we wonder, are there no Pou motifs that exist (or enriched) independently of Sox according to the authors? Please clarify.
- It would be useful if some genome browser snapshots would be provided that contain ATAC-seq, MNase-seq, and ChIP-seq data, to highlight examples of genes that reflect different binding affinities between Pou5f3 and Sox19b to their motifs.

Figure 4 (and related)

- Bottom of Page 5: "we immunoprecipitated embryonic chromatin" – please remove "embryonic"
- Fig. 4a, b
 - It is not immediately clear how panel a and b compare. Part of the confusion stems from the use of 1-5 already in this panel while in the text this solely refers to enhancer types.
 - If the enhancers in b are a subset of a, it is unclear how the H3K27Ac signal in b (wt, type 1) can be higher than in a.
 - The cartoons of embryos in the "nearest genes" column is not helpful.
- Fig. 4c
 - The accompanying text is not correctly described and should be rewritten to match with the more

quantitative aspect of the data presented in the figure. "... H3K27ac deposition by each TF depended on the presence of sox or pou:sox motifs in the genomic regions". This reads as a qualitative approach the authors have taken. It should say instead "the presence and number". Please comment on the changes in H3K27ac levels between wild-type and mutant in the main text, with respect to number of TF motifs.

- Fig. 4d-g and Section "Four types of differential enhancers define four types of transcriptional response to Pou5f3 and Sox19b"

- This section needs to be rewritten. It is not written in a linear manner and it does not inclusively describe all four enhancer types even though it sets out that way. Clusters 3&4 are singled out, although the "4/7 top cluster genes ..." is not referred to by the authors in the main text as Cluster 4, so it reads as if the 4/7 refer to the 7/13 from Cluster 3. This is only pointed out in Fig. S4D legend.

- Fig. 4d-g and especially Fig. S4 are noisy figures, and individual panels should be referred to in the text rather than as a whole, to attribute the regulatory insights to specific enhancer types.

- To look at the differential regulation among the enhancer types, the authors want to relate TF binding, H3K27ac deposition, and transcriptional output (through H3K4me3 as shown in the figure), although the latter approach should be outlined in the main text.

- Would it be better to represent Fig. 4d-g, Fig. S4, and Fig. 5a grouped by enhancer types? The reader has to flip through 3 figures to correlate TF binding, H3K27ac deposition, RNA levels.

- Table S2: regarding the results from GREAT analysis: for clusters 1,4,5 – terms in the top 5 rankings are listed, but for clusters 2,3 – noticeably other terms that are not related to gastrulation have been left out. Specifically, Cluster 3 shows the top 7 terms with Rank3&4 not included. And Cluster 2 covers a great range of Ranks. Please comment. Consider including a list of the top 20 and highlight the gastrulation terms, rather than cherry-pick.

- Page 6: text to fix

- o "Hence Pou5f3 binding to type 1 and 2 enhancers displaced nucleosomes and directly induced local H3K27 acetylation". Please remove the word "directly". The data is suggestive, but biochemical data is missing to support a direct interaction between Pou5f3 binding and acetylation effects. There could be another TF involved, or accessory proteins to acetylation writers.

- o "Sox19b and Pou5f3 can act on type 2 enhancers by either co-binding pou:sox motif or by binding different motifs". The latter is vague. The author only mentions pou:sox and sox motifs, so there is no need for the authors to consider other motifs if they strictly focus on these two motifs.

- o "Repression... not associated with pioneer activity of repressing TFs ..." we cannot grasp the meaning of this sentence.

- o Fig. S3: We do not agree with the the conclusion "Since type 4 enhancers were bound by Nanog and two maternal TFs implicated in mesendoderm specification, Eomesodermin and FoxH1..." According to Fig. S3ch, Eomes, FoxH1, and Nanog binding is enriched in all enhancer types, and even to a lesser extent on Type 3 and 4 enhancers.

Figure 5 (and related)

- Fig. S6a: It is not clear what the +/- in the matrix mean.

- Fig. S6c and Table S3: Comparative analysis among the RNAseq data is not described anywhere, and references are missing in the legend. The columns in Table S3 that refer to other studies are not defined (what does No/Yes or numbers refer to in each column in the comparative analysis?)

- Fig. 5c & Fig. S6d: We find it difficult making the jump from a matrix of a single gene (S6a, which is not full and shows only a few time points) to a matrix of combined genes (5c and S6d). Beneficial to have more matrices from several genes as examples and supporting evidence, so that it makes interpretation of 5c and S6d easier to process. We can't even say we agree with the authors or not, because we go from a single layer (single gene) to multi-layer matrix that incorporates all the RNAseq data.

- Fig. 5d: We find the Venn diagrams much easier to interpret than 5c. Agree with the authors' conclusions that single mutants have a greater effect on zygotic transcripts expression, and thus

transcriptional delay, compared to the double mutant. And this mirrors the developmental delay in gastrulation observed in Fig. 1. However, the authors have left out the number of transcripts that are downregulated in the single MZspg that do not overlap with the double mutant or MZsox19b (ie. there are no numbers in only the blue part of the Venn diagrams). Please indicate them so that the observed effect mentioned above applies is also reflected in transcripts that do not overlap with MZsox19b. Only including the effects of single TF LOF on shared AND non-shared transcripts can the conclusion "Pou5f3/Sox19b balance appeared to be more important for timely onset of bulk transcription, than the presence of both factors in the embryo".

- According to Lee et al. (doi: 10.1038/nature12632)), comparative analysis among the single loss-of-function mutants/morphants indicate that Nanog LOF > MZspg > SoxB1 QKD in terms of their transcriptome-wide effects. This is inconsistent with the authors' data that MZsox19b has a greater effect on the zygotic transcriptome compared to MZspg. Can the authors comment on this?

- Fig. 5ef: Main text on page 8 - Not sure if it is the intention of the authors to relate genes regulated by Pou5f3 and Sox19b to specific enhancer types discussed in Figure 4. If so, please clarify on the choice of targets that are selected. What does "general ZGA delay"? Do the authors consider transcripts that are downregulated in the double mutant and non-overlapping with single mutant as roles in general ZGA delay?

Figure 6:

- Fig. 6d:
 - incorrectly referred to as Fig. 6g on page 8
 - For the dorsalization experiments, in Fig. 6d and Fig. 2cde, the authors fluctuate between the use of Msox19bsp and Msox19bsp +/- . Please clarify this discrepancy. Do they refer to the same thing and the +/- means zygotic contribution from wild-type sperm? Or is there a heterozygous line used. The author proposes a model that either Sox19b or Pou5f3 should be maternally present in the embryo for correct D/V patterning, then the usage of the heterozygous argues against the either TF needs to be maternally present "to safeguard normal D/V" patterning, since the heterozygote would have dosage of both Sox19b and Pou5f3.
 - ***The authors draw a conclusion that is at odds with what was proposed in Fig. 2g for the functions of Pou5f3 and Sox19b in D/V patterning:
Fig. 6d: "the synergistic activity of Pou5f3 and Sox19b at ZGA controls D/V axis formation"
Fig. 2g "D/V patterning depends on either Pou5f3 or Sox19b to be maternally present".

Discussion:

- The discussion is rather short and could benefit from extended speculations and clarifications on how Pou5f3 and Sox19b interact independently or sequentially on chromatin and specific targets and how this translates to their synergy in biological processes like D/V patterning. The molecular aspects from genomic studies (Figures 3, 4) and their function in biological processes (Figures 1, 2) are not tightly connected. The discussion can be strengthened by making reference to the 4 enhancer types, distinct molecular features regarding to the enhancer types in relation to Pou5f3- and/or Sox19b-associated changes for H3K27ac deposition, and how regulation of these transcripts connect with the observed developmental defects and changes in gene expression (figures 5, 6). Parts of Supplemental Notes can be moved here?

- What do the authors define as "first and second periods of zygotic transcription"?

- The usage of "select rather than activate zygotic genes" is vague. The authors can better describe the features of genes that are selected and bound by Pou5f3 and Sox19b. We find this is not well-described in Fig. 6 nor conveyed in the Discussion.

Reviewer #3:

Remarks to the Author:

In this manuscript, Gao et al. examine the relationship between two reprogramming pioneer factors Pou5f3 and Sox19b in driving early gene expression profiles and phenotypic outcomes during zebrafish development. This is a topic that is likely to be interesting to the broad audience of Nature Communications as orthologs of these factors drive reprogramming events in mammalian tissue culture systems. Unfortunately, because of the abbreviated format and lack of clarity in the writing, it is challenging to assess the experimental support for the mechanistic conclusions in the manuscript.

In many cases, justification for focusing on a specific subset of genomic loci or developmental timepoint is not clear. Some examples are provided:

1. How are the PSN loci defined? (They were originally defined in a prior manuscript from the Onichtchouk laboratory, but for this manuscript must be redefined.) Furthermore, some explanation for the particular focus on this subset of loci must be included. Sites individually bound by Pou53 would be interesting to assess. If there is evidence that regions bound by individual factors are not biologically relevant this should be mentioned explicitly when justifying the selection of the PSN sites for analysis.
2. Why are the various experiments performed at different developmental time points? (MNase at 4.3 hpf, H3K27ac at late blastula). This is particularly confusing given the model based on phenotypes put forward in Figure 2f-h where the ratio of the two proteins is suggested to be important earlier than either of these two timepoints to determine developmental timing.
3. What were the criteria for "top genes" for each differential enhancer noted on page 7?

Clarity in writing is also important for data interpretation at many other places in the manuscript. Some examples are provided:

1. What is non-consensus binding? This is not clearly defined on page 5.
2. How are "open" and "closed" regions defined in the MNase-seq experiment?
3. The clustering of the H3K27ac peaks into five groups should be better defined as these are the basis for future analysis. Furthermore, the classes should not be classified as "active" versus "repressed" since the classification is based on acetylation and not activity.

The deletion in Sox19b must be confirmed to be a null allele. It is possible that a downstream start codon could produce a truncated product that acts as dominant negative/neomorph and that this causes a more severe phenotype than the null and that this activity might be dependent on Pou5f3 function. This would explain the phenotypic suppression in the double mutant.

The title for Figure 3 doesn't fit with what is shown. It mentions independent binding by Sox19b and Pou5f3, but all sites analyzed are PSN sites so they both should bind all sites analyzed. Instead, this figure analyzes chromatin accessibility and the motifs enriched under these regions.

What evidence is there that Pou5f3 directly affects H3K27ac? The argument on page 6, "Hence, Pou5f3 binding to type 1 and type 2 enhancers displaced nucleosomes and directly induced local H3K27ac" does not seem to be clearly supported from the correlative data. Couldn't Pou5f3-mediated accessibility increase binding of another factor or factors that then recruits the acetyltransferase? Have motif searches been done for these different enhancers? In addition, has the timing of chromatin accessibility and H3K27ac establishment been tested? What data support the fact that chromatin accessibility precedes acetylation as shown in Figure 7a?

On page 7-8 in discussing the time resolved RNA-seq, the authors try to deconvolute whether the changes in gene expression are causing the developmental delay or are due to the developmental delay, but this is not clear as written. Thus, concerns remain as to where the changes in gene expression are due to the developmental delay or causing it.

The model in Figure 7a is confusing in so far as it introduces P300, which is never mentioned in the text at all. In Figure 7d, why are there individual Pou5f3 and Sox19b-bound loci since all the analysis in the paper was focused on regions where both factors bind?

While there is already a ton of data analysis in the manuscript, MNase-seq and/or H3K27ac on the double mutant would be very informative for the model put forward.

Reviewer #4:

Remarks to the Author:

In this study, Gao et al. analyzed Pou5f3 and Sox19b mutants to understand the functional differences and interplay between the two TFs in early zebrafish development. The study revealed that Pou5f3 and Sox19b function differently on nucleosome displacement and have distinct regulatory roles in gastrulation and organogenesis. Furthermore, the authors suggested that the balance of the two TFs is important for ZGA timing, which is a surprising result. The conclusion is important for readers in the field of developmental biology, epigenetics, and reprogramming. However, the paper lacks several crucial data to support their claim, which are required before publication.

Major

1. The reviewer is not well convinced with their conclusion that the balance of Pou5f3 and Sox19b determine the timing of ZGA. In Figure 5c d, the difference between MZspg and MZsox19bspg is not obvious. Using the RNA-seq data, the expression of all zygotic-only genes (excluding those that do not have maternal mRNAs) should be shown as heatmap so that readers can easily evaluate the delay of ZGA at genome-wide level in mutant embryos.

Also, it is unclear whether the balance of Pou5f3 and Sox19b regulates the timing of ZGA or the expression level of some genes. With Figure 5e, one would speculate that the expression timing is not changed but absolute expressions are reduced in single mutants. It would be more informative to analyze using PolII ChIP-seq for precise evaluation of ZGA timing, since RNA-seq of early embryos have both maternal and zygotic transcripts.

2. The prediction tool of nucleosome positioning used in this study was established based on yeast data (Kaplan et al). However, this prediction method is not suitable for vertebrate genomes, and in fact, it has been shown in zebrafish that predicted nucleosome organization shows very weak correlation (Zhang et al., 2014 Genome Res). Also the Kaplan rule does not hold true in hypomethylated regions of the medaka-fish genome (Nakamura et al., 2017 Epigenetics & Chromatin). The authors should show the validity of using this tool in zebrafish by genome-wide comparison of prediction and MNase-seq data. Alternatively, it would be more informative to show nucleosome occupancy using MNase-seq data of pre-ZGA embryos.

3. Figure 3c, d; The effect of sox19b at closed sites seems quite small. The subtraction (Fig 3d) shows no significant differences, and therefore, it is difficult to conclude that Sox19b affects on both open and closed regions.

4. Figure 4b; It is not clear whether nucleosome displacement or binding itself directly or indirectly induced H3K27ac. This needs to be discussed.

5. It is also not clear whether changes of H3K27ac levels in enhancer type 3 and 4 is direct or indirect. For this, the authors need to show Pou5f3 and Sox19b ChIP binding as heatmap in Figure 4b.

6. Overall, the story of this manuscript seems complicated or not straightforward. Genes activated at ZGA are mostly housekeeping genes and some developmental genes required for gastrulation and later organogenesis. However, most of the analyses only focused on developmental genes.

The first story on the role of Sox19b and Pou5f3 in displacement of nucleosomes deals with all genes, but the rest of the experiments focused on developmental genes related with the mutant phenotypes, demonstrating that Sox19b and Pou5f3 differentially regulate a subset of these developmental genes.

How do Sox19b and Pou5f3 exhibit distinct roles during early and later embryogenesis? It is not clear how the first story is related with other late stories and how the two factors works differently.

To address this, binding targets of each factors, and their overlaps, need to be clarified by re-analyzing of published ChIP-seq data.

Minor

6. There are two Figure 2b, one should be 2e.

Dear reviewers! We highly appreciate the helpful comments of all four of you, and hope that we were able to address all your concerns.

In response to the reviewer requests we made the following major changes in the parts of the manuscript: To improve the clarity of the presentation, we changed the order of the manuscript's sections. Namely, we first showed the embryonic phenotypes (Fig.1, Fig.2), and then RNA-seq time curves analysis (Fig.3, Fig.4), MNase-seq (Fig.5), H3K27ac and enhancer analysis (Fig.6). In Fig.7, we connected the enhancers and transcription. Fig.8 illustrates the extended discussion.

Mutant phenotypes: we provided statistics of the developmental delay in MZ*sox19b* mutants and performed Morpholino titration and rescue experiments.

RNA-seq time curve analysis:

- a) To better explain the principles of RNA-sense tool, we provide the supplementary Movie 2, where we did our best to explain how it works. We hope that people will use RNA-sense.
- b) We visualized our results as a heatmap and included statistical analysis, which supports our conclusions regarding the compensatory effects of Pou5f3 and Sox19b to the bulk transcription during the major ZGA wave onset.
- c) We demonstrate that at the absence of both TFs, the transcription ratio of *BMPs* to *chordin* is biased from the beginning of their zygotic expression. This explains the phenotypic dorsalization of the double mutant.

MNase and H3K27ac:

- a) This part was criticized by all the reviewers. We made a completely new and easier analysis of only the regions around Pou5f3 and SoxB1 ChIP-seq peaks. We recovered the former differential enhancer clusters 1,2 and 3 and characterized their properties in a more systematic way than we did before. We connected all enhancers with transcription curves and evaluated statistical differences in transcription driven by each enhancer class.

We restructured the whole manuscript, therefore highlighting all changes in the text file will make it unreadable. We highlight the changes where possible (see the Gao_et_al_color_marked file). In the other cases we refer to the former and current figure, or indicate the chapter. The point-by-point answers follow below.

---	---
REVIEWER COMMENTS	AUTHORS RESPONSE
Reviewer #1 (Remarks to the Author): The manuscript by Gao et al addresses the independent roles and interplay between two key pluripotency factors, Pou5f3 and Sox19b during development. This is an important question, with relevance for understanding both zygotic genome activation and the pathways regulating early embryogenesis. The authors uncover a rather complicated relationship, which they do a good job of explaining at a macroscopic level. At a more granular level, there are patches of the manuscript (especially toward the end of the results) that are a bit more challenging to follow. These concerns are somewhat alleviated by thoughtfully prepared figures. Data and software availability is presented in a very clear and accessible way. The methods provide very good detail for reproducibility, with a few exceptions.	
Most notably, this reviewer was unable to find the Key resources table cited in the methods, which presumably would have contained antibody information, morpholino sequences ect.	Our apologies! All the requested information is now in the Data Resources table (published and generated during this work) at the end of the Methods section
Figure 1:	
1b: The author's sox19b mutant is an indel. Recently, there has been significant data suggesting that nonsense mediated decay of transcripts from these types of mutations can trigger poorly understood compensation mechanisms (see for example El Brolosy et al 2019). Ideally, transcript analysis of	We present the transcript analysis in the supplementary data (new Fig.S3a) and comment it in the first chapter of the results (p5, yellow-marked text): "The TALEN-induced sox19b mutation resulted in premature stop codon before the first intron

sox19b mutant should be presented in supplemental data to clarify whether RNA in this mutant is subject to nonsense mediated decay, and the authors should comment on this issue in the text.	of sox19b; nonsense mediated mRNA decay in this type of mutants can trigger compensatory response by upregulation of the genes that exhibit sequence similarity with the mutated gene's mRNA³¹. To investigate if the transcription of SoxB1 genes is changed in the MZsox19b mutant, we quantified the levels of sox19b, sox19a, sox3 and sox2 by RNA-seq(Fig. S3a). sox19b maternal message was reduced 15-fold already before MBT, indicating that nonsense mediated decay takes place. Although sox19a, sox3 and sox2 bear the closest sequence similarity to sox19b, we did not detect compensatory upregulation of these genes which were rather delayed in MZsox19b (Fig.S3a-c)"
Fig 1b /S1: While the imaging approach is clear, the way the authors are calculating the rate of cell division is not fully clear from the methods section. It seems the method used in Fig 1b and S1a are different. Ideally, the 10th cell cycle should be included in fig S1a with statistics indicating significance. In addition, some quantification (of cell size/number of nuclei?) should accompany the image in figure 1b. At present, cycle 10 appears to be the time point when the authors suggest lengthening of the cell cycle occurs, yet it seems to be the only time point lacking quantification.	We are grateful for the reviewer 1 for raising this and the next point! We performed in-vitro fertilization, optimized staining, and scored the early cell cycle lengths from 5 to 10 cycle using large numbers of embryos. (see new Fig.1b, new Fig.S1, legend and the methods). We could not confirm a delay at 10th cell cycle in two experiments, and we removed the statement about the early delay.
Figure 1c: The images in figure 1c are of remarkable quality. However, they represent only a single embryo. Two additional embryos are shown in the supplementary movie. Because developmental delay is a trickier phenotype to assess, Ideally, I'd like some more detail here. How many embryos were examined, and was the extent of delay comparable between all embryos? Also, what was the method of generating embryos? Given these are maternal zygotics, I'm assuming these were generated by comparing wtXwt crosses to crosses between homozygous mutants. If this is the case, were embryos from multiple independent crosses compared, and if so, how many?	In response to this comment, we provided the missing information in the beginning of the Methods (yellow marked text) : "Embryos were obtained by natural crossing (4 males and 4 females in 1.7 l breeding tanks, Techniplast) or by in-vitro fertilization, as indicated. Wild-type and mutant embryos from natural crosses were collected in parallel in 10-15 minute intervals and raised in egg water at 28.5°C until the desired stage." At the new Figure S2, we provide group pictures for one parallel cross, and statistics for the other 7 parallel independent crosses, where we compared wtXwt progeny from the parents of different age, with MZsox19bXMZsox19b progeny from parents also from different age and from two subsequent generations of the mutants. In each experiment, we scored the phenotypic delay at three stages: dome, shield and bud; embryo numbers are provided. As one can see from the data, the delay was reproduced 7 times out of 7 for shield and bud, but only 4/7 times for dome, which is a blastula stage. Considering these results, we removed all strong statements about the exact timing of the delay from the main text. We made new single embryo pictures for Fig.1 from the 9th independent cross. Delay timing at the main Fig.1d was corrected accordingly. Finally, we observed and quantified developmental delays in 10th and 11th independent crosses at post-gastrulation stage, for the non-injected WT and MZsox19b embryos which were used as controls for the rescue experiment (the answer to the next comment, Fig. S3g, left part of the graphs).

Figure 2: Figure 2b: How can the authors be sure that their Morpholino deletion of SoxB1 is a complete knockdown of the proteins? If it were to be a partial knockdown, could that impact interpretation?	For SoxB1 QKD in the wild-type: we are sure that it is NOT a complete knockdown. Maternal Sox19b is present in WT-QKD at 1K-cell stage, and it is visible in the Western blot (we published it at 2013 (ref. 10). We now included this information into the introduction, to make it clear why we made the MZSox19b mutant: we wanted to get rid of maternal Sox19b protein. MZSox19b – TKD is most likely a complete knockdown of SoxB1, at least at the early stages we are interested in: Sox19b is absent because of the mutation, Sox2, Sox3, and Sox19a are virtually absent at the beginning of ZGA, because they are zygotic transcripts (see new Fig. S3a). If the sum of maternal Sox19b and zygotic Sox2, Sox3, and Sox19a were critical for gastrulation, we would have seen the morphological differences between WT-QKD and MZsox19b – TKD (we actually expected to see it). But we did not see such a difference (Fig.2b), hence, Sox2, 3 and 19a become critical only at the end of gastrulation. Non-complete inhibition of Sox2,3 and 19b would not affect this conclusion. Okuda et al, 2010, who designed the QKD morpholinos, have shown that inhibition of Sox2, Sox3, and Sox19a by QKD morpholinos at bud stage, where the concentration of these proteins is much higher than after ZGA, is about 90%; while Sox19b was barely detectable at this stage.
There is also some concern re morpholino toxicity causing delay in these experiments. St. control morpholinos tend to provide poor controls, as all morpholinos behave differently. Rescue in supplemental data seems partial, and numbers aren't provided. Caveats for this experiment should be more explicitly stated, and controls clearly articulated.	In response to these criticisms, we changed the St control morpholino to TKDco (TKD-matched control) and performed two experiments to titrate the morpholinos and estimate their toxicity (New Fig. S3d-f, described in the figure legend and in the chapter “SoxB1 Morpholino knockdown” in Methods, marked yellow). We then performed another two new rescue experiments, more carefully controlling for the morpholino effects: The representative phenotypes for one of them are shown at Fig. 2c, statistics for both experiments is presented at the new figure S3g, and it shows that the rescue is complete. We discuss the experiment and Morpholino issues in the respective section “MZsox19b-TKD rescue experiments” of the Methods (marked yellow): .
The authors write: “We concluded that maternal and zygotic functions of SoxB1 proteins are uncoupled: zygotic SoxB1 activity becomes critical starting from the end of gastrulation, maternal Sox19b protein sets the timing of gastrulation”—I’m somewhat uncomfortable with this statement. Many mutants cause developmental delay, often due to indirect effects. While it is clear that sox19b loss impacts the timing of gastrulation, I’m not sure that the data presented supports the idea that maternal sox19 proteins set the time of gastrulation. Perhaps more cautions wording could be applied here.	We agree (at this point, we have not shown yet that maternal Sox19b has any function). We removed the second part of the phrase. We write in the first page of the results (marked yellow). “We concluded that combined zygotic activity of Sox2/3/19a/19b proteins becomes critical for the embryo starting from the end of gastrulation. The developmental delay in MZsox19b may reflect non-essential earlier requirement for maternal Sox19b. Sox19b may act redundantly with Nanog or Pou5f3, during major ZGA onset, as suggested previously9”....

Fig 2c: In addition to providing the reference, the authors should comment on whether spg is thought to be null allele in the text, as this is important for interpretation of their data.	We did. In the results section (marked yellow): “To investigate the early requirements for maternal Sox19b and Pou5f3, we obtained a double mutant MZsox19bsp_g by crossing MZsox19b to Pou5f3 null-mutant MZspg m793 26.”.In the Methods: “ Spg^{m793} allele carries an A->G point mutation in the splice acceptor site of the first intron of Pou5f3 gene, which results in the frameshift starting at the beginning of the second exon, prior to the DNA-binding domain. Spg^{m793} is considered to be null allele. “
Fig2. In figure 2 it would be helpful if more extensive comparison of the MZ sox19b, MZspg and double MZ mutants was presented in the main figure. In particular, 2 c and 2d would be easier to understand if the single MZ mutants were included. Number of embryos with phenotype/ number assessed should be indicated for all panels, as well as panels in s1 e and f.	We included single MZ mutants (Fig.2d), and numbers of embryos to this and all other panels of the figure 2.
Figure 3. The authors write : In sum, we found that Sox19b and Pou5f3 act as independent pioneer factors during maternal-to-zygotic transition. They displace nucleosomes by binding to different consensus motifs. Non-consensus binding of two factors results in the opposing effects of local chromatin accessibility genome-wide. However, as best this reviewer can see, binding was not assessed in these experiments, only nucleosome displacement. If Pou5f and sox19b Chip data is incorporated here, it needs to be more clearly stated. If this data only assesses nucleosome displacement the more careful language should be used to delineate what is shown vs inferred.	Figure 3 is now Figure 5. We agree with the comment. We re-mapped ChIP-seq for Pou5f3 and SoxB1(ref. 10) to danrer11 assembly, and took single (P,S) and double (SP) peaks for analysis. We show TF binding and nucleosome displacement back to back (heatmaps Fig.5a-d). We see two effects: nucleosome displacement on the ChIP-seq peaks, (5g,h), and nucleosome displacement all over genome, which is not sequence-specific (5i,j). We can now conclude that the first effect is due to TF specific binding. And we cannot exclude that the second effect is due to the widespread non-specific binding of the same TFs. We summarize it at the end of the corresponding results section: “We propose, although have not proven, that opposing genome-wide changes of nucleosome landscape in MZsox19b and MZspg may be due to the widespread non-consensus binding of the TFs, which are abundantly present at 4.3 hpf. In summary, we found that Sox19b and Pou5f3 act as independent pioneer factors even on co-occupied sites. They displace nucleosomes by binding to different consensus motifs (Pou5f3 acts on pou and pou:sox, Sox19b on sox motifs)”
The methods suggest this analysis and other genome wide was performed on Zv9 assembly, which is quite old. Is this correct? Although it likely will not affect their conclusions, ideally analysis would have been performed using the Zv11 assembly which has been available for quite a while.	We remapped all our data to the most recent assembly (and this did not affect our conclusions).
Figure 4 The methods seem to suggest that for the following libraries there was only replicate: WT K27ac Chip, MZspg K27ac Chip, WT K4me3 Chip, MZspg K4me3 Chip, MZsox19b K4me3 Chip. And then two replicates for MZsox19b. Can the authors clarify? If only one replicate was actually performed, can the authors please provide the rational/justification for the lack of replicates, as it seems insufficient.	This is correct. We could not get enough ChIP-seq material for the library preparation, so we had to pool two good ChIP-seq experiments from the WT and for MZspg together. We explain this in the methods now (marked yellow p.35): “In order to convert a small amount of DNA into indexed libraries for Next Generation Sequencing (NGS) on the Illumina platform, we used the

	NEBNext® Ultra™ DNA Library Prep Kit. As the DNA outcome of individual WT K27ac ChIP and MZspg K27ac ChIP experiments did not reach the input DNA limit for this kit (5 ng), we pooled together the material from two successful ChIP experiments, as well as corresponding inputs, for the library preparation in these genotypes. Two libraries were prepared from two single MZsox19b K27ac ChIP experiments." To provide the justification why we are using single replicates, we estimated the variance between two biological replicates of the same genotype (MZsox19b) versus the variance between the different genotypes, and show using two methods, that the variance between the replicates is negligible compared with the variance between the genotypes (Figure S8 a-b, yellow marked text at p35, Methods).
Assuming these are on pooled embryos, how many? From an individual pairwise cross or multiple pairwise matings?	We pooled the material at several steps of the protocol, and this is now included in the Methods, sub-section Chromatin immunoprecipitation (ChIP) for histone marks (yellow marked text p.33-34): "The embryos were obtained from natural crossings in mass-crossing cages (4 males + 4 females). 5-10 cages were set up per genotype, the eggs from different cages were pooled. The freshly laid eggs of MZsox19b, MZspg mutants and wild-type were collected in 10-15 min intervals. Unfertilized eggs were removed at 2-4 cell stage.<>. The nuclei were frozen in liquid nitrogen and stored at -80 °C. The nuclei collected in different days were pooled together to reach the total number of 2.5-3 million, which was used to start one ChIP experiment". To clarify: Collection of sufficient material required at least 3-4 days, followed by ChIP-seq and quality control (sub-section ChIP quality control and library preparation for histone mark ChIP-seq). Pooling the ChIPs for MZspg and the WT occurred after these stages (see the previous comment).
Figure 5 Here, and throughout the manuscript, the authors seem to refer to ZGA as occurring as burst at the 1000 cell stage, whereas current data (see review by Vastenhouw et al 2019) suggests this is a more gradual process that initiates around the 128 cell stage. The authors may want to revise their writing to better reflect this current view.	Figure 5 is Figure 3 now. We revised our writing and introduced "the major wave of ZGA" and the term "MBT" throughout the manuscript to better reflect this view.
For RNA seq, the authors write "In total, we sequenced 78 samples in four biological replicates (Fig.5a) and 23 technical replicates for WT and single mutants. However, it is difficult to parse out how many replicates of what, and which were biological vs technical from this statement. Authors should be clearer on this point. Assuming these are on pooled embryos, how many? From an individual pairwise cross or multiple pairwise matings?	We clarified this issue in the Methods, yellow marked text at p.36-37, sub-section "RNA-seq time curves: material collection, processing and sequencing. The freshly laid eggs were obtained from natural crosses in mass-crossing cages (4 males + 4 females). 5-10 cages were set up per genotype and eggs from different cages were pooled. At least 600 freshly laid eggs per each genotype collected within 10-15 minutes were taken for a single experiment. To

	match the developmental curves as precisely as possible, the material was simultaneously collected for two genotypes in parallel. In each of five experimental days, the material was collected for two genotypes, as specified below: MZsox19b-rep1 and MZspg-rep1, WT-rep1 and MZspg-rep2, WT-rep2 and MZsox19b-rep2, WT-rep3 and MZspg-rep2, WT-rep1 and MZsox19spg-rep1, WT-rep4 and MZsox19spg-rep2. 45-60 minutes after the egg collection, we ensured that the embryos of both genotypes are at 2-4 cell stage, removed non-fertilized eggs and distributed the embryos to 8 dishes per genotype, 40-45 embryos per dish, to obtain 8 time points per genotype. The temperature was kept at 28.5°C throughout the experiment. The embryos from one dish per genotype were snap-frozen in liquid nitrogen every 30 minutes, starting from 2.5 hpf (pre-ZGA, 256 cell stage, 8th cell cycle) till midgastrula (6 hpf). In total, we collected 4 biological replicates of the time curve for the wild-type, and two biological replicates for each of MZsox19b, MZspg and MZsox19bspg mutants<> To estimate the non-biological variation between the samples, we prepared and sequenced two technical libraries for each time point in the following time curves WT-rep1, MZsox19b-rep1 and MZspg-rep1 “ The next Methods sub-section, RNA-seq time curves: data processing and visualization, explains how we treated the replicates for analysis.
Figure 5 is by far the most complicated figure. Stylistically, reducing reliance on abbreviations w, s, p and d and instead writing these out (where possible) would likely reduce the mental effort required of the reader when assessing this figure.	Figure 5 is now Fig.3. The criticism about this figure was coming from all the reviewers and was well taken. We moved the explanations of RNA-sense procedure to the Movie 2 (we tried our best to explain it, as we hope that people will use it). Technical details of our post-RNA-sense analysis (correlation plot and Venn diagrams) are now shown in the supplementary Fig.S4. Main Fig.3 shows the zygotic genes as a heatmap, which we feel is easier to understand, with full annotations. We also found a simple way to compare gene expression at the same scale, which is relative expression values (log2 FC to 2.5 hpf for all the subsequent timepoints), which allowed us to visualize the time curve analysis better. We hope it improved the presentation.
Discussion: The authors do a nice job of summarizing their conclusions, however the analysis they perform is complicated. Additional discussion of the assumptions made in this study and caveats for interpretation would be welcome (see some suggestions in comments above).	This helped! We expanded the discussion.
Reviewer #2 (Remarks to the Author): In "Pluripotency factors select gene expression repertoire at Zygotic Genome Activation", by Gao et al investigate the contributions of two transcription factors, Pou5f3 and Sox19b, in zygotic genome activation in zebrafish. This is an interesting study that certainly has the potential to significantly further our	

understanding of the genome activators, Sox19b and Pou5f3, in zebrafish. However, the current manuscript is very hard to follow and it is therefore difficult to assess whether the data supports the conclusions. We therefore think it would be premature to publish this work. We suggest that the authors clarify their manuscript to help the reader reach the conclusions and interpretations drawn. We would be happy to review the work again when this has been done. Below we have listed some of our concerns and suggestions that will hopefully help the authors to improve their manuscript. SPECIFIC COMMENTS	
Introduction (page 3)  • End of paragraph 1: the authors should include the following citation for Dux in mouse and human: Hendrickson et al. 2017, doi: 10.1038/ng.3844 	Done. it is ref.8 now
 • Pou5f3 and Sox19b are introduced as TFs with pioneer activity "In case of several genome activators, like Pou5f3, Sox19b, and Nanog in zebrafish, it remains an open question, how their broad pioneering activity at ZGA relates to their different functions later in development". The authors do not present any evidence that support the claim that Pou5f3, Sox19b, and Nanog are established pioneer factors. Although genomics data correlate nucleosome displacement with their binding and suggest that the zebrafish homologs can behave like other genome activators or like their mammalian counterparts, this is a model and there is no biochemical data supporting the pioneer factor feature (direct binding to nucleosome). Please clarify this in the text and provide references. 	Done (p3, the text marked blue): "Nucleosome positioning plays a dominant role in regulating genome access by TFs. The widespread action of genome activators is thought to result from their ability to act as pioneer factors, first displacing nucleosomes so that other TFs can bind¹². Indeed, reduction or loss of genome-activating TFs in Drosophila, zebrafish and Xenopus resulted in the decreased chromatin accessibility on their binding sites^{11,13-16}. Out of them, direct pioneer binding to nucleosomes was demonstrated that far only for Drosophila genome activator Zelda¹⁷. The mechanisms underlying nucleosome-displacing activity of zebrafish and Xenopus activators are less clear: they may bind to nucleosomes similarly to their mammalian homologs¹⁸, or compete with nucleosomes for DNA binding¹⁹, or both"
The authors vaguely mention the studies on the "four different scenarios" for the mechanism of interaction between Pou5f1 and Sox2 with one another and with chromatin. Please elaborate on this to provide a more molecular context that help to nuance between the models and set up the problem for how the current study will contribute to our mechanistic understanding of Pou5f3 and Sox19b interaction in genome activation.	Done. (p3, blue-marked text)" Until recently, POU5F1 and SOX2 were thought to act cooperatively, binding as heterodimers to bipartite pou:sox cognate motifs²¹. This view was challenged by Soufi et al.(2015), who demonstrated that POU5F1 and SOX2 target distinct motifs on the nucleosome-wrapped DNA¹⁸, and by four studies that suggested different scenarios of how POU5F1 and SOX2 interact with each other and with chromatin in embryonic stem (ES) cells. These scenarios are: 1) assisted loading, whereby SOX2 first engages the target DNA, then assists the binding of POU5f1²²; 2) negative reciprocity, where POU5F1 and SOX2 sometimes help and sometimes hinder each other in binding to the genome²³; 3) conditional cooperativity of POU5f1 and SOX2 binding, depending on the motif positions in the nucleosomal context²⁴ and 4) independent binding, even at co-occupied sites²⁵"
Figure 1 and related  • Fig. 1a: The authors begin this figure with a time point at which the MZsox19b embryo is already developmentally delayed (15-min). As a reference point, please start with 	Done, see the new Figure 1 .

images showing no detectable delay between the 2 genotypes (for example, a stage between the 6-9th cell cycle).	
 • Fig. 1: There is no mention (in the legend or Methods section) of how many embryos were used in this figure, nor the number of experiments these results represent. • Fig. S1a: why is the 10th cell cycle quantification not included? 	We are grateful for the reviewer for raising these points! We provide the new data, summarized in the figures Fig.S1 and S2. We performed in-vitro fertilization, optimized the staining, and scored the early cell cycle lengths from 5 to 10 cycle using large numbers of embryos. (see new Fig.S1, legend and the methods). We could not confirm a delay at 10th cell cycle in two experiments, so we removed the statement about the early delay. S2: we addressed the statistics of the delay (we had seen the delay in 6 generations of the mutants and made several movies, but we had never quantified it previously just because the delay was so obvious). We provide group pictures for one parallel cross and statistics for the other 7 parallel independent crosses, where we compared wtXwt progeny from the parents of different age, with MZsox19bXMZsox19b progeny from parents also from different age and from two subsequent generations of the mutants. In each experiment, we scored the phenotypic delay at three stages: dome, shield and bud; embryo numbers are provided. As you can see from the data (Fig.S2), the delay was reproduced 7 times out of 7 for shield and bud, but only 4/7 times for dome, which is a blastula stage. Considering these results, we removed all strong statements about the exact timing of the delay from the main text. We made new single-embryo pictures for Fig.1 from the 9th independent cross. Delay timing at the main Fig.1d was corrected according to the average. Finally, we observed and quantified developmental delays in 10th and 11th independent crosses at post-gastrulation stage, for the non-injected WT and MZsox19b embryos which were used as controls for the rescue experiment (new Fig. S3g, left part of the graphs)
Figure 2 and related  • Fig. 2a: First, the use of 8 hpf is questionable, as Fig.1 had no 8 hpf. Presumably because all 4 SoxB1 members are expressed by this stage. In fact, they are already present by 30% epiboly (near 5hpf). Second, it is clear that the RNA expression patterns for sox2, sox3, and sox19a are already different between the two genotypes because of the time delay in MZsox19b. Although the authors want to make the point that the other Sox members are still present in MZsox19b, it is important to check that the expression pattern is also intact. For this, please include in situ data that match with earlier wild-type time point. 	We included the data for 6 hpf as Fig.2a, (the staining is ubiquitous) and the expression time curves for all 4 SoxB1 members from 2.5 to 6 hpf in the WT and MZsox19b as Fig.S3a. We discuss it at the main text (yellow marked text on p.5): “To investigate if the transcription of SoxB1 genes is changed in the MZsox19b mutant, we quantified the levels of sox19b, sox19a, sox3 and sox2 by RNA-seq (Fig. S3a). sox19b maternal message was reduced 15-fold already before MBT, indicating that nonsense mediated decay takes place. Although sox19a, sox3 and sox2 bear the closest sequence similarity to sox19b, we did not detect compensatory upregulation of these genes which were rather delayed in MZsox19b (Fig.S3a-c)”.
Fig. 2b: We are puzzled by the results shown here and in Fig. S1D to make the point of the specific maternal and zygotic functions of SoxB1 proteins. - The rescue experiment (via injection of sox19b mRNA)	Done. We performed and quantified two new rescue experiments. One of them is shown in s Fig.2c, quantification for both experiments is shown in Fig.S3g.

should be included in the main figure.	
- We agree that with the conclusion that "zygotic SoxB1 activity (although please specify Sox2/3/19a)....	Patterned zygotic expression of Sox19b in the nervous system starts at gastrulation, from 75% epiboly on: (https://zfin.org/ZDB-IMAGE-060307-16, https://zfin.org/ZDB-IMAGE-060307-17, images from Okuda et al, 2006); therefore we used "zygotic Sox2/3/19a/19b", all four are expressed zygotically.
- We agree that with the conclusion that "zygotic SoxB1 activity (although please specify Sox2/3/19a) becomes critical starting from the end of gastrulation" based on the panels WT QKD and MZsox19b TKD, where in the absence of Sox2, Sox3, Sox19a protein, post-gastrulation defects are obvious. - However, we do not agree with the authors' conclusion that "maternal Sox19b protein sets the timing of gastrulation" (page 4). If this were the case, one would expect that the conditions in which there are still maternal Sox19b protein present (WT QKD, MZsox19b TKD + sox19b mRNA), the timing of gastrulation would be restored and the phenotype displayed at 10 hpf should be the same as WT. As they do not match, and there is still a delay, then the available early expressed Sox19b protein cannot correct for the timing of gastrulation. [unless it is not sufficient, or proper dosage not reached, but this is not a point the authors make].	The criticism is well taken – at this point, we cannot state that the Sox19b has a maternal function, because Sox19b injection at 1-cc does not rescue the timing of gastrulation. We removed the conclusion "maternal Sox19b protein sets the timing of gastrulation" from the text. The end of the first sub-section of results (p5, yellow-marked text) states now: "We concluded that combined zygotic activity of Sox2/3/19a/19b proteins becomes critical for the embryo starting from the end of gastrulation. The developmental delay in MZsox19b may reflect non-essential earlier requirement for maternal Sox19b. Sox19b may act redundantly with Nanog or Pou5f3, during major ZGA onset, as suggested previously9".
Fig. 2c-e (where 2e is erroneously called 2b): We do not think there is sufficient evidence from 2cde to draw models about the distinct functions of Pou5f3 and Sox19b. The ratio of Sox19b and Pou5f3 is brought into consideration, yet only maternal mutants were studied and to bring quantitation into the model, the authors would need to play with the levels of individual TFs (as RNA or protein contribution). The models in 2f-h do not fit here as no evidence has been presented yet to support them. In addition, Fig. 7 has a similar schematic diagram already	We agree. We 1) removed the scheme with "Ratio" at ZGA. 2) We moved Chordin Morpholino rescue experiment from figure 6 to figure 2g, so that we can make a conclusion that maternal Pou5f3 and Sox19b are important for D/V patterning here.
The model proposed for 2g is at odds with Fig. 6d. See below***. - ***The authors draw a conclusion that is at odds with what was proposed in Fig. 2g for the functions of Pou5f3 and Sox19b in D/V patterning: Fig. 6d: "the synergistic activity of Pou5f3 and Sox19b at ZGA controls D/V axis formation" Fig. 2g "D/V patterning depends on either Pou5f3 or Sox19b to be maternally present".	Thank you for noticing this! Now we feel that both formulations are equally bad. We changed it the following way in relation to the phenotypes shown at Fig.2: (blue marked text at p.6). "Reduction of Chordin levels by morpholinos was sufficient to rescue Msox19bsp phenotype (Fig 2g). This result suggests combinatorial action of maternal Pou5f3 and Sox19b,.e. two TFs act additively inducing ventral regulators, and/or repressing dorsal regulators. The combined changes override the self-regulatory capacities of dorso-ventral gene network, while changes in single mutants can be buffered". The molecular support for this statement is shown in a new panel Fig.3g, which demonstrates that the ratio of chordin to bmp2b and bmp7a transcripts is skewed in the double mutant starting from ZGA onset, and that it is not the case in both single mutants.
Figure 3 (and related)  • End of page 4-start of page 5: The choice of 4.3 hpf is not rationalized, and considering how Figure 1 and 2 examined various time points, and looked at the contributions of each factor in the both gastrulation and post-gastrulation events, unclear. 	Fig.3 is now Fig. 5 (we put all genomic assays after RNA time curves analysis). We rationalized the choice of 4.3 hpf at the beginning of the results chapter "Sox19b and Pou5f3 act as independent pioneer factors" as follows (blue marked text at the page 8-9 : " In order to connect the effects of Pou5f3 and Sox19b on transcription with their

	effects on chromatin, we chose 4.3 hpf time point in the middle of transcription time curve, before the end of MZT, to make a snapshot of chromatin state in the single mutants. Choosing 4.3 hpf time point enabled us to profile chromatin marks, which are quite low in the earlier stages.
 • Page 5: about the statement "SoxB1 proteins can recognize two types of cognate sequences: bipartite pou:sox motifs, also bound by Pou5f3, and sox motif" we note that  - As it reads, the same could also be said about Pou5f3: Pou5f3 can also recognize two types of cognate sequences: bipartite pou:sox motifs and pou motif. - Pou-only motifs are not discussed at all here. We looked at Veil et al. and all of the Onichtchouk lab previous papers, and there is never any mention of the Pou motif alone. - So we wonder, are there no Pou motifs that exist (or enriched) independently of Sox according to the authors? Please clarify. 	Thank you for the suggestion! It is done (Fig.5e, Table S3 for the motifs in minimal MEME format). Pou-only motifs exist and we included them into MNase and H3K27ac analysis (Fig 5, Fig 6). Pou and pou:sox motifs seem to be directly bound only by Pou5f3, but not by Sox19b. We speculate that Sox facilitates Pou access to these motifs in the discussion.
It would be useful if some genome browser snapshots would be provided that contain ATAC-seq, MNase-seq, and ChIP-seq data, to highlight examples of genes that reflect different binding affinities between Pou5f3 and Sox19b to their motifs.	We provide genome browser snapshot at Fig.5f, which shows ATAC-seq, MNase-seq, and ChIP-seq genomic data and also the density of all three consensus motif matches and non-consensus binding cues, which we define as High Nucleosome Affinity Regions.. With such a high density of random motif matches, which are obviously not bound by TFs (or we cannot see it in ChIP-seq), we think it would be misleading to pick them as examples in our case. In Fig.5 g,h we compared nucleosome displacement in the mutants on ChIP-seq peaks bound by both factors (SP peaks), +/- motifs, and came to the conclusion that even if both TFs are bound at the same region, they displace nucleosomes by acting on different motifs.
Figure 4 (and related)  • Bottom of Page 5: "we immunoprecipitated embryonic chromatin" – please remove "embryonic" 	Fig.4 is now Fig.6. Done (blue marked text at the beginning of p.10).
Fig. 4a, b  - It is not immediately clear how panel a and b compare. Part of the confusion stems from the use of 1-5 already in this panel while in the text this solely refers to enhancer types. 	Fig.4 a,b ->Fig. 6a Previously we looked at H3K27ac peaks at the whole genome, defined the enhancers, and then projected them to ChIP-seq data (this was our former panels 4 a and b). We have re-analysed our data, taking only the regions around ChIP-seq peaks for Pou5f3 and SoxB1. This approach narrowed down and simplified the analysis, and facilitated interpretation of the results. We performed new k-means clustering only around the peaks, and recovered former enhancer clusters 1-3, but not 4 (which was to be expected, as class 4 did not correlate with Pou5f3 and SoxB1 genomic binding). The heatmap is shown at the Fig. 6a.
- If the enhancers in b are a subset of a, it is unclear how the H3K27Ac signal in b (wt, type 1) can be higher than in a.	This is a very valid comment! The heatmap in a (whole genome) contained less highly H3K27-acetylated regions, than the heatmap in b (subset around the ChIP-seq peaks) The automated contrasting of the deeptools heatmaps is very nice, but the visual effects

	pretty much depend on how you sort your data and on the number of genomic regions you take for analysis (the sorting was the same, but the number was different in former 4 a and b). To compensate for it, we now 1) sorted the heatmaps by descending H3K27ac signal (Fig.6a) 2) provide supporting statistics for this heatmap on Fig. 6b, which validates the names of the enhancer groups (by downregulation of H3K27ac mark): co-dependent (CD), Pou5f3-dependent (PD) and Sox19b-dependent (SD).
- The cartoons of embryos in the "nearest genes" column is not helpful.	We removed them, and we list 5 top MF and BP GO: categories instead (Fig.6d)
Fig. 4c - The accompanying text is not correctly described and should be rewritten to match with the more quantitative aspect of the data presented in the figure. "... H3K27ac deposition by each TF depended on the presence of sox or pou:sox motifs in the genomic regions". This reads as a qualitative approach the authors have taken. It should say instead "the presence and number". Please comment on the changes in H3K27ac levels between wild-type and mutant in the main text, with respect to number of TF motifs	the analysis in former Fig.4c was made on all the enhancers around the TF peaks, without considering which cluster they belong to. We replaced it with more detailed statistics, where we looked at the motif dependency by cluster (CD,PD or SD), in two sets of data (H3K27ac change and MNase change in the mutants). This analysis is presented at Fig.6f,g and Fig.S8 i,j,
• Fig. 4d-g and Section "Four types of differential enhancers define four types of transcriptional response to Pou5f3 and Sox19b" - This section needs to be rewritten. It is not written in a linear manner and it does not inclusively describe all four enhancer types even though it sets out that way. Clusters 3&4 are singled out, although the "4/7 top cluster genes ..." is not referred to by the authors in the main text as Cluster 4, so it reads as if the 4/7 refer to the 7/13 from Cluster 3. This is only pointed out in Fig. S4D legend. - Fig. 4d-g and especially Fig. S4 are noisy figures, and individual panels should be referred to in the text rather than as a whole, to attribute the regulatory insights to specific enhancer types. - To look at the differential regulation among the enhancer types, the authors want to relate TF binding, H3K27ac deposition, and transcriptional output (through H3K4me3 as shown in the figure), although the latter approach should be outlined in the main text .	We have split this section into two, The text in the sub-chapter "Pluripotency factors regulate H3K27 acetylation on three enhancer types.", p.10, discusses how nucleosome displacement and H3K27ac change depend on the motifs relates to the whole Fig.6 and Fig. S8. The relationships between enhancers and transcription are now addressed by the following and sub-chapter "PD and CD enhancers are associated with the major ZGA wave", p 10, and illustrated in Fig. 7. And Fig.S9. Here, we linked all expressed transcripts to enhancers (Table S5), and analyse the statistical differences between the groups.
Would it be better to represent Fig. 4d-g, Fig. S4, and Fig. 5a grouped by enhancer types? The reader has to flip through 3 figures to correlate TF binding, H3K27ac deposition, RNA levels.	We now present H3K27ac heatmap at Fig. 6. and all related genomic data (motifs, TF binding, nucleosome displacement, nucleosome prediction) on Fig.S8. The links of enhancers to transcripts is presented at the next figure.
- Table S2: regarding the results from GREAT analysis: for clusters 1,4,5 – terms in the top 5 rankings are listed, but for clusters 2,3 – noticeably other terms that are not related to gastrulation have been left out. Specifically, Cluster 3 shows the top 7 terms with Rank3&4 not included. And Cluster 2 covers a great range of Ranks. Please comment. Consider including a list of the top 20 and highlight the gastrulation terms, rather than cherry-pick.	We list 5 top MF and BP GO: categories for all clusters around TF-binding peaks at the Fig.6d; for Pou5f3- dependent, Codependent (former type 1 and 2) and Sox19b-dependent enhancers (type 3), and for the groups of unchanged enhancers (former type 5)
Page 6: text to fix o "Hence Pou5f3 binding to type 1 and 2 enhancers displaced nucleosomes and directly induced local H3K27 acetylation". Please remove the word "directly". The data is suggestive, but biochemical data is missing to support a direct interaction between Pou5f3 binding and acetylation effects. There could	We agree with all comments. The whole part is re-written, with the respect to comments and the results of the more specific analysis which we have done: we only looked at pou, pou:sox and sox motifs. The text at p.10, marked blue: "We found, as expected, that on PD enhancers Pou5f3 binding to pou:sox

be another TF involved, or accessory proteins to acetylation writers.  o "Sox19b and Pou5f3 can act on type 2 enhancers by either co-binding pou:sox motif or by binding different motifs". The latter is vague. The author only mentions pou:sox and sox motifs, so there is no need for the authors to consider other motifs if they strictly focus on these two motifs. o "Repression... not associated with pioneer activity of repressing TFs ..." we cannot grasp the meaning of this sentence. 	and pou motifs promoted both nucleosome displacement and H3K27 acetylation (Fig. S8i). On SD enhancers, nucleosome displacement and H3K27 acetylation depended on Sox19b binding to sox motifs (Fig. S8j). On codependent (CD) enhancers, Sox19b effects were non-sequence specific (Fig.6f), but Pou5f3 effects depended on pou and pou:sox motifs (Fig.6g). The most parsimonious explanation for this result is that Sox19b non-consensus binding to CD enhancers displaces nucleosomes and facilitates Pou5f3 loading on pou or pou:sox motifs nearby. Pou5f3 acts downstream of Sox19b and promotes local H3K27 acetylation.
Figure 5 (and related)	Fig 5 is now Fig.3 and was substantially redone. We have a technical comment on the data, related to this figure: We have now re-run RNA-sense on biological replicates only (technical replicates were erroneously included before). The conclusions did not change, but the numbers in the Venn diagrams are slightly different.
 • Fig. S6a: It is not clear what the +/- in the matrix mean. 	Instead of the figure S6c we provide supplementary movie 2, where we did our best to explain how RNA-sense works.
Fig. S6c and Table S3: Comparative analysis among the RNAseq data is not described anywhere, and references are missing in the legend. The columns in Table S3 that refer to other studies are not defined (what does No/Yes or numbers refer to in each column in the comparative analysis?)	Fig. S6c is now Fig. S4a. We provide the references in the figure and in the legend. Table S3 is now Table S1 (first sheet), - we changed the headings to achieve clarity.
 • Fig. 5c & Fig. S6d: We find it difficult making the jump from a matrix of a single gene (S6a, which is not full and shows only a few time points) to a matrix of combined genes (5c and S6d). Beneficial to have more matrices from several genes as examples and supporting evidence, so that it makes interpretation of 5c and S6d easier to process. We can't even say we agree with the authors or not, because we go from a single layer (single gene) to multi-layer matrix that incorporates all the RNAseq data. 	We provide the Movie S2 for RNA-sense instead of Fig.S6a, and we hope that after our explanations the complete correlation matrix (former Fig S6d -> Fig S4b) will be more understandable.
Fig. 5d: We find the Venn diagrams much easier to interpret than 5c. Agree with the authors' conclusions that single mutants have a greater effect on zygotic transcripts expression, and thus transcriptional delay, compared to the double mutant. And this mirrors the developmental delay in gastrulation observed in Fig. 1. However, the authors have left out the number of transcripts that are downregulated in the single MZspg that do not overlap with the double mutant or MZsox19b (ie. there are no numbers in only the blue part of the Venn diagrams). Please indicate them so that the observed effect mentioned above applies is also reflected in transcripts that do not overlap with MZsox19b. Only including the effects of single TF LOF on shared AND non-shared transcripts can the conclusion "Pou5f3/Sox19b balance appeared to be more important for timely onset of bulk transcription, than the presence of both factors in the embryo".	We found a better way to visualize all the data, which we now present in the main Fig.3: the heatmap (Fig.3b) and relative expression graphs (Fig. 3d-g), and put the Venn diagrams into the supplementary. We labelled the number of transcripts that are downregulated only in MZspg on all Venn diagrams (former Fig.5c now Fig.S4c). We also added Fig S4d, which shows two Venn diagrams for the 2957 early transcripts, switching UP at 2.5 to 4 hpf. The left Venn shows downregulation before 4.5 hpf: "only MZspg" and "only MZsox19b" parts are approximately the same. The right Venn diagram shows downregulation of the same transcripts after 4.5 hpf: "only MZsox19b" part is much bigger (we commented on that in the main text). We also provide the new Fig. S5, with the dynamics of the whole maternal and zygotic transcription groups (a,b) and zygotic subgroups (c-h).

 • According to Lee et al. (doi: 10.1038/nature12632)), comparative analysis among the single loss-of-function mutants/morphants indicate that Nanog LOF > MZspg > SoxB1 QKD in terms of their transcriptome-wide effects. This is inconsistent with the authors' data that MZsox19b has a greater effect on the zygotic transcriptome compared to MZspg. Can the authors comment on this? 	We commented on this in the discussion: (blue marked text, p. 12) “Third, we observed an unanticipated antagonism of Pou5f3 and Sox19b at major ZGA onset (Fig. 8c). The transcriptional changes in each of MZsox19b and MZspg single mutants exceeded those in the MZsox19bspg (Fig.3b). This was in odds with the results of Lee et al.,⁹ who reported that Sox19b loss-of-function (LOF) has the mildest effects on transcription, compared with combined LOFs. We explain this discrepancy by the presence of maternal Sox19b protein in SoxB1 QKD which was used in this study. “
Fig. 5ef: Main text on page 8 - Not sure if it is the intention of the authors to relate genes regulated by Pou5f3 and Sox19b to specific enhancer types discussed in Figure 4. If so, please clarify on the choice of targets that are selected. What does "general ZGA delay"? Do the authors consider transcripts that are downregulated in the double mutant and non-overlapping with single mutant as roles in general ZGA delay?	We excluded the term “general ZGA delay” as a bad definition. We included the new Fig.7, where we linked all the transcripts to all enhancers and scored overrepresentation of PD, CD and SD enhancers in the transcription groups. We provide the mean expression profiles for the groups, show the statistical difference between them, and a couple of examples for each of them.
Figure 6:	Fig.6 is now split to Fig.2g (former 6d, Chordin morpholino rescue experiments) and the new Fig 6, the zygotic genes upregulated in the mutants (reanalyzed, related to the Table S2).
 • Fig. 6d:  - incorrectly referred to as Fig. 6g on page 8 - For the dorsalization experiments, in Fig. 6d and Fig. 2cde, the authors fluctuate between the use of Msox19bspg and Msox19bspg +/- . Please clarify this discrepancy. Do they refer to the same thing and the +/- means zygotic contribution from wild-type sperm? Or is there a heterozygous line used. The author proposes a model that either Sox19b or Pou5f3 should be maternally present in the embryo for correct D/V patterning, then the usage of the heterozygous argues against the either TF needs to be maternally present "to safeguard normal D/V" 	Msox19bspg +/- and Msox19b spg refer to the same thing and the +/- meant zygotic contribution from wild-type sperm. We removed +/- to be consistent and to avoid misunderstanding.
 - ***The authors draw a conclusion that is at odds with what was proposed in Fig. 2g for the functions of Pou5f3 and Sox19b in D/V patterning:  Fig. 6d: "the synergistic activity of Pou5f3 and Sox19b at ZGA controls D/V axis formation" Fig. 2g "D/V patterning depends on either Pou5f3 or Sox19b to be maternally present". 	We answered this comment on Fig.2 above
Discussion:  • The discussion is rather short and could benefit from extended speculations and clarifications on how Pou5f3 and Sox19b interact independently or sequentially on chromatin and specific targets and how this translates to their synergy in biological processes like D/V patterning. The molecular aspects from genomic studies (Figures 3, 4) and their function in biological processes (Figures 1, 2) are not tightly connected. The discussion can be strengthened by making reference to the 4 enhancer types, distinct molecular features regarding to the enhancer types in relation to Pou5f3- and/or Sox19b-associated changes for H3K27ac deposition, and how regulation of these transcripts connect with the observed developmental defects and changes in gene expression 	We are very grateful for the helpful suggestions of this reviewer , and we implemented them in the new discussion and Fig.8.

(figures 5, 6). Parts of Supplemental Notes can be moved here? .	
• What do the authors define as "first and second periods of zygotic transcription"?	This definition was lost between the introduction and discussion in the previous variant. It is now at the end of the discussion(p13, blue marked text): Early studies of Alexander Neyfakh described two distinct periods of zygotic gene function in teleost fish^{72,73}. The genes expressed from mid to late blastula (first period) provided the instructions for gastrulation. The genes expressed starting from midgastrula (second period) provided the instructions for organogenesis. We show that maternal Pou5f3 and Sox19b ensure the proper time gap between the first and second period in two situations. First, Pou5f3 suppressed H3K27 acetylation of some SD enhancers and premature transcription of associated genes (Fig. 8d). Similar phenomenon was recently documented in mouse ES cells, where Pou5f1 suppresses Sox2- dependent enhancers of neural differentiation genes²⁵. Second, the group of transcripts enriched in regulators of differentiation and patterning for all tissues is synergistically induced in the absence of both Pou5f3 and Sox19b at ZGA onset, thereby significantly changing the early gene expression repertoire (Fig. 8e) “ :
The usage of "select rather than activate zygotic genes" is vague. The authors can better describe the features of genes that are selected and bound by Pou5f3 and Sox19b. We find this is not well-described in Fig. 6 nor conveyed in the Discussion.	"select rather than activate zygotic genes" is removed. Due to the new extended analysis, we have more reasons to state, that Pou5f3 and Sox19b activate ventral genes and are dispensable for dorsal, which means that the major ZGA wave is regional – and that is what we tried to convey in the discussion.
Reviewer #3 (Remarks to the Author): In this manuscript, Gao et al. examine the relationship between two reprogramming pioneer factors Pou5f3 and Sox19b in driving early gene expression profiles and phenotypic outcomes during zebrafish development. This is a topic that is likely to be interesting to the broad audience of Nature Communications as orthologs of these factors drive reprogramming events in mammalian tissue culture systems. Unfortunately, because of the abbreviated format and lack of clarity in the writing, it is challenging to assess the experimental support for the mechanistic conclusions in the manuscript.	
In many cases, justification for focusing on a specific subset of genomic loci or developmental timepoint is not clear. Some examples are provided: 1. How are the PSN loci defined? (They were originally defined in a prior manuscript from the Onichtchouk laboratory, but for this manuscript must be redefined.) Furthermore, some explanation for the particular focus on this subset of loci must be included. Sites individually bound by Pou53 would be interesting to assess. If there is evidence that regions bound by individual factors are not biologically relevant this should be mentioned explicitly when justifying the selection of the PSN	We agree with the comment. As a response to this criticism, we reanalyzed the data, focusing on the ChIP-seq peaks for Pou5f3-only(P), SoxB1- only (S) and SP; Figs 5, 6, S7 and S8 of the manuscript.

sites for analysis.	
2. Why are the various experiments performed at different developmental time points? (MNase at 4.3 hpf, H3K27ac at late blastula). This is particularly confusing given the model based on phenotypes put forward in Figure 2f-h where the ratio of the two proteins is suggested to be important earlier than either of these two timepoints to determine developmental timing.	We apologize for this unclarity! MNase and H3K27ac were done exactly at the same stage (4.3 hpf, or dome stage, is one of the late blastula stages). We rationalized the choice of 4.3 hpf at the beginning of the results chapter “Sox19b and Pou5f3 act as independent pioneer factors” as follows (text on page 8 marked blue): “In order to connect the effects of Pou5f3 and Sox19b on transcription with their effects on chromatin, we chose 4.3 hpf time point in the middle of transcription time curve, before the end of MZT, to make a snapshot of chromatin state in the single mutants. Choosing 4.3 hpf time point enabled us to profile chromatin marks, which are quite low in the earlier stages”
3. What were the criteria for “top genes” for each differential enhancer noted on page 7?	The criteria were the high number of enhancers of the same type around the promoter of the gene. Now this part is rewritten (see the new Fig.7), and we don’t use “Top genes” anymore. We provide Table S5, where we classify all zygotic transcripts by presence, number and type of enhancers +/- 20 kb from TSS. We show the examples of genes with more than 3 enhancers of the same type in Fig.7.
Clarity in writing is also important for data interpretation at many other places in the manuscript. Some examples are provided: 1. What is non-consensus binding? This is not clearly defined on page 5.	We define it now at p.9 (text marked green): “We next asked which binding cues contribute to nucleosome displacement by each TF. SoxB1 and Pou proteins can recognize their consensus sox or pou motifs, respectively, or bipartite pou:sox motifs, which they are thought to bind together ⁶¹ (Fig. 5e, Table S3). Apart of that, TFs occupy DNA with specific shape (“shape motifs”) regardless of whether or not these correspond to high information content sequence motifs. This phenomenon was called non-consensus binding and is not fully understood ^{62,63} . We have previously shown that Sox19b and Pou5f3 bind to high nucleosome affinity regions (HNARs), featuring high predicted DNA shape parameter propeller twist (PT ^o) values and high in vitro predicted nucleosome occupancy ¹⁶ . To characterize the possible changes in nucleosome landscape caused by Sox19b and Pou5f3 consensus and non-consensus binding, we included sequence-specific motifs and non-consensus binding cues in our analysis. Fig. 5b and Fig. S7b show enrichment for non-consensus binding cues on Pou5f3 and SoxB1 binding regions. The example genomic region of pou5f3 gene shows distribution of non-consensus binding cues (HNARs), consensus binding motifs, TF binding and nucleosome displacement (Fig.5f)”
2. How are “open” and “closed” regions defined in the MNase-seq experiment?	We do not use these definitions anymore.
3. The clustering of the H3K27ac peaks into five groups should be better defined as these are the basis for future analysis. Furthermore, the classes should not be classified as “active” versus “repressed” since the classification is based on acetylation and not activity	We have re-analysed our data, taking only the regions around ChIP-seq peaks for Pou5f3 and SoxB1. This approach narrowed down the analysis, but greatly simplified it and facilitated interpretation of

	the results. We performed new k-means clustering only around the peaks, and recovered former enhancer clusters 1-3, but not 4 (which was to be expected, as class 4 did not correlate with Pou5f3 and SoxB1 genomic binding). We describe it on p 10 (green marked text): “To infer the putative enhancer activity changes in the mutants, we selected Pou5f3 and/or Sox19b-binding peaks, flanked by H3K27- acetylated regions in at least one genotype, and clustered them by H3K27ac signals in three genotypes using k-means (Methods). This resulted in four clusters of enhancers, by downregulation of H3K27ac in the mutants (Table S3): CD - codependent, PD- Pou5f3 - dependent, SD - Sox19b-dependent, and U – unchanged (see the heatmap at Fig.6a and the examples at Fig.6d). We describe the details in the Methods, in the sub-chapter “Selection and analysis of putative enhancers”(green marked text on p.36): “We considered TF- binding peak as putative enhancer, if H3K27- acetylation of the 1 kb region around the peak exceeded arbitrary threshold in at least one genotype. The Pou5f3- only, SoxB1- only, and double binding peaks (P,S and SP) were considered. Technically, we used the following procedure. TF- binding peaks BED files were extended to 1 kb (+/- 500 bp from the peak summits). H3K27ac MZsox19b-rep-1, H3K27ac MZspg and H3K27ac WT Bigwig files were converted to bedgraph format, all the regions where log2 ChIP/Input was less than 0.5 (arbitrary value) were filtered out. The resulting bedgraph files were joined with TF-peak BED files. The peak was considered as putative enhancer, If at least 500 bp around the peak were H3K27-acetylated over the arbitrary threshold (0.5 log2ChIP/Input). 70% of SP, 43% of S and 30% of P peaks were selected as enhancers using this criteria(Table S2). To assign four enhancer types, k-means algorithm (ploheatmap program in deepTools) was applied to the list of the putative enhancers, with cluster number 4. The CD, PD and SD groups were robustly recovered also when using the higher number of clusters: higher k- numbers resulted in splitting “unchanged” to several groups, by H3K27ac distribution or abundance. For statistical analysis and violine plots, H3K27acetylation per 1 kb was calculated for each enhancer in each genotype. H3K27 acetylation change in the mutant was calculated as log2 mut/WT ratio (data available as Source data file). Enhancers was linked to the closest transcription start sites of ENSEMBL genes within +/-20 kb, and to corresponding ENSEMBL transcripts”
The deletion in Sox19b must be confirmed to be a null allele. It is possible that a downstream start codon could produce a truncated product that acts as dominant negative/neomorph and that this causes a more severe phenotype than the null	We can not exclude this possibility, but we find it very unlikely. We show now in the Fig.S3a that the levels of maternal sox19b are reduced 15-fold already before the

and that this activity might be dependent on Pou5f3 function. This would explain the phenotypic suppression in the double mutant.	zygotic transcription start. There is no template to translate: sox19b mutation is a frameshift in the first exon, and it causes nonsense-mediated sox19b mRNA decay.
The title for Figure 3 doesn't fit with what is shown. It mentions independent binding by Sox19b and Pou5f3, but all sites analyzed are PSN sites so they both should bind all sites analyzed. Instead, this figure analyzes chromatin accessibility and the motifs enriched under these regions.	We completely agree with the comment. We analyzed P,S and SP peaks in the current version of the manuscript. The title of the Fig.5 (which is the former Fig.3) is now "Pou5f3 and Sox19b displace nucleosomes using different binding cues".
What evidence is there that Pou5f3 directly affects H3K27ac? The argument on page 6, "Hence, Pou5f3 binding to type 1 and type 2 enhancers displaced nucleosomes and directly induced local H3K27ac" does not seem to be clearly supported from the correlative data. Couldn't Pou5f3-mediated accessibility increase binding of another factor or factors that then recruits the acetyltransferase?	We agree and we removed the word "directly". There is an evidence that Pou5f3 binds DNA on its cognate sites, but there is no evidence that acetylation is direct. It may be mediated by interaction with another factor.
Have motif searches been done for these different enhancers?	Yes, the motif searches have been done. We found C2H2 Zn- Fn motifs, HLH domain motifs, ASCL-type motifs and three types of repeats. We tested if these motifs preferentially associate with one type of enhancer, but they did not; therefore we did not mention them.
In addition, has the timing of chromatin accessibility and H3K27ac establishment been tested?	Chromatin accessibility on the regulatory elements and H3K27ac are established in parallel, gradually, during zygotic genome activation. This was shown in several publications, including Bogdanovic, et al.. Genome Res 22, 2043-2053, (2012) Liu, G., Wang, W., Hu, S., Wang, X. & Zhang, Y. Genome Res 28, 998-1007, (2018). Palfy, M., Schulze, G., Valen, E. & Vastenhouw, N. L. PLoS Genet 16,(2020). Sato, Y. et al. Development 146, (2019).
What data support the fact that chromatin accessibility precedes acetylation as shown in Figure 7a?	Fig. 7a is now 8a. We did not intend to state that there is a time gap between TF binding, nucleosome displacement, and acetylation. We have re-drawn the scheme to simplify it and remove this impression. These can be simultaneous events – i.e. when TFs bind to the DNA, they displace the central nucleosome and recruit acetyltransferase to flanking nucleosomes. We included into the introduction the references to four different models explaining Sox/Pou interactions in vivo. Out of them, 8a is a model which explains our observations in the most parsimonious way, but we don't have a strict proof for it.
On page 7-8 in discussing the time resolved RNA-seq, the authors try to deconvolute whether the changes in gene expression are causing the developmental delay or are due to the developmental delay, but this is not clear as written. Thus, concerns remain as to where the changes in gene expression are due to the developmental delay or causing it	We were not planning to address the causal connection between the developmental delay and transcription, and we have re-written this part of the results to remove this impression. What we wish to show here is that the gene expression changes are broader in the single mutants than in the double mutant.
The model in Figure 7a is confusing in so far as it introduces P300, which is never mentioned in the text at all.	We removed P300 from the scheme (now Fig.8a)

In Figure 7d, why are there individual Pou5f3 and SoxB1-bound loci since all the analysis in the paper was focused on regions where both factors bind?	This scheme is now in Figure 8d and re-drawn; it shows the biological balance between Pou5f3- and SoxB1- transcriptional targets, and it is related to the discussion: Pou5f3 keeps the neural genes from premature expression .
While there is already a ton of data analysis in the manuscript, MNase-seq and/or H3K27ac on the double mutant would be very informative for the model put forward.	We could not accomplish this experiment due to technical reasons (the experiments require large amounts of material, which we could not collect in time). As this reviewer noticed, the manuscript is overloaded with data, which were presented in abbreviated format, compromising the clarity. Given the page limits and the need to streamline the results it would be difficult to fit another data set into the paper without compromising clarity. We do not feel that the presentation will significantly benefit from it. We re-structured the manuscript, putting the RNA time curves analysis comparing the single and double mutants earlier in the text. We also simplified and streamlined the genomic analysis. We hope that the conclusions we were able to draw are robust and self-sufficient without additional experiments.
Reviewer #4 (Remarks to the Author): In this study, Gao et al. analyzed Pou5f3 and Sox19b mutants to understand the functional differences and interplay between the two TFs in early zebrafish development. The study revealed that Pou5f3 and Sox19b function differently on nucleosome displacement and have distinct regulatory roles in gastrulation and organogenesis. Furthermore, the authors suggested that the balance of the two TFs is important for ZGA timing, which is a surprising result. The conclusion is important for readers in the field of developmental biology, epigenetics, and reprogramming. However, the paper lacks several crucial data to support their claim, which are required before publication.	
Major 1. The reviewer is not well convinced with their conclusion that the balance of Pou5f3 and Sox19b determine the timing of ZGA. In Figure 5c d, the difference between MZspg and MZsox19bsp is not obvious. Using the RNA-seq data, the expression of all zygotic-only genes (excluding those that do not have maternal mRNAs) should be shown as heatmap so that readers can easily evaluate the delay of ZGA at genome-wide level in mutant embryos.	We agree that the statement “the balance of Pou5f3 and Sox19b determine the timing of ZGA” is too strong, but we wish to insist that there are indeed compensatory relationships between Pou5f3 and Sox19b at the dawn of the major ZGA. On the request of this reviewer, we provided a heatmap in Fig.3b with the numbers of zygotically delayed transcripts. The mRNA present in the embryo at each given early time point is a mix of maternal and zygotic transcripts for more than 70% of genes (“maternal-zygotic”, Harvey et al., ref⁵²). Together with zygotic-only genes, the heatmap includes maternal-zygotic genes, for which zygotic transcription overrides maternal RNA decay and we see them switching UP. We included this clarification in the text (page 6, highlighted in pink: For all the genes in switch UP groups, zygotic increase of the transcript levels exceeds maternal RNA decay.
Also, it is unclear whether the balance of Pou5f3 and Sox19b regulates the timing of ZGA or the expression level of some genes. With Figure 5e, one would speculate that the expression timing is not changed but absolute expressions are reduced in single mutants. It would be more informative to analyze using PolII ChIP-seq for precise evaluation of ZGA	This is a valid concern. In response to this comment, we show the statistics of the relative zygotic expression (log2 FC to 2.5 hpf) and mean profiles for the zygotic and maternal genes (Fig.S5 a,b) and for the subgroups of zygotic transcripts (Fig. 3d-e

timing, since RNA-seq of early embryos have both maternal and zygotic transcripts.	and Fig.S5c-f). The differences in zygotic expression between the genotypes are significant from the earliest time points (3-3.5 hpf)
2. The prediction tool of nucleosome positioning used in this study was established based on yeast data (Kaplan et al). However, this prediction method is not suitable for vertebrate genomes, and in fact, it has been shown in zebrafish that predicted nucleosome organization shows very weak correlation (Zhang et al., 2014 Genome Res). Also the Kaplan rule does not hold true in hypomethylated regions of the medaka-fish genome (Nakamura et al., 2017 Epigenetics & Chromatin). The authors should show the validity of using this tool in zebrafish by genome-wide comparison of prediction and MNase-seq data. Alternatively, it would be more informative to show nucleosome occupancy using MNase-seq data of pre-ZGA embryos.	We greatly appreciate this reviewer comment! We have no doubt that the tool of Kaplan et al., should not be used for in-vivo nucleosome predictions. In the previous version of our manuscript we used this tool unconventionally, to find the regions with the DNA properties, promoting non-consensus binding of transcription factors (we call them "High Nucleosome Affinity Regions", or HNARs). In our previous work (Veil et al., 2019, Genome Research) we found, that TFBS of Nanog, Pou5f3 and SoxB1 have increased Kaplan values. We now show it in this manuscript too, in the figures Fig. 5b and Fig. S7b. We explain it on page 10: "We asked next, which binding cues contribute to nucleosome displacement by each TF, and if they are the same. SoxB1 and Pou proteins can recognize their own consensus sox or pou motifs, respectively, or bipartite HMG/POU pou:sox motifs, where they are thought to bind together (Fig. 5e). Apart of that, it was shown that TFs preferentially occupy DNA with specific structures ("shape motifs") regardless of whether or not these correspond to high information content sequence motifs; the phenomenon was called non-consensus binding and is still not fully understood (78-80). As we have previously demonstrated that Sox19b and Pou5f3 bind to the High Nucleosome Affinity Regions (HNARs) in the context of high predicted DNA shape parameter Propeller Twist (PT°) values and high in-vitro predicted nucleosome occupancy (20), we regarded these DNA features as promoting non-consensus binding". In the resubmitted version of the manuscript, we are using PT° instead of Kaplan's program, to avoid misunderstanding. If we would use Kaplan's program again, everyone will think that we are looking for the nucleosomes. But we are not looking for the nucleosomes, we are looking for the properties of DNA.
. Figure 3c, d; The effect of sox19b at closed sites seems quite small. The subtraction (Fig 3d) shows no significant differences, and therefore, it is difficult to conclude that Sox19b affects on both open and closed regions.	We have removed this statement, and we have reanalyzed the data in a different way (see Fig.5 and S7 of the resubmitted manuscript). We scored two quantities: nucleosome displacement on the consensus motifs (Fig.5g,h), and nucleosome displacement elsewhere in the genome (Fig. 5 i,j). Both effects are real; for Sox19b the strength of genome-wide nucleosome displacement on HNARs is comparable with that on the specific motifs, for Pou5f3 it is weaker. Additional analysis is on Fig.S7.
4. Figure 4b; It is not clear whether nucleosome displacement or binding itself directly or indirectly induced H3K27ac. This needs to be discussed	The data were reanalyzed. H3K27ac is now Fig.6a (Fig. 8c), nucleosome displacement is on S8g. We analyzed H3K27ac and nucleosome displacement in parallel, for dependency on consensus motifs in each enhancer type – Fig.65 f,g and Fig.S8ij. We state, based on this analysis, that nucleosomes are displaced and H3K27ac

	binding is promoted by the same TF. Nucleosome displacement and H3K27ac can be uncoupled, i.e. TF directly or indirectly interacts with acetyltransferase and with nucleosome remodeller – or they can be dependent on each other. However, we cannot address the causal connection between these two things from the perspective of our data, we can only speculate on that.
5. It is also not clear whether changes of H3K27ac levels in enhancer type 3 and 4 is direct or indirect. For this, the authors need to show Pou5f3 and Sox19b ChIP binding as heatmap in Figure 4b.	To simplify and streamline the H3K27ac analysis, which was too complicated previously, we now analysed only direct enhancers, defining them from the very beginning as H3K27ac-enriched regions around Pou5f3 and ChIP binding peaks (see Methods). We recovered the former enhancers 1,2 and 3 (now Pou5f3-dependent, PD, codependent, CD, and Sox19b-dependent, SD. The Pou5f3 and SoxB1 ChIP binding heatmap is shown in Fig. S8d. However, we did not recover the former type 4, which was to be expected, as type4 was underrepresented around Pou5f3 and SoxB1 peaks.
6. Overall, the story of this manuscript seems complicated or not straightforward. Genes activated at ZGA are mostly housekeeping genes and some developmental genes required for gastrulation and later organogenesis. However, most of the analyses only focused on developmental genes.	Addressing this criticism we separated developmental and housekeeping genes using GO analysis of the groups (Fig.3c). Groups A and B on the heatmap 3a are developmental and they are discussed separately.
The first story on the role of Sox19b and Pou5f3 in displacement of nucleosomes deals with all genes, but the rest of the experiments focused on developmental genes related with the mutant phenotypes, demonstrating that Sox19b and Pou5f3 differentially regulate a subset of these developmental genes.	Taking this criticism into account we restructured the paper, trying to keep specific and non-specific effects separately.
How do Sox19b and Pou5f3 exhibit distinct roles during early and later embryogenesis? It is not clear how the first story is related with other late stories and how the two factors works differently. To address this, binding targets of each factors, and their overlaps, need to be clarified by re-analyzing of published ChIP-seq data.	There is no published ChIP-seq data for Sox19b, unfortunately. The binding specificity of SoxB1 genes is the same, and we were using our old published SoxB1 ChIP-seq(Leichsenring et al., Science, 2013). Binding targets of SoxB1 and their overlaps with Pou5f3 were analyzed in this paper. What we have done now is to link all zygotic genes to enhancers, sort them by enhancer type, and compare the mean expression dynamics (see the results in the new Fig. 7)
Minor 6. There are two Figure 2b, one should be 2e.	fixed

- 1 Bogdanovic, O. *et al.* Dynamics of enhancer chromatin signatures mark the transition from pluripotency to cell specification during embryogenesis. *Genome Res* **22**, 2043-2053, doi:10.1101/gr.134833.111 (2012).
- 2 Liu, G., Wang, W., Hu, S., Wang, X. & Zhang, Y. Inherited DNA methylation primes the establishment of accessible chromatin during genome activation. *Genome Res* **28**, 998-1007, doi:10.1101/gr.228833.117 (2018).

- 3 Palfy, M., Schulze, G., Valen, E. & Vastenhouw, N. L. Chromatin accessibility established by Pou5f3, Sox19b and Nanog primes genes for activity during zebrafish genome activation. *PLoS Genet* **16**, e1008546, doi:10.1371/journal.pgen.1008546 (2020).
- 4 Sato, Y. *et al.* Histone H3K27 acetylation precedes active transcription during zebrafish zygotic genome activation as revealed by live-cell analysis. *Development* **146**, doi:10.1242/dev.179127 (2019).

Reviewers' Comments:

Reviewer #1:

Remarks to the Author:

The current version of the manuscript by Gao et al, describes pou5f3 and Sox19b function in the early zebrafish embryo, which is likely to be of significant interest to researchers interested in pluripotency and development during the mid-blastula transition. The rigor of the manuscript has greatly improved in the revision, and the authors have responded well to all of the concerns raised by this reviewer. Stylistically the revised manuscript is perhaps even more dense than the original, and the lack of a singular narrative continues to make this a more challenging manuscript to read/follow. However, these concerns are somewhat mitigated by the value of information presented for interpreting functions of these key TFs in the early embryo, and by the fact that the multiple conclusions of the manuscript are summarized in the discussion. It is possible moving some additional data to supplementary to better highlight key points in the main figures could also help improve the accessibility of the manuscript. For figure 8, numbering these to correspond to the discussion and points including more informational text in the actual figure (so the reader doesn't need to keep going back to the legend) could also help with clarity. Of minor note, I would recommend changing color scheme of heat maps to be red/green colorblind friendly. I'm also not sure I understand panel 3d/4d

Reviewer #2:

Remarks to the Author:

In their revised manuscript, the authors addressed all of our concerns. Additional experiments, controls, and analyses have been performed. The manuscript reads much better and the presentation and data interpretation of RNA-sense has been made more understandable and supportive of the authors' conclusions. We therefore think the manuscript is ready for publication.

Some suggestions and minor suggestions:

RELATED TO FIGURES AND LEGENDS

Figure 1

Explain why IVF was used vs natural crosses.

MZSox19b embryos are obviously smaller than wt embryos. Acknowledge this in text.

Figure 2

Would it be clearer to move panels a-c to Figure 1? To group all data on MZsox19b?

Legend a: add 'in wt and MZsox19b embryos'

It is not abundantly clear what SoxB1 refers to. It is mentioned once in the text but it would benefit the reader if it is explained every time it is mentioned.

Figure 4

Figure letters in legend are not consistent. Missing legend for 4D.

Figure 5

Panel e: Does it not make more sense to refer to sox:pou instead of pou:sox, given the order of the motifs?

Panel f: Why show Pou5f3 out of all the possible genes? Is a bit confusing.

Panel f: There are no examples for ONLY HNAR non-consensus or ONLY consensus motifs. Including these would be helpful (in relation to panels g and h).

Figure 7

Panel c: There is no H3K4me3 signal, even in wt. Why?

We would find it useful and relevant to also add chromatin accessibility signals from ATAC-seq here.

Figure S1

Panel a: We find it hard to compare images between WT vs MZsox19b. Better to include nuclei counting?

Panel b: It is unclear how cell cycle length is indicated here (y-axis).

Typo in figure legend: "main test" should read "main text"

Video Movie S2

Could not be played

Figure legends

Quite often, remarks in Figure legends 'Note that...' would be useful to include in the text.

TEXT

We feel the title does not capture the content of the paper.

Page 3

"Out of them, direct pioneer binding to nucleosomes was demonstrated that THUS far only for Drosophila genome activator Zelda"

Consistency of mammalian POU5F1 naming in Paragraph 3

We would like to see the data being related back to the 4 models that are introduced in paragraph 3.

Page 5

Missing word: "The duration of the pre-MBT cell cycles was the same BETWEEN the MZsox19b and wild-type embryos"

"Apart of that" should be "apart from that" (also on page 9). Or use "In addition?"

Page 9

Comparative analysis of MNase-seq experiments is correctly referenced (ref. 16), but in the Methods section (page 33), the remapping of MNase signals in MZspg and WT (remapped from ref. 20) appears to reference another paper and not Veil et al (ref. 16).

Page 10

"examples in Fig.6d" should read "examples in Fig.6e"

Page 12

In the second paragraph: 'First, on the mechanistic level...' we feel a bit more information should be given to connect the conclusions to the results.

Related to comment for Figure 7: The authors indicate the following: "we show that Pou5f3 sequence-specific binding activates early zygotic genes by increasing chromatin accessibility and promoting H3K27 acetylation on Pou5f3- dependent and -codependent (CD) enhancers (Fig. 6g, Fig. 7)." There is actually no figure that includes both H3K27ac and ATAC-seq data. We feel in the scope of this paper, these signals could be included for example genes to support the authors statement.

Reviewer #3:

Remarks to the Author:

The clarity of this revised manuscript by Gao, Veil et al. is much improved, but there are still areas requiring further improvement/clarification. The counterintuitive compensation of transcript levels in the Pou5f3, Sox19b double mutant as compared to the single mutants is intriguing. The authors present additional data more rigorously characterizing the Sox19b mutant phenotype. These data, included in Figures 1 and 2, are convincing and rigorously controlled. However, there remain some questions regarding the mechanistic conclusions that can be drawn from these data.

Throughout the discussion of the RNA-seq time course, the authors fail to discern what effects are likely direct versus indirect, despite the fact that their RNA data sets extend over hours of development. The transcripts produced during the earliest wave of genome activation include transcription factors that affect expression later. Thus, it is likely that a proportion of the effects on transcript levels is not directly due to loss of Sox19b or Pou5f3 binding to the cis-regulatory regions of the genes identified by RNA-seq, but instead due to indirect effects of their earlier absence. At the end of the section describing the bulk transcription, the authors write, "In sum we demonstrated, that Pou5f3 and/or Sox19b activate 24% of the zygotic transcripts . . ." Whether or not the authors intended it this way, this sounds like these factors are directly responsible for this activation. Additional analysis trying to tease apart the direct from indirect effects would significantly strengthen the conclusions that can be drawn regarding the antagonistic functions of these two factors.

Similarly, it is clear from Figure S3B that additional SoxB1 family members are expressed during this time course. These family members might be able to substitute for Sox19b at some loci later in development, yet the authors do not discuss these factors or how they might contribute to gene expression patterns.

In parts, the conclusions in the paper are overstated. The authors need to increase their clarity about whether the data upon which conclusions are based are in the manuscript, in previously published papers, or in both. For example, in the Discussion the speculation about binding of each factor in the absence of the other and the opposing roles is not based on any binding data since none of the data in this manuscript analyze binding of either factor. Similarly, the first conclusion of the Discussion is that Pou5f3 and Sox19b modify chromatin independently even at regions where they bind together. This is not totally clear to me as the authors never investigate binding of each factor in the absence of the other. Here and throughout the manuscript, the authors need to be cautious about the broad conclusions they are drawing from the data included and their analysis of these data. Some of the mechanistic conclusions are severely limited because of lack of data in the double mutant. While it is understandable that the MNase and H3K27ac ChIP data may not be able to be generated in the double mutant, the authors must acknowledge that the mechanistic conclusions regarding the compensation are limited by the absence of these data.

The use of single replicates for the H3K27ac ChIP-seq for both WT and MZspg is concerning. While it is understandable that this material is challenging to generate, reproducibility between replicates is important to confirm.

Supplemental figures showing the reproducibility between replicates of RNA-seq and MNase-seq should be included.

Minor:

The citation of Fernandez-Garcia for the binding of Zelda to nucleosomes is correct. However, McDaniel et al. Mol Cell 2019 published these results initially and have more detailed analysis of this binding. This citation should be included along with the current citation 17.

There is typo in the third paragraph Introduction. "POU5f1" should be "POU5F1".

The authors discuss both in the Introduction and Discussion that histone levels serve as general transcriptional repressors before ZGA and that the competition between histones and transcription factors time ZGA. While the papers they cite certainly provide evidence that histone levels play a role, as written these statements are too strongly worded. There certainly remain a number of

genes whose regulation is not directly controlled by this competition, but rather these observations may also include indirect effects of histone levels on cell-cycle timing.

The authors explain the difference in their analysis from that previously published by Lee et al. as due to the fact that the mutant generated eliminates maternally provided Sox19b protein. What evidence is there of a pool of maternally provided protein? Given that Sox19b was implicated in genome activation in the Lee et al. paper based on having high translation rates in the early embryo, it seems unlikely that there is also a significant amount of maternally supplied protein.

The authors should avoid using red and green in their heatmaps so that they are interpretable for color blind readers.

Explanation of the headings for the columns in the Supplemental Tables should be provided in the legends for these tables.

Movie S2 explaining the RNA-sense did not work for me.

Reviewer #4:

Remarks to the Author:

Gao et al restructured the paper and added some new experiments/analyses to examine the function of Sox19b and Pou5f3 in zebrafish ZGA. However, several major conclusions of this study are still insufficiently supported by their data, and it seems that the study lacks significance and novelty in its present form. Readability of the manuscript has been improved, but it is still difficult to relate the phenotypic analyses (Fig 1,2) to molecular analyses (Fig 3~7). Below are major concerns.

Their conclusion that the balance of Pou5f3 and Sox19b is important for zygotic transcription is not well supported. Although the level of bulk zygotic transcription at 3.5hpf is higher in MZsox19bspg than in MZsox19b and MZspg, the difference is very subtle (Fig S5a middle), and as shown in Fig S5a (left), MZsox19bspg and MZspg show almost same dynamics. The percentage of downregulated genes in MZsox19bspg and MZspg does not seem significant (24% vs 32%). In Fig 3b, only 364 genes (in group B) shows downregulation in single mutants. So they can only say that the balance is important only for those 364 genes.

At the global level, it seems that MZsox19bspg and MZspg do not show a great difference to each other. They also do not show double mutant data in other analyses (MNase and H3K27ac), and thus the underlying mechanism is completely unknown.

(Fig 5d & 8a); Why do the authors conclude that only Sox19b contributes to chromatin accessibility? The result of Fig 5d shows that nucleosomes at SP peaks are displaced also in MZspg, suggesting that the accessibility also depends on Pou5f3. Furthermore, the relationship between SP peaks and H3K27ac clusters are not clear. The conclusion shown in Fig. 8a is not appropriate.

(Fig 7) The paragraph describing the association between enhancers and transcription is difficult to follow. The data presented are not supportive enough to conclude that Sox19b-dependent enhancers are not involved in the transcriptional activation. SD-linked genes are downregulated in MZsox19b, and thus, it is likely that these genes are activated by Sox19b. H2K27ac ChIP for MZsox19bspg is also required.

The relationship between the first two figures (Fig1,2) and the latter (Fig 3~7) is still unclear.

What is the molecular basis of the developmental delay? Do they think that the group B genes downregulated only in single mutants are responsible for this?

Figure 3b shows that a fraction of zygotic genes are not activated, rather than delayed.

The authors claim that the MZsox19bspq mutants were less delayed, but the data presented are not clear (Fig 2f). The movie is difficult to compare the subtle differences. They should show quantitative data.

Dear Reviewers,

Thank you in advance for looking at our manuscript again! In the revised version of our manuscript, we performed a new key set of experiments, Assay for Transposase-Accessible Chromatin (ATAC-seq) in the double and single mutants as well as in the wild-type, at two blastula stages. The data are deposited in GEO, link for the reviewers:

To review GEO accession GSE188364:

Go to <https://www.ncbi.nlm.nih.gov/geo/query/acc.cgi?acc=GSE188364>

Enter token cdejemkknabpet into the box

We defined the set of 102 945 accessible chromatin regions (ARs) as an intersection of the ATAC – seq peaks called from all six replicates for the wild-type embryo. We then analyzed the chromatin accessibility changes on these regions in the single and double mutants and linked them to the changes in transcription. The figures 5 – 8 are replaced with the new versions, the corresponding text is re-written. All the main conclusions in our resubmitted manuscript are based on the new analysis of chromatin accessibility and transcriptional changes. This is reflected in the new Discussion.

We found all your comments appropriate. We thank the reviewers 1 and 2 for useful and detailed suggestions of how to improve the presentation of our manuscript. We also thank the reviewers 3 and 4 for their serious criticisms and for indicating the overstatements in our conclusions. Their critical comments prompted us to make the new experiments which clarified the state of things. We address each of the specific reviewer comments below.

Point-by-point answer to the reviewer comments:

Reviewer 1:

Reviewer #1 (Remarks to the Author):

The current version of the manuscript by Gao et al, describes *pou5f3* and *Sox19b* function in the early zebrafish embryo, which is likely to be of significant interest to researchers interested in pluripotency and development during the mid-blastula transition. The rigor of the manuscript has greatly improved in the revision, and the authors have responded well to all of the concerns raised by this reviewer. Stylistically the revised manuscript is perhaps even more dense than the original, and the lack of a singular narrative continues to make this a more challenging manuscript to read/follow. However, these concerns are somewhat mitigated by the value of information presented for interpreting functions of these key TFs in the early embryo, and by the fact that the multiple conclusions of the manuscript are summarized in the discussion.

R1-1. It is possible moving some additional data to supplementary to better highlight key points in the main figures could also help improve the accessibility of the manuscript. For figure 8, numbering these to correspond to the discussion and points including more informational text in the actual figure (so the reader doesn't need to keep going back to the legend) could also help with clarity. I'm also not sure I understand panel 3d/4d

A1-1. We agree. To improve the clarity of presentation, we reorganized the figures 1 and 2 (1 refers to – *MZsox19b*, 2 refers to- *MZsox19b*spg now). We moved the former panel 3d (supporting statistics for current Fig.3d) to the supplementary. We removed the former panel 4d (supporting statistics for the current Fig. 4c) to simplify the presentation. We performed a new set of ATAC-seq experiments on all mutants; the figures 5-8 are new. We moved all MNase-seq on the single mutants to the supplementary and used them as a support data for the main ATAC-seq data set. As we did not see the correlation between global developmental delay in *MZsox19b* and chromatin accessibility by ATAC-seq in *MZsox19b* (Fig. S8g), we removed the most speculative part of discussion (where we implied that such a connection exists. It may still exist, but we don't have convincing data to show it). We restricted the discussion to three points, which are novel and validated by our study. We hope that the manuscript became more conclusive and is easier to read now.

R1-2. Of minor note, I would recommend changing color scheme of heat maps to be red/green colorblind friendly.

A1-2. Done (Fig.3b, Fig. S4c).

Reviewer #2 (Remarks to the Author):

In their revised manuscript, the authors addressed all of our concerns. Additional experiments, controls, and analyses have been performed. The manuscript reads much better and the presentation and data interpretation of RNA-sense has been made more understandable and supportive of the authors' conclusions. We therefore think the manuscript is ready for publication.

Some suggestions and minor suggestions:

RELATED TO FIGURES AND LEGENDS

Figure 1

R2-1. Explain why IVF was used vs natural crosses.

A2-1. Done. Page 26, the first paragraph of the Methods: "*In vitro* fertilization was used to precisely synchronize the embryos for scoring the duration of pre-ZGA cell cycles (experiment in Fig.S1), natural crosses were used for all other experiments".

R2-2. MZSox19b embryos are obviously smaller than wt embryos. Acknowledge this in text.

A2-2. Done. The second phrase in the "Results" section, page 5: The MZ*sox19b* embryos lacking both maternal and zygotic Sox19b, and M*sox19b* embryos lacking maternal Sox19b developed into fertile adults, albeit more slowly than controls, and were smaller in size (Movie S1).

Figure 2

R2-3. Would it be clearer to move panels a-c to Figure 1? To group all data on MZ*sox19b*?

A2-3. Yes, it is much clearer! This suggestion is very much appreciated, it is done (see new Fig.1)

R2-4. Legend a: add 'in wt and MZ*sox19b* embryos'

A2-4. Now Fig.2a is Fig.1d, corrected (p.15)

R2-5. It is not abundantly clear what SoxB1 refers to. It is mentioned once in the text but it would benefit the reader if it is explained every time it is mentioned.

A2-5. We agree, and we changed that. All factors are named in the third paragraph of results (p.5), on the p.8 (the first paragraph of the sub-chapter "Changes in chromatin accessibility underlie the changes in zygotic expression repertoire in the double MZ*sox19b*spg mutant"). "SoxB1 and Pou protein families"- p9, third lane.

Figure 4

R2-6. Figure letters in legend are not consistent. Missing legend for 4D.

A2-6. Figure 4 is simplified, legend now is consistent.

Figure 5

R2-7. Panel e: Does it not make more sense to refer to *sox:pou* instead of *pou:sox*, given the order of the motifs?

A2-7. We agree! Done (Fig.5e, Fig.6d)

R2-8. Panel f: Why show Pou5f3 out of all the possible genes? Is a bit confusing. Panel f: There are no examples for ONLY HNAR non-consensus or ONLY consensus motifs. Including these would be helpful (in relation to panels g and h).

A2-8. We removed the panel f, and we have re-done and re-written the whole second part of the manuscript starting from the Fig.5, because of the following reasons. We obtained the new data by ATAC-seq technique, and compared the results with MNase-seq. We do see by MNase-seq, that the nucleosomes are preferentially removed from HNARs with high PT^o/*in-vitro* nucleosome occupancy/GC content in MZ*sox19b*, but we do not see this effect in ATAC-seq. Two techniques are different: MNase-seq shows the nucleosome positioning all over the embryo, while ATAC-seq reflects only the cell population with accessible chromatin. The result simply means that the global nucleosome repositioning in MZ*sox19b* occurs on the regions which are not accessible for Tn5. It is unlikely, that these effects are directly connected to transcription. We replaced the whole figure 5 with the straightforward analysis of ATAC-seq data, focusing on the sequence-specific binding of the TFs. We also found that real and *in-vitro* nucleosome occupancy/GC content of all accessible regions at blastula, including those which are not directly bound by Pou5f3 and Sox19b, is higher than genomic average: they are all HNARs! (Fig. 5g, S6f-i). The counterintuitive finding, that all gene regulatory regions in human genome have high predicted nucleosome occupancy and GC content was published at 2010 (Tillo, D., Kaplan, N., Moore, I.K., Fondufe-Mittendorf, Y., Gossett, A.J., Field, Y., Lieb, J.D., Widom, J., Segal, E., Hughes, T.R., 2010. High nucleosome occupancy is

encoded at human regulatory sequences. PLoS One 5, e9129) and cited 106 times since then. The authors proposed that high nucleosome preference is directly encoded at regulatory sequences in the human genome to restrict access to regulatory information. This principle seems to be universal, as we already saw in our previous work (HNARs, Veil et al., 2019). We see it in this work as well.

Figure 7

R2-9. Panel c: There is no H3K4me3 signal, even in wt. Why?

A2-9 Now Fig. 8b: we have chosen another example to avoid confusion.

R2-10. We would find it useful and relevant to also add chromatin accessibility signals from ATAC-seq here.

A2-10. ATAC-seq in four genotypes is added (Fig.8b)

R2-11. Figure S1

Panel a: We find it hard to compare images between WT vs MZsox19b. Better to include nuclei counting? Panel b: It is unclear how cell cycle length is indicated here (y-axis).

A2-11. We included nuclei counts and the stages in Fig.S1a. The new graph type (with the same data as before) in the Panel b, shows how many embryos and in which stage we scored in each time point.

R2-12. Typo in figure legend: "main test" should read "main text"

A2-12. We found this typo under Fig.S3; corrected (p.8, supplementary Material). Thanks a lot!

R2-13. Video Movie S2. Could not be played

A2-13. We reformatted the video, it should play now.

Figure legends

R2-14. Quite often, remarks in Figure legends 'Note that...' would be useful to include in the text.

A2-14. It is a very good idea. We included it in two positions in the sub-chapter "Sox19b and Pou5f3 act as independent pioneer factors", p.9-10, and also in the following chapters.

TEXT

R2-15. We feel the title does not capture the content of the paper.

A2-15. We agree that the title did not capture the content of the previous variant of the paper, but we would argue that it fits to the content now (see the abstract and the results on Fig. 5 and the corresponding text).

Page 3

R2-16 "Out of them, direct pioneer binding to nucleosomes was demonstrated that THUS far only for Drosophila genome activator Zelda" Consistency of mammalian POU5F1 naming in Paragraph 3

A2-16 Thank you for spotting this! Both corrected.

R2-17. We would like to see the data being related back to the 4 models that are introduced in paragraph 3.

A2-17. We find that this is a very good point! We included it in the second paragraph of the discussion, p. 12: "Out of four different scenarios for the interactions of the mammalian homologues in ES cells, which were outlined in the introduction²³⁻²⁶, our results agree with the conclusions of the last study: POU5F1 and SOX2 operate in a largely independent manner even at co-occupied sites²⁶".

Page 5

R2-18. Missing word: "The duration of the pre-MBT cell cycles was the same BETWEEN the MZsox19b and wild-type embryos". "Apart of that" should be "apart from that" (also on page 9). Or use "In addition?"

A2-18. We corrected: "The duration of the pre-MBT cell cycles was the same for MZsox19b and wild-type embryos (Fig.1b, Fig. S1)" and " In addition,..." on p.6

Page 9

R2-19. Comparative analysis of MNase-seq experiments is correctly referenced (ref. 16), but in the Methods section (page 33), the remapping of MNase signals in MZspg and WT (remapped from ref. 20) appears to reference another paper and not Veil et al (ref. 16).

A2-19. Corrected in the methods.

Page 10

R2-20 "examples in Fig.6d" should read "examples in Fig.6e"

A2-20 Fig.6 now is different, text removed.

Page 12

R2-21. In the second paragraph: 'First, on the mechanistic level...' we feel a bit more information should be given to connect the conclusions to the results.

A2-21. We agree, we referenced to the results: " First, on the mechanistic level, we found that Sox19b and Pou5f3 are both involved in establishment of chromatin accessibility at blastula stages and act mostly independently. Sox19b promotes accessibility on sox motifs, Pou5f3 on pou and sox:pou motifs (Fig. 6d-f). Sox19b and Pou5f3 can act additively or redundantly if their cognate motifs are present nearby (Fig. S7g-i)".

R2- 22. Related to comment for Figure 7: The authors indicate the following: "we show that Pou5f3 sequence-specific binding activates early zygotic genes by increasing chromatin accessibility and promoting H3K27 acetylation on Pou5f3- dependent and -codependent (CD) enhancers (Fig. 6g, Fig. 7)." There is actually no figure that includes both H3K27ac and ATAC-seq data. We feel in the scope of this paper, these signals could be included for example genes to support the authors statement.

A2-22. We agree. The ATAC-seq signals in four genotypes and H3K27ac are now on the Fig. 8b.

Reviewer #3 (Remarks to the Author):

The clarity of this revised manuscript by Gao, Veil et al. is much improved, but there are still areas requiring further improvement/clarification. The counterintuitive compensation of transcript levels in the Pou5f3, Sox19b double mutant as compared to the single mutants is intriguing. The authors present additional data more rigorously characterizing the Sox19b mutant phenotype. These data, included in Figures 1 and 2, are convincing and rigorously controlled. However, there remain some questions regarding the mechanistic conclusions that can be drawn from these data.

R3-1. Throughout the discussion of the RNA-seq time course, the authors fail to discern what effects are likely direct versus indirect, despite the fact that their RNA data sets extend over hours of development. The transcripts produced during the earliest wave of genome activation include transcription factors that affect expression later. Thus, it is likely that a proportion of the effects on transcript levels is not directly due to loss of Sox19b or Pou5f3 binding to the cis-regulatory regions of the genes identified by RNA-seq, but instead due to indirect effects of their earlier absence. At the end of the section describing the bulk transcription, the authors write, "In sum we demonstrated, that Pou5f3 and/or Sox19b activate 24% of the zygotic transcripts..." Whether or not the authors intended it this way, this sounds like these factors are directly responsible for this activation. Additional analysis trying to tease apart the direct from indirect effects would significantly strengthen the conclusions that can be drawn regarding the antagonistic functions of these two factors.

A3-1. We completely agree with the criticism, it was a weakest point in our paper. Hopefully it is not anymore. We performed ATAC-seq experiments in two stages (3.7 hpf and 4.3 hpf), in single and double mutants and the wild-type. We selected 102 945 accessible regions (ARs), present in the wild-type at both stages (ATAC-seq peaks, overlapping between each of the three replicates at 4.3 and 3.7 hpf, 6 replicates in total). We used the enrichment for sequence-specific motifs as a measure of direct binding, and tried to tease apart direct and indirect effects of the TFs (see the new Figures 5- 7, the corresponding text and the supplementary figures).

R3-2. Similarly, it is clear from Figure S3B that additional SoxB1 family members are expressed during this time course. These family members might be able to substitute for Sox19b at some loci later in development, yet the authors do not discuss these factors or how they might contribute to gene expression patterns.

A3-2. The zygotic family members indeed substitute for Sox19b; the normal development of MZsox19b mutant is explained by the presence of SoxB1 factors at later stages. Triple morpholino knockdown in MZsox19b on Fig.1e shows that the embryo does not develop normally after gastrulation, if they are knocked out. This phenotype can be rescued by Sox19b (Fig.1f), or by any of SoxB1 factors (Okuda 2010). It was previously shown, that all four SoxB1 factors act redundantly, and their target genes were extensively characterized (Okuda et al., 2006, Okuda et al., 2010). Our study does not add any new information to that. The purpose of our study was to

dissect the role of maternal *sox19b*, which was not addressed previously, and its interaction with Pou5f3 at the earliest zygotic stages, which was also not addressed previously. In response to this comment we included the explanatory sentence in p8, the second phrase in the sub-chapter "Changes in chromatin accessibility underlie the changes in zygotic expression repertoire in the double MZ*sox19b*spg mutant": "Zygotic expression of redundant SoxB1 family members Sox3, Sox19a and Sox2 at these stages is still low (Fig.S3a), which enabled us to the effects of maternal Sox19b on chromatin accessibility"

R3-3. In parts, the conclusions in the paper are overstated. The authors need to increase their clarity about whether the data upon which conclusions are based are in the manuscript, in previously published papers, or in both. For example, in the Discussion the speculation about binding of each factor in the absence of the other and the opposing roles is not based on any binding data since none of the data in this manuscript analyze binding of either factor.

A3-3. We agree with this criticism, and we removed the abovementioned speculation from our discussion

R3-4. Similarly, the first conclusion of the Discussion is that Pou5f3 and Sox19b modify chromatin independently even at regions where they bind together. This is not totally clear to me as the authors never investigate binding of each factor in the absence of the other.

A3-4. We agree, we did not investigate binding. We investigated the changes of chromatin accessibility on the cognate motifs in single mutants, and now also in the double mutants. In response to this comment, we reformulated the second paragraph of the discussion, to make it clear: "First, on the mechanistic level, we found that Sox19b and Pou5f3 are both involved in establishment of chromatin accessibility at blastula stages and act mostly independently. Sox19b promotes accessibility on *sox* motifs, Pou5f3 on *pou* and *sox:pou* motifs (Fig. 6d-f). Sox19b and Pou5f3 can act additively or redundantly if their cognate motifs are present nearby (Fig. S7g-i). Out of four different scenarios for the interactions of the mammalian homologues in ES cells, which were outlined in the introduction²³⁻²⁶, our results agree with the conclusions of the last study: POU5F1 and SOX2 operate in a largely independent manner even at co-occupied sites²⁶"

R3-5. Here and throughout the manuscript, the authors need to be cautious about the broad conclusions they are drawing from the data included and their analysis of these data. Some of the mechanistic conclusions are severely limited because of lack of data in the double mutant. While it is understandable that the MNase and H3K27ac ChIP data may not be able to be generated in the double mutant, the authors must acknowledge that the mechanistic conclusions regarding the compensation are limited by the absence of these data.

A3-5. This is a very valid comment. We agree. We performed the complete set of ATAC-seq experiments in all mutants, including the double mutant, and in two developmental stages to address these concerns (main Fig. 5-7, corresponding text and supplementary data).

R3-6. The use of single replicates for the H3K27ac ChIP-seq for both WT and MZspg is concerning. While it is understandable that this material is challenging to generate, reproducibility between replicates is important to confirm.

A3-6. We could not make the second replicate for WT and MZspg because of the technical reason – our Covaris sonifier machine is irreversibly broken, buying the new one did not work for us. This prompted us to make ATAC-seq and draw all our main conclusions from ATAC-seq experiments. We restructured the manuscript in such a way that the whole H3K27ac data set can be excluded, if this reviewer will find it necessary. However, we hope that we still can show H3K27ac data, because they perfectly agree with ATAC-seq (Fig. S9c,d). On the four types of direct cis-regulatory elements which we define based only on ATAC-seq, the same transcription factor which was essential for chromatin accessibility was also essential for H3K27ac (Fig.7e, compare with Fig.7b).

R3-7. Supplemental figures showing the reproducibility between replicates of RNA-seq and MNase-seq should be included.

A3-7. Done. The reproducibility between RNA-seq replicates is shown on Fig. S4a. We replaced MNase-seq with ATAC-seq data set; the reproducibility between ATAC-seq replicates is shown on Fig. S6a.

Minor:

R3-8. The citation of Fernandez-Garcia for the binding of Zelda to nucleosomes is correct. However, McDaniel et al. Mol Cell 2019 published these results initially and have more detailed analysis of this binding. This citation should be included along with the current citation 17.

A3-8. Done.

R3-9. There is typo in the third paragraph Introduction. "POU5f1" should be "POU5F1".

A3-9. Corrected.

R3-10. The authors discuss both in the Introduction and Discussion that histone levels serve as general transcriptional repressors before ZGA and that the competition between histones and transcription factors time ZGA. While the papers they cite certainly provide evidence that histone levels play a role, as written these statements are too strongly worded. There certainly remain a number of genes whose regulation is not directly controlled by this competition, but rather these observations may also include indirect effects of histone levels on cell-cycle timing.

A3-10. We agree. As we did not see the correlation between global developmental delay in *MZsox19b* and chromatin accessibility by ATAC-seq in *MZsox19b* (Fig. S8g), we removed the most speculative part of discussion, where the histone levels were mentioned.

R3-11. The authors explain the difference in their analysis from that previously published by Lee et al. as due to the fact that the mutant generated eliminates maternally provided Sox19b protein. What evidence is there of a pool of maternally provided protein? Given that Sox19b was implicated in genome activation in the Lee et al. paper based on having high translation rates in the early embryo, it seems unlikely that there is also a significant amount of maternally supplied protein.

A3-11. We removed this part of the discussion. However, we would like to mention that although there is no direct evidence for maternally supplied Sox19b protein, there are two indirect pieces of evidence. First, we saw maternal protein it by Western Blot in SoxB1 QKD morpholino knockdown with pan-SoxB1 antibodies, at MBT (1024 cells) and before MBT (512 cell stage, Leichsenring et al., 2013, Fig.S8, G,H, see QKD lane). One can of course argue, that the pre-MBT translation was not completely suppressed by morpholinos, therefore the second piece of evidence: there are obvious maternal differences in RNA-seq at 2.5 hpf (before MBT), between *MZsox19b* and WT (Fig. S4a, PCA at 2.5 hpf). Lee et al (2013) made ribosome profiling, this technique does not give an information about pre-existing proteins: the maternal protein can be there.

R3-12. The authors should avoid using red and green in their heatmaps so that they are interpretable for color blind readers.

A3-12. Done (Fig. 3b, Fig. S4c)

R3-13. Explanation of the headings for the columns in the Supplemental Tables should be provided in the legends for these tables.

A3-13. Done. We included the "legend" spreadsheets in the Tables S1-S5.

R3-14. Movie S2 explaining the RNA-sense did not work for me.

A3-14. Our fault. We reformatted the movie, now it should work.

Reviewer #4 (Remarks to the Author):

R4-1. Gao et al restructured the paper and added some new experiments/analyses to examine the function of Sox19b and Pou5f3 in zebrafish ZGA. However, several major conclusions of this study are still insufficiently supported by their data, and it seems that the study lacks significance and novelty in its present form. Readability of the manuscript has been improved, but it is still difficult to relate the phenotypic analyses (Fig 1,2) to molecular analyses (Fig 3~7). Below are major concerns.

A4-1. We agree. We performed a new set of ATAC-seq experiments to address the concerns (see the new Figures 5-8, text and supplementals)

R4-2. Their conclusion that the balance of Pou5f3 and Sox19b is important for zygotic transcription is not well supported. Although the level of bulk zygotic transcription at 3.5hpf is higher in *MZsox19b*spg than in *MZsox19b* and *MZspg*, the difference is very subtle (Fig S5a middle), and as shown in Fig S5a (left), *MZsox19b*spg and *MZspg* show almost same dynamics. The percentage of downregulated genes in *MZsox19b*spg and *MZspg* does not seem significant (24% vs 32%). In Fig 3b, only 364 genes (in group B) shows downregulation in single mutants. So they can only say that the balance is important only for those 364 genes.

A4-2. We agree. We shortened the paragraph and remove the statement about the balance of all genes (marked at p.6). We elaborate on the differences in transcription between the double mutants and *MZspg*: groups B (364 transcripts) and E (255 transcripts) are downregulated in *MZspg*, but not in double mutants (619 transcripts in total, Fig 3b). Thus, the balance is important for those 619 transcripts. The chromatin on the "compensated" enhancers of those genes is also

less accessible only in MZspg but not in double mutants (“down in MZspg not double”, Fig. S8f). This means, that there is an antagonistic action of Pou5f3 and Sox19b on chromatin, direct or indirect. As the “compensated” enhancers are not enriched in the motifs for both Pou5f3 and Sox19b (Fig. S8c), we conclude that antagonistic interaction between these proteins are indirect (marked in p.10, Fig. S8c,f).

R4-3. At the global level, it seems that MZsox19bspg and MZspg do not show a great difference to each other. They also do not show double mutant data in other analyses (MNase and H3K27ac), and thus the underlying mechanism is completely unknown.

A4-3. In response to this criticism, we made ATAC-seq experiments in all genotypes. Now, we show the data for the double mutant. We show, that the transcriptional difference between MZspg and MZsox19bspg is explained by indirect antagonistic action of Pou5f3 and Sox19b on chromatin. We also show, that the global transcriptional delay in MZsox19b does not correlate with changes in chromatin accessibility in MZsox19b (marked at p.10, Fig. S8c-g).

R4-4. Why do the authors conclude that only Sox19b contributes to chromatin accessibility? The result of Fig 5d shows that nucleosomes at SP peaks are displaced also in MZspg, suggesting that the accessibility also depends on Pou5f3.

A4-4. We apologize if it sounded like that: we did not conclude that only Sox19b contributes to chromatin accessibility. We resolved the question now in the sub-chapter “Sox19b and Pou5f3 act as independent pioneer factors” p9, and following text.

R4-5. Furthermore, the relationship between SP peaks and H3K27ac clusters are not clear. The conclusion shown in Fig. 8a is not appropriate.

A4-5. We agree with this criticism and removed the scheme in Fig.8a.

R4-6. (Fig 7) The paragraph describing the association between enhancers and transcription is difficult to follow.

A4-6. We agree. The Fig.7 is replaced, the paragraph is removed.

R4-7. The data presented are not supportive enough to conclude that Sox19b-dependent enhancers are not involved in the transcriptional activation. SD-linked genes are downregulated in MZsox19b, and thus, it is likely that these genes are activated Sox19b. H2K27ac ChIP for MZsox19bspg is also required.

A4-7. We don't agree. We analyzed the new independent ATAC-seq data set for the single and double mutants. We still see that Sox19b-dependent enhancers are not involved in transcriptional activation at ZGA (group 3 in the new Fig. 7d). As for transcription: half of all transcripts were downregulated in MZsox19b. The part of transcripts which was downregulated in MZsox19b only and not in the other mutants is a large group C on the Fig. 3b (1064 transcripts). This group did not correlate with H3K27ac changes in MZsox19b (the previous variant of our manuscript). Group C also did not correlate with ATAC-seq changes in MZsox19b (the current version of our manuscript, Fig. S8g), and it also did not correlate with Sox19b-dependent enhancers (we checked that, data not shown, but it can be easily checked using ChI-squared test and Table S5).

R4-8. The relationship between the first two figures (Fig1,2) and the latter (Fig 3~7) is still unclear. What is the molecular basis of the developmental delay? Do they think that the group B genes downregulated only in single mutants are responsible for this?

A4-8. We agree with the criticism, and we removed the point about the differences of the developmental delays in the double mutants and MZspg from the text. We focus on the differences in transcription between the double mutants and MZspg: groups B (364 transcripts) and E (255 transcripts) are downregulated in MZspg, but not in double mutants (619 transcripts in total, Fig 3b). Chromatin on the enhancers of those genes is also less accessible only in MZspg but not in double mutants (“down in MZspg not double”, Fig. S8f). Taken together, we explain the compensatory effects on groups B and E by indirect antagonistic action of Pou5f3 and Sox19b on chromatin. The developmental delay in MZsox19b is obvious, both at the level of the phenotype and at the level of transcription. Group C (1064 transcripts, Fig. 3b) does not correlate with chromatin changes in MZsox19b (Fig. 8g). We still do not know why MZsox19b single mutants are delayed, but we at least excluded one of the possible mechanisms.

R4-9. Figure 3b shows that a fraction of zygotic genes are not activated, rather than delayed.
A4-9. We agree. We added "Analysis of differential expression revealed delays in zygotic transcription (or absence of transcription for some genes)..." in p.6.

R4-10. The authors claim that the MZsox19bspq mutants were less delayed, but the data presented are not clear (Fig 2f). The movie is difficult to compare the subtle differences. They should show quantitative data.

A4-10. We agree, this is a valid comment. We removed all the claims about comparative phenotypic delays.

Reviewers' Comments:

Reviewer #3:

Remarks to the Author:

The resubmitted manuscript from Gao, Veil et al is much improved in both clarity and in the data included. The addition of the ATAC-seq from both single and double mutants allowed the authors to address important issue regarding direct and indirect effects in their prior submission. We are now satisfied with how the authors addressed our prior concerns.

We urge the authors to refocus the abstract on the data presented in the manuscript. The % of accessible regions and the increase in accessibility do not appear to be the major conclusions from the manuscript, as written. The increased accessibility at GC-rich enhancers is only one small part of the 8 figures included in the manuscript.

In Figure 3, the figure legend has MZsox19bsp, but in the figure it is labelled double. It is worth keeping the legend and figure consistent for clarity.

In Figure 5c, the hash mark near the 27 on the scale is confusing as it looks as if this is a negative 27. Small edits would make this more clear.

Reviewer #4:

Remarks to the Author:

The manuscript by Gao et al. has been greatly improved with new data, and they addressed all of my previous concerns. The study revealed how pou5f3 and Sox19b function during ZGA in zebrafish embryos, and has an impact in the field of early development and pluripotency. Thus, the manuscript is now ready for publication.

Dear reviewers,

We are happy that we could address your concerns! We answer three critical suggestions of the Reviewer 3 below:

REVIEWERS' COMMENTS

Reviewer #3 (Remarks to the Author):

The resubmitted manuscript from Gao, Veil et al is much improved in both clarity and in the data included. The addition of the ATAC-seq from both single and double mutants allowed the authors to address important issue regarding direct and indirect effects in their prior submission. We are now satisfied with how the authors addressed our prior concerns.

R3-1. We urge the authors to refocus the abstract on the data presented in the manuscript. The % of accessible regions and the increase in accessibility do not appear to be the major conclusions from the manuscript, as written. The increased accessibility at GC-rich enhancers is only one small part of the 8 figures included in the manuscript.

A3-1. We agree. We modified the abstract by excluding the % and CG content, and included the following two sentences instead:

We distinguish four types of direct enhancers by differential requirements for Pou5f3 or Sox19b. We demonstrate that changes in chromatin accessibility of enhancers underlie the changes in zygotic expression repertoire in the double mutants.

R3-2. In Figure 3, the figure legend has MZsox19bsp, but in the figure it is labelled double. It is worth keeping the legend and figure consistent for clarity.

A3-2. We changed "MZsox19bsp" to "double" in the Fig.3 legend

In Figure 5c, the hash mark near the 27 on the scale is confusing as it looks as if this is a negative 27. Small edits would make this more clear.

A3-3. Thank you for noticing this. Done.

Reviewer #4 (Remarks to the Author):

The manuscript by Gao et al. has been greatly improved with new data, and they addressed all of my previous concerns. The study revealed how pou5f3 and Sox19b function during ZGA in zebrafish embryos, and has an impact in the field of early development and pluripotency. Thus, the manuscript is now ready for publication.

** See Nature Research's author and referees' website at www.nature.com/authors for information about policies, services and author benefits

This email has been sent through the Springer Nature Tracking System NY-610A-NPG&MTS

Confidentiality Statement:

This e-mail is confidential and subject to copyright. Any unauthorised use or disclosure of its contents is prohibited. If you have received this email in error please notify our Manuscript Tracking System Helpdesk team at <http://platformsupport.nature.com>.

Details of the confidentiality and pre-publicity policy may be found here <http://www.nature.com/authors/policies/confidentiality.html>